# TropNNC: Structured Neural Network Compression Using Tropical Geometry

**Konstantinos Fotopoulos**[*]                                           *kostfoto2001@gmail.com*
*School of ECE, National Technical University of Athens, Greece*

**Petros Maragos**                                                      *maragos@cs.ntua.gr*
*School of ECE, National Technical University of Athens, Greece*
*Institute of Robotics, Athena Research Center, Maroussi, Greece*
*HERON - Hellenic Robotics Center of Excellence, Athens, Greece*

**Panagiotis Misiakos**                                                 *pmisiakos@ethz.ch*
*Department of Computer Science, ETH Zurich, Switzerland*

**Reviewed on OpenReview:** *https://openreview.net/forum?id=u7DRq1icmY*

## Abstract

We present TropNNC, a framework for compressing neural networks with linear and convolutional layers and ReLU-type activations using tropical geometry. By representing a network's output as a tropical rational function, TropNNC enables structured compression via reduction of the corresponding tropical polynomials. Our method identifies redundancy via similarity and improves upon the geometric approximation of previous work by adaptively selecting the weights of retained neurons. We relate it to SVD and spectral clustering, and provide insights into network compression beyond the specific setting considered. We provide the tightest known theoretical compression bound, and the first successful application of tropical geometry to convolutional layers. TropNNC requires access only to network weights – no training data – and achieves competitive performance on MNIST, CIFAR, and ImageNet, matching strong baselines such as ThiNet and CUP.

## 1 Introduction

Deploying deep neural networks on resource-constrained devices remains challenging due to their computational and storage demands. Most methods for reducing network complexity are based on pruning via estimated importance scores. These include unstructured pruning, which alters the network's structure by eliminating individual unimportant weights, presenting challenges in practical applications (Wen et al., 2016). To overcome these, structured pruning methods, such as channel-level pruning, has been proposed (He & Xiao, 2024). These estimate the importance of entire channels, pruning them accordingly and maintaining the network's structure. This structured approach ensures compatibility with existing deep learning libraries and offers several advantages: it reduces memory footprint, and facilitates further compression and acceleration through methods like parameter quantization. The most widely used among them, such as ThiNet (Luo et al., 2017) NISP (Yu et al., 2018) and HRank (Lin et al., 2020), are data-driven: they make use of a dataset to obtain distributional information guiding the estimation of importance scores. On the other hand, there exist SVD compression methods (Marinó et al., 2023). These apply SVD and low-rank factorizations to the weight matrices of the network. They are mostly data-free and do not rely on importance scores. However, the factorization doubles the number of layers, and leaves the width of half of them unchanged.

Parallel to these advancements, tropical geometry (Maclagan & Sturmfels, 2021) has emerged as a promising mathematical framework with applications in machine learning (Maragos et al., 2021; Gärtner & Jaggi,

---

[*]Corresponding author.

2008) and beyond. Recently, it has been applied to the theoretical study of neural networks. For example, Zhang et al. (2018) demonstrated the equivalence of ReLU-activated neural networks with tropical rational mappings. Together with other works, like those of Charisopoulos & Maragos (2018); Alfarra et al. (2023), they used tropical geometry to compute bounds on the number of linear regions of neural networks equal to the one in (Montufar et al., 2014). Smyrnis & Maragos (2020) use tropical geometry for pruning.

**Contributions.** In this paper, we explore the application of tropical geometry in the compression of ReLU-type neural networks. Our contributions include:

- Proposing TropNNC, an algorithm that leverages tropical geometry and the Hausdorff distance for the structured compression of neural networks. TropNNC compresses networks layer-wise by approximating the zonotopes corresponding to each layer using an iterative process, finding redundancy via similarity instead of importance. Our algorithm is data-free: it does not require a training dataset.
- Connecting our iterative algorithm to SVD and spectral clustering, providing a refinement to the bound for the functional approximation of tropical polynomials presented in (Misiakos et al., 2022), and using it to handle ReLU-type activations and provide strong theoretical compression guarantees.
- Relating and comparing our theoretical framework to broader pruning approaches, and providing insights into network compression.
- Evaluating our algorithm empirically on the MNIST, CIFAR and ImageNet datasets, applying it also successfully to convolutional layers. Our method outperforms prior tropical geometric pruning methods, and achieves competitive performance compared to the data-driven ThiNet, and superior performance compared to CUP (Duggal et al., 2021), particularly in the VGG architecture.

This work demonstrates the potential of tropical geometry in enhancing neural network compression techniques. To the best of our knowledge, it is the first tropical geometric pruning method that is competitive with strong baselines and applied to the convolutional layers of networks of non-trivial size. It is also one of the few compression methods that come with theoretical guarantees. Even if not currently state-of-the-art, we believe our contribution is a necessary step in this direction. Proofs are provided in the appendix.

## 2 Related Work

Our work falls under the category of structured pruning (He & Xiao, 2024). Here, approaches include data-free norm-based pruning (Li et al., 2017; He et al., 2019), KL-divergence based pruning (Luo & Wu, 2020), data-driven activation-based pruning (Lin et al., 2020; Sui et al., 2021), and data-driven filter pruning based on reconstruction loss minimization. Within the latter category, ThiNet (Luo et al., 2017) is a representative method that greedily removes filters contributing least to the next layer's input, based on a data-driven reconstruction objective. Our approach resembles ThiNet in its structured layer-wise compression, but unlike ThiNet, it is data-free. Although layer-wise pruning is increasingly being replaced by global strategies – such as NISP (Yu et al., 2018) and SPvR (Hassan et al., 2025), which consider the importance of filters relative to the final layer – ThiNet remains a strong structured data-driven baseline to which we compare our method. More recently, data-free methods have been proposed. For example, CUP (Duggal et al., 2021) hierarchically clusters similar channels and selects representatives by $L^1$ norm. Our method builds on the same intuition of redundancy reduction via clustering, but introduces three key extensions: (i) modifying the clustering vectors, (ii) adapting the distance thresholds of hierarchical clustering to layer dimensions, and (iii) using tropical geometry to select representatives.

We build on Misiakos et al. (2022), who used a discrete form of the Hausdorff distance to geometrically bound the error between tropical polynomials, framing network reduction as a zonotope order reduction problem, and proposing a basic clustering-based compression method. Their method clusters similar neurons and replaces each cluster with the representative obtained by taking the mean of the input and output weights of the cluster. We refine their central theorem using the standard Hausdorff distance and introduce a novel algorithm that formulates compression as a *simultaneous zonotope approximation* problem. Our method makes a better initial estimation of the compressed network's weights, and refines them by an iterative process. Our approach leads to provably stronger theoretical guarantees. We also do not limit experiments to fully connected layers, and instead handle both convolutional layers and batch normalization.

We draw parallels to SVD and spectral clustering. Usual SVD network compression (Marinó et al., 2023) low-rank factorizes matrices, doubling the depth and leaving the width of a lot of layers unchanged. Instead, we use tropical geometry to allow us to handle consecutive matrices as one, even with non-linear activation between them, which to the best of our knowledge differentiates our method from prior work.

## 3 Tropical Geometry of Neural Networks

Tropical algebra studies matrix-vector operations based on the arithmetic of the tropical semiring (Cuninghame-Green, 1979; Butkovič, 2010). Tropical geometry is the counterpart of algebraic geometry in the tropical setting. The tropical semiring can refer to either the *min-plus semiring* or the *max-plus semiring*. In this work, we adhere to the convention of using the *max-plus semiring* $(\mathbb{R}_{\max}, \vee, +)$, defined as the set $\mathbb{R}_{\max} = \mathbb{R} \cup \{-\infty\}$ equipped with two binary operations: $\vee$ (ordinary max) and $+$ (ordinary sum).

Within the max-plus semiring, we can define tropical polynomials, which correspond to convex piecewise linear functions, and tropical rational functions, which correspond to general continuous piecewise linear functions. We can also define Newton polytopes of tropical polynomials, which connect tropical algebra with polytope theory. For a detailed introduction to these concepts refer to Appendix A.2.

### 3.1 Neural Networks with Piecewise Linear Activations

Tropical geometry provides a mathematical framework for analyzing neural networks with piecewise linear activation functions. In this work, we primarily focus on ReLU-activated networks, with additional activations handled later.

**ReLU Activations.** Consider a network which consists of an input layer $\mathbf{x} = (x_1, \ldots, x_d)$, a hidden layer $\mathbf{f} = (f_1, \ldots, f_n)$ of ReLU units, and an output layer $\mathbf{v} = (v_1, \ldots, v_m)$. The input, hidden, and output layers are connected through linear transformations represented by matrices $\mathbf{A}$ and $\mathbf{C}$. Each neuron $i$ has input weights and bias given by $\mathbf{A}_{i,:} = (\mathbf{a}_i^T, b_i)$ and output weights $\mathbf{C}_{:,i}^T = (c_{1i}, \ldots, c_{mi})$. We assume the output layer has no bias. Such a network is depicted in Figure 1.

The output of the ReLU unit $i$ is given by:

$$f_i(\mathbf{x}) = \text{ReLU}(\mathbf{a}_i^T \mathbf{x} + b_i) = \max\{\mathbf{a}_i^T \mathbf{x} + b_i, 0\}.$$

This expression represents a tropical polynomial of rank 2, with one term being the constant 0. The extended Newton polytope $\text{ENewt}(f_i)$ of $f_i$ is an edge with one endpoint at the origin $\mathbf{0}$ and the other endpoint at $(\mathbf{a}_i^T, b_i)$. The $j$-th component of the output layer $v_j$ can be computed as follows:

$$v_j = \sum_{i \in [n]} c_{ji} f_i = \sum_{i:c_{ji}>0} |c_{ji}| f_i - \sum_{i:c_{ji}<0} |c_{ji}| f_i = p_j - q_j.$$

In the above expression, $|c_{ji}| f_i$ are tropical polynomials. Thus, $p_j$ and $q_j$ are tropical polynomials formed by the addition of tropical polynomials. Consequently, $v_j$ is a tropical rational function. We call $p_j$ the *positive polynomial* and $q_j$ the *negative polynomial* of $v_j$.

**Zonotopes.** The extended Newton polytope of $|c_{ji}| f_i$ is an edge with one endpoint at the origin $\mathbf{0}$ and the other at $|c_{ji}|(\mathbf{a}_i^T, b_i)$. Using Proposition A.1, the extended Newton polytope $P_j$ of $p_j$ is the Minkowski sum of the *positive* generators $\{|c_{ji}|(\mathbf{a}_i^T, b_i) : c_{ji} > 0\}$, and the polytope $Q_j$ of $q_j$ is the Minkowski sum of the *negative* generators $\{|c_{ji}|(\mathbf{a}_i^T, b_i) : c_{ji} < 0\}$. Thus, $P_j, Q_j$ are zonotopes (see Appendix A.2 for a definition). We refer to $P_j$ as the *positive zonotope* and $Q_j$ as the *negative zonotope* of $v_j$.

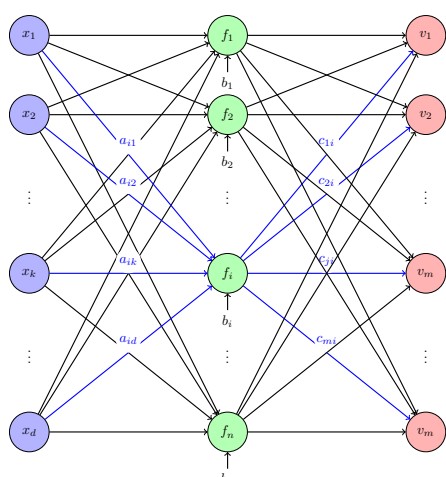

Figure 1: Neural network with one hidden ReLU layer. The first linear layer has weights $\{\mathbf{a}_i^T\}$ with bias $\{b_i\}$ corresponding to node $i \in [n]$ and the second has weights $\{c_{ji}\}$ between nodes $j \in [m], i \in [n]$.

## 4 Approximation based on Hausdorff distance

In this section, we present our refined theorem, which uses the Hausdorff distance in its standard continuous form to bound the error between two tropical polynomials. In the following, the distance between two points $\mathbf{u}$ and $\mathbf{v}$ is denoted as $\text{dist}(\mathbf{u}, \mathbf{v}) := \|\mathbf{u} - \mathbf{v}\|$, where $\|\cdot\|$ denotes the standard $L^2$ Euclidean norm. The distance between a point $\mathbf{u}$ and a set $V$ is defined as $\text{dist}(\mathbf{u}, V) = \text{dist}(V, \mathbf{u}) := \inf_{\mathbf{v} \in V} \|\mathbf{u} - \mathbf{v}\|$.

**Definition 1** (Hausdorff distance). *Let $S, \tilde{S}$ be two subsets of $\mathbb{R}^d$. The Hausdorff distance $H(S, \tilde{S})$ of the two sets is defined as*

$$H(S, \tilde{S}) := \max \left\{ \sup_{\mathbf{u} \in S} \text{dist}(\mathbf{u}, \tilde{S}), \sup_{\mathbf{v} \in \tilde{S}} \text{dist}(S, \mathbf{v}) \right\}.$$

In the case of polytopes $P, \tilde{P}$, due to their convexity and compactness, the suprema in the above expression are attained, and in fact by points in the vertex sets $V_P, V_{\tilde{P}}$ of the polytopes (see Lemma A.4).

Misiakos et al. (2022) used the discrete form of the Hausdorff distance, defined as the Hausdorff distance of the vertex sets of the two polytopes ($DH(P, \tilde{P}) := H(V_P, V_{\tilde{P}})$), to bound the error of polynomial approximation. We refine their result using the Hausdorff distance in its standard form.

**Theorem 2.** *Let $p, \tilde{p} \in \mathbb{R}_{\max}[\mathbf{x}]$ be two tropical polynomials with extended Newton polytopes $P = \text{ENewt}(p)$ and $\tilde{P} = \text{ENewt}(\tilde{p})$. Then,*

$$\frac{1}{\rho} \max_{\mathbf{x} \in B} |p(\mathbf{x}) - \tilde{p}(\mathbf{x})| \leq H(P, \tilde{P})$$

*where $B = \{\mathbf{x} \in \mathbb{R}^d : \|\mathbf{x}\| \leq r\}$ and $\rho = \sqrt{r^2 + 1}$.*

Since $V_P \subseteq P, V_{\tilde{P}} \subseteq \tilde{P}$, we have that $\text{dist}(\mathbf{u}, \tilde{P}) \leq \text{dist}(\mathbf{u}, V_{\tilde{P}}), \text{dist}(P, \mathbf{v}) \leq \text{dist}(V_P, \mathbf{v})$. The equality in the aforementioned inequalities is achieved only in special cases and thus the bound $H(P, \tilde{P})$ we provide is in general tighter than $DH(P, \tilde{P})$ from Misiakos et al. (2022).

By applying the triangle inequality and Theorem 2 we obtain the following result for two neural networks of compatible input and output dimension.

**Corollary 3.** *Let $\mathbf{v}$ and $\tilde{\mathbf{v}}$ be the outputs of two neural networks as in Figure 1. Then, the following inequality holds:*

$$\frac{1}{\rho} \max_{\mathbf{x} \in B} \|\mathbf{v}(\mathbf{x}) - \tilde{\mathbf{v}}(\mathbf{x})\|_1 \leq \sum_{j=1}^{m} \left( H(P_j, \tilde{P}_j) + H(Q_j, \tilde{Q}_j) \right),$$

*where $\|\cdot\|_1$ denotes the $L^1$ norm.*

## 5 Compression Algorithm

From Cor. 3 it is evident that to compress a ReLU network consisting of a pair of consecutive linear layers, one has to choose compressed weight matrices $\widetilde{\mathbf{A}} \in \mathbb{R}^{K \times (d+1)}$ and $\widetilde{\mathbf{C}} \in \mathbb{R}^{m \times K}$ with $K < n$ such that $\widetilde{P}_j \approx P_j$ and $\widetilde{Q}_j \approx Q_j$ for all $j \in [m]$, where the approximation relation is in terms of the Hausdorff distance between the zonotopes.

### 5.1 Single output

If we have $m = 1$ then the problem can readily be translated into a *zonotope approximation problem*—the task of approximating a zonotope with another zonotope that has fewer generators. This problem is known as *zonotope order reduction* (Yang & Scott, 2018). In our case, the approximation must happen in terms of the Hausdorff distance.

**Algorithm for single output network.** For single-output networks, our approach uses K-Means to cluster positive and negative generators, and replaces each cluster with a single representative. Unlike

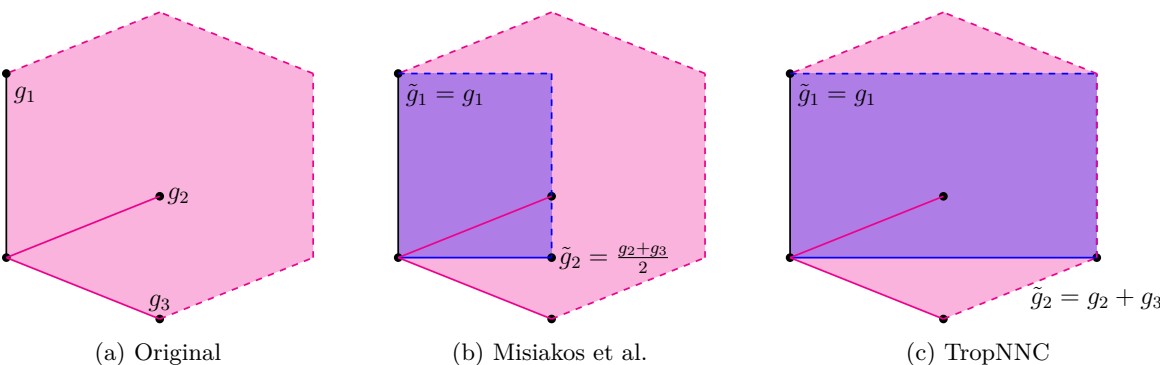

Figure 2: Example execution of TropNNC compared to the method of Misiakos et al. (2022). The original zonotope in Figure 2a corresponds to a network with 3 neurons, each providing one of its generators. To compress the hidden layer to 2 neurons, we equivalently seek to approximate the original zonotope by zonotopes of order 2. We cluster the generators/neurons into 2 clusters based on similarity. Misiakos et al. (Fig. 2b) replace the generators of a cluster by the cluster center. Instead, TropNNC (Fig. 2c) replaces the cluster by the sum of the generators of the cluster, leading to a better zonotope approximation.

Misiakos et al. (2022), who define the representative as the center/mean of the cluster, we achieve a better zonotope approximation by taking the representative to be the sum of the cluster.

Our Algorithm 1 is depicted below. Figure 2 illustrates an example execution of TropNNC compared to the method of Misiakos et al. (2022). The refined Theorem 2 allows us to take projections instead of directly comparing vertices. By taking a longer representative via the sum, the projection (and distance) of "one term" vertices remains constant, while the distance from "multiple term" vertices reduces, giving an advantage to our algorithm. For our algorithm, the following bound and its corollary hold.

---

**Algorithm 1** TropNNC, Single output

1: Split the generators $|c_i|(\mathbf{a}_i^T, b_i)$ into positive and negative generators:

$$\{|c_i|(\mathbf{a}_i^T, b_i) : c_i > 0\},$$
$$\{|c_i|(\mathbf{a}_i^T, b_i) : c_i < 0\}.$$

2: Execute K-means with $K/2$ centers on the positive generators $\{|c_i|(\mathbf{a}_i^T, b_i) : c_i > 0\}$, and $K/2$ centers on the negative generators $\{|c_i|(\mathbf{a}_i^T, b_i) : c_i < 0\}$.

3: Obtain positive and negative cluster representatives:

$$\{|\tilde{c}_i|(\tilde{\mathbf{a}}_i^T, \tilde{b}_i) : i \in C_+\},$$
$$\{|\tilde{c}_i|(\tilde{\mathbf{a}}_i^T, \tilde{b}_i) : i \in C_-\},$$

where $C_+ \sqcup C_- = [K]$ and $|\tilde{c}_i|(\tilde{\mathbf{a}}_i^T, \tilde{b}_i)$ is the **sum of the generators** of cluster $i$.

4: For each $i \in [K]$ construct a (hidden layer) neuron with input weights and bias $|\tilde{c}_i|(\tilde{\mathbf{a}}_i^T, \tilde{b}_i)$.

5: For each constructed neuron $i$ set the output weight to 1 if the neuron corresponds to a representative of a positive cluster ($i \in C_+$), otherwise set it to -1.

---

**Theorem 4.** *Single-output TropNNC with $K$ clusters on a network with output $v$ produces a neural network with output $\tilde{v}$ satisfying:*

$$\frac{1}{\rho} \max_{\mathbf{x} \in B} |v(\mathbf{x}) - \tilde{v}(\mathbf{x})| \le \sum_{i=1}^{n} \min\{||c_i(\mathbf{a}_i^T, b_i)||, \delta_{max}\},$$

*where $\delta_{max}$ is the largest distance from a generator to its corresponding cluster center.*

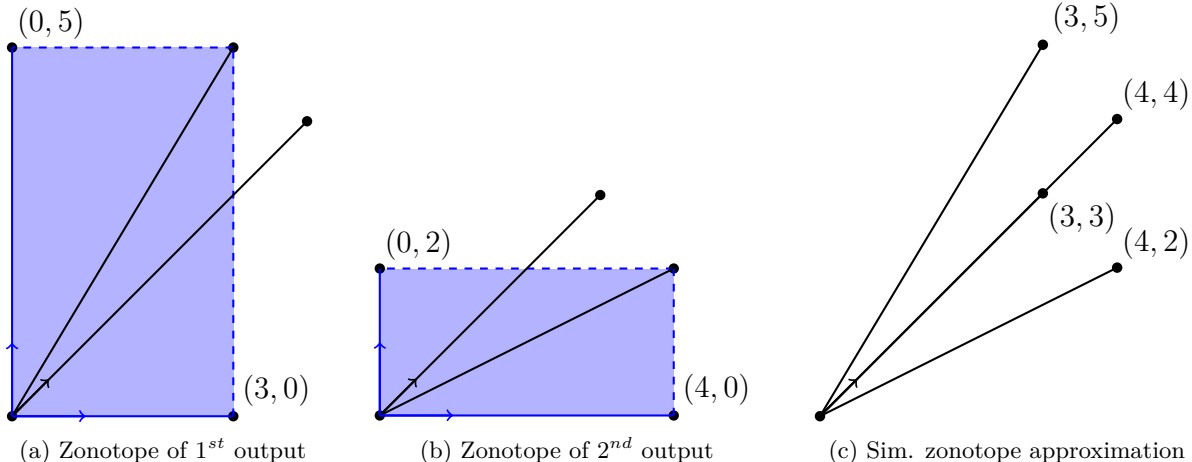

Figure 3: Example of simultaneous zonotope approximation for a network with 2 outputs and 2 hidden neurons. With multiple zonotopes, one is forced to choose a common direction between the ideal diagonals.

**Corollary 5.** *The above bound is tighter than the bound of Zonotope K-means of Misiakos et al. (2022).*

Note that a smaller $\delta_{max}$ indicates more compact clusters and greater similarity between the neurons of each cluster. In such cases, we have high redundancy, and Theorem 4 guarantees a small approximation error for TropNNC, i.e. TropNNC is well-behaved. This contrasts the situation in (Misiakos et al., 2022), where the respective bound of their algorithm was found to not be monotonic. We believe this is not merely an artifact of their analysis. The refinement of Theorem 2 is what enabled the above results.

### 5.2 Multi-output

We now consider the multi-output case with $m \in \mathbb{N}$. Notice an interesting property of the zonotopes $P_j, Q_j$: they share the directions $(\mathbf{a}_i^T, b_i)$ of their generators. For instance, output $v_1$ might have a positive generator $|c_{1i}|(\mathbf{a}_i^T, b_i)$ of zonotope $P_1$, while output $v_2$ might have a negative generator $|c_{2i}|(\mathbf{a}_i^T, b_i)$ of zonotope $Q_2$. These generators are parallel to each other, with common direction $(\mathbf{a}_i^T, b_i)$. Hence, our original positive and negative zonotopes have parallel generators, and we aim to approximate them with new positive and negative zonotopes, also with parallel generators. We refer to this complex approximation problem as *simultaneous zonotope approximation*.

**Example 1.** *Suppose we have a neural network with a single hidden layer as in Figure 1, with dimensions $d = 1, n = m = 2$. Consider input weights $(\mathbf{a}_1^T, b_1) = (1, 0), (\mathbf{a}_2^T, b_2) = (0, 1)$ and output weights $c_{11} = 3, c_{12} = 5, c_{21} = 4, c_{22} = 2$. In this example, for simplicity we took all output weights to be positive so that we only deal with positive zonotopes. The zonotopes of the two outputs will be two parallelograms with parallel edges, as illustrated in Figure 3. The zonotope of the first output is generated by $c_{11}(\mathbf{a}_1^T, b_1) = (3, 0)$ and $c_{12}(\mathbf{a}_2^T, b_2) = (0, 5)$, and of the second output by $c_{21}(\mathbf{a}_1^T, b_1) = (4, 0)$ and $c_{22}(\mathbf{a}_2^T, b_2) = (0, 2)$. Say we want to reduce the hidden neurons to $K = 1, \tilde{\mathbf{f}} = (\tilde{f}_1)$. If we could approximate each output's zonotope separately, we could simply apply the single output algorithm and approximate each parallelogram by its diagonal. However, these diagonals are not parallel to each other, and thus can not occur by a single hidden neuron $\tilde{f}_1$ with input weights $(\tilde{\mathbf{a}}_1^T, \tilde{b}_1)$. Instead, we have to choose a single common direction $(\tilde{\mathbf{a}}_1^T, \tilde{b}_1)$ for both output zonotopes. We can however choose a different magnitude for each output along this common direction. As will be presented in the algorithm below, for the common direction we choose the vector $(\tilde{\mathbf{a}}_i^T, \tilde{b}_i) = \frac{(\mathbf{a}_1^T, b_1) + (\mathbf{a}_2^T, b_2)}{2} = (0.5, 0.5)$. For each output $j$, $\tilde{c}_{j1}$ is chosen so that the edge is as close to the diagonal as possible. Specifically, we choose $\tilde{c}_{11} = 3 + 5 = 8$ and $\tilde{c}_{21} = 2 + 4 = 6$. The approximation procedure can be seen in Figure 3c.*

### 5.2.1 Non-iterative Algorithm for multi-output network.

For the multi-output case, we perform clustering of similar neurons based on the set of clustering vectors $\{(\mathbf{a}_i^T, b_i, \mathbf{C}_{:,i}), i \in [n]\}$, and replace each cluster with a single representative. Unlike Misiakos et al. (2022), who take the cluster center/mean as the representative, we form the representative as follows: For every cluster $k \in [K]$ with clustered neuron indexes $I_k$ and vectors $\{(\mathbf{a}_i^T, b_i, \mathbf{C}_{:,i}^T), i \in I_k\}$ take $(\tilde{\mathbf{a}}_k^T, \tilde{b}_k)$ to be the mean of $\{(\mathbf{a}_i^T, b_i), i \in I_k\}$, take $\widetilde{\mathbf{C}}_{:,k}^T$ to be the sum of $\{\mathbf{C}_{:,i}^T, i \in I_k\}$, and form the representative of the cluster $(\tilde{\mathbf{a}}_k^T, \tilde{b}_k, \widetilde{\mathbf{C}}_{:,k}^T)$. The complete procedure is shown in Algorithm 2.

The weights returned by Algorithm 2 bear resemblance to the neuron merging technique, as seen for example in (Kim et al., 2020), where it is used as a final step for compensating the pruned neurons of importance-based algorithms. However, here we derived it from a tropical geometric perspective. This allows us to analyze its properties more thoroughly. We coupled it with clustering. This transforms the method from a compensating step of an importance-based algorithm to a well-performing standalone method. As shown in the proof of Theorem 7 and discussed below, we have a connection to SVD, and the "quality" of the clustering (explicit in our case, implicit in the case of previous algorithms) directly affects the usefulness of the technique. Importance-based methods, however, do not prioritize the quality of the implicit clusters.

---

**Algorithm 2** (Non-iterative) TropNNC for Multi-output networks

---

1: Execute clustering on the clustering vectors $(\mathbf{a}_i^T, b_i, \mathbf{C}_{:,i}^T)$ for $i \in [n]$, forming $K$ clusters $\{(\mathbf{a}_i^T, b_i, \mathbf{C}_{:,i}^T) \mid i \in I_k\}$ for $k \in [K]$.

2: **For each $k \in [K]$, form the cluster representative $(\tilde{\mathbf{a}}_k^T, \tilde{b}_k, \widetilde{\mathbf{C}}_{:,k}^T)$ as follows**:

   (i) Compute $(\tilde{\mathbf{a}}_k^T, \tilde{b}_k)$ as the **mean of the input weights and biases** of the vectors in the cluster, i.e., the mean of the set $\{(\mathbf{a}_i^T, b_i) \mid i \in I_k\}$.

   (ii) Compute $\widetilde{\mathbf{C}}_{:,k}^T$ as the **sum of the output weights** of the vectors in the cluster, i.e., the sum of the set $\{\mathbf{C}_{:,i}^T \mid i \in I_k\}$.

3: Construct the new hidden layer:

   (i) For the input weights, set the $k$-th row of the weight-bias matrix to $(\tilde{\mathbf{a}}_k^T, \tilde{b}_k)$.

   (ii) For the output weights, set the $k$-th column to $\widetilde{\mathbf{C}}_{:,k}$.

---

### 5.2.2 Iterative Algorithm for multi-output network.

To improve the approximation of Algorithm 2, we formulate an optimization problem that takes the output of Algorithm 2 and with an iterative process produces weights that achieve a better simultaneous zonotope approximation.

Motivated by Algorithm 1, assuming the number of null neurons are few (see Definition A.5), we want in terms of every output $j$ the cluster representative $\tilde{c}_{jk}(\tilde{\mathbf{a}}_k^T, \tilde{b}_k)$ to be a close approximation to the cluster sum $\sum_{i \in I_k} c_{ji}(\mathbf{a}_i^T, b_i)$. Thus, for every cluster $k$, we have unknowns $(\tilde{\mathbf{a}}_k^T, \tilde{b}_k, \widetilde{\mathbf{C}}_{:,k}^T) = (\tilde{a}_{k1}, \ldots, \tilde{a}_{kd}, \tilde{b}_k, \tilde{c}_{1k}, \ldots)$, and we wish to find a solution which minimizes the following criterion:

$$\sum_{j=1}^{m} \left\| \tilde{c}_{jk}(\tilde{\mathbf{a}}_k^T, \tilde{b}_k) - \sum_{i \in I_k} c_{ji}(\mathbf{a}_i^T, b_i) \right\|^2, \tag{1}$$

where $m$ is the number of outputs, and $I_k$ is the set of neurons of cluster $k$.

The above optimization problem can be solved by means of iterative alternating minimization. Specifically:

1. Fixing the input weights $(\tilde{\mathbf{a}}_k^T, \tilde{b}_k)$ and minimizing with respect to $\widetilde{\mathbf{C}}_{:,k}$, the terms of the sum are independent and thus can be minimized separately. For each term of the sum, the optimal error occurs if we project the sum $\sum_{i \in I_k} c_{ji}(\mathbf{a}_i^T, b_i)$ onto $(\tilde{\mathbf{a}}_k^T, \tilde{b}_k)$. We have:

$$\tilde{c}_{jk} = \frac{\left\langle \sum_{i \in I_k} c_{ji}(\mathbf{a}_i^T, b_i), (\tilde{\mathbf{a}}_k^T, \tilde{b}_k) \right\rangle}{\left\| (\tilde{\mathbf{a}}_k^T, \tilde{b}_k) \right\|^2} \tag{2}$$

2. Fixing the output weights $\widetilde{\mathbf{C}}_{:,k}$, and minimizing with respect to the input weights $(\tilde{\mathbf{a}}_k^T, \tilde{b}_k)$, we take the derivative of criterion (1) with respect to $(\tilde{\mathbf{a}}_k^T, \tilde{b}_k)$ and set it to zero. We have:

$$2 \sum_{j=1}^m \left[ \tilde{c}_{jk}(\tilde{\mathbf{a}}_k^T, \tilde{b}_k) - \sum_{i \in I_k} c_{ji}(\mathbf{a}_i^T, b_i) \right] \tilde{c}_{jk} = 0 \Leftrightarrow$$

$$\Leftrightarrow (\tilde{\mathbf{a}}_k^T, \tilde{b}_k) \sum_{j=1}^m \tilde{c}_{jk}^2 = \sum_{j=1}^m \sum_{i \in I_k} c_{ji}(\mathbf{a}_i^T, b_i) \tilde{c}_{jk} \Leftrightarrow$$

$$(\tilde{\mathbf{a}}_k^T, \tilde{b}_k) = \frac{\sum_{j=1}^m \sum_{i \in I_k} c_{ji}(\mathbf{a}_i^T, b_i) \tilde{c}_{jk}}{\sum_{j=1}^m \tilde{c}_{jk}^2} \tag{3}$$

We can initialize the iteration with the representative obtained from Algorithm 2. The resulting procedure is detailed in Algorithm 3. We note that, in practice, the number of null generators is not negligible, and thus criterion (1) also constitutes a heuristic method. For this reason, the number of iterations should not be excessive. We provide Theorem 6 for the approximation error of our algorithm. The use of projections was once more critical for obtaining the bound.

---

**Algorithm 3** Iterative TropNNC for Multi-output networks

---

1: Form the initial cluster representatives $(\tilde{\mathbf{a}}_k^T, \tilde{b}_k, \widetilde{\mathbf{C}}_{:,k}^T)$ by executing the non-iterative algorithm.
2: **for** iter $= 1$ to $num\_iter$ **do**
3:    **for** $k \in [K]$ **do**
4:       **for** $j \in [m]$ **do**
5:          $\widetilde{c}_{jk} \leftarrow \frac{\langle \sum_{i \in I_k} c_{ji}(\mathbf{a}_i^T, b_i), (\tilde{\mathbf{a}}_k^T, \tilde{b}_k) \rangle}{\|(\tilde{\mathbf{a}}_k^T, \tilde{b}_k)\|^2}$
6:       **end for**
7:       $(\tilde{\mathbf{a}}_k^T, \tilde{b}_k) \leftarrow \frac{\sum_{j=1}^m \sum_{i \in I_k} c_{ji}(\mathbf{a}_i^T, b_i) \widetilde{c}_{jk}}{\sum_{j=1}^m \tilde{c}_{jk}^2}$
8:    **end for**
9: **end for**
10: Construct the new linear layer as in the non-iterative algorithm.

---

**Theorem 6.** *When run with $K$ clusters, a variant of Iterative TropNNC produces a neural network with output $\tilde{\mathbf{v}}$ satisfying:*

$$\frac{1}{\rho} \max_{\mathbf{x} \in B} \|\mathbf{v}(\mathbf{x}) - \tilde{\mathbf{v}}(\mathbf{x})\|_1 \le \sqrt{m} \sum_{i=1}^n \min \left\{ \|\mathbf{C}_{:,i}\| \|(\mathbf{a}_i^T, b_i)\|, \frac{l_{k(i)}}{N_{min}} + \|\mathbf{E}_i\|_F \right\} + \sum_{j=1}^m \sum_{i \in N_j} |c_{ji}| \|(\mathbf{a}_i^T, b_i)\|,$$

*where:*

- *$N_j$ is the set of null neurons with respect to output $j$.*
- *$k(i)$ is the cluster of neuron $i$.*
- *$N_{min}$ is the minimum cardinality of the non-null generators of any cluster.*
- *$l_k$ is the objective value of the optimization criterion for cluster $k$.*
- *$\epsilon_{j,i}$ is the difference/error between $c_{ji}(\mathbf{a}_i^T, b_i)$ and the cluster mean $\frac{\sum_{i \in I_{jk}} c_{ji}(\mathbf{a}_i^T, b_i)}{|I_{jk}|}$.*
- *$\mathbf{E}_i = [\epsilon_{1,i}, \ldots, \epsilon_{m,i}]$.*
- *$\| \cdot \|_F$ is the Frobenius norm.*

**Non-ReLU Activation.** Our method was developed for networks using the ReLU activation function. However, alternative activation functions exist and are frequently used in modern networks. GELU (Hendrycks & Gimpel, 2016) and SiLU (Elfwing et al., 2018) are smoothened variants of ReLU, while leaky ReLU (Maas et al., 2013) allows for non-zero gradients on negative values. To extend our method for

GELU and SiLU, notice that these behave similarly to ReLU with an additional small error: For GELU (SiLU is similar) we have $e(x) = e(-x) = |\text{GELU}(x) - \text{ReLU}(x)| = |x|(1 - \Phi(|x|))$, $\sup_{x \in \mathbb{R}} e(x) \approx 0.17$ and $\|\mathbf{v}_{\text{GELU}}(\mathbf{x}) - \mathbf{v}_{\text{ReLU}}(\mathbf{x})\|_1 \leq 0.17\|\text{vec}(C)\|_1$. By a triangle inequality, we obtain again the bound of Theorem 6 with an additional $0.17\|\text{vec}(C)\|_1$ error. For leaky ReLU, notice that $\text{LReLU}(x) = \max(x, \alpha x)$ and we obtain zonotopes with generators $[\alpha|c_{ji}|(\mathbf{a}_i^T, b_i), |c_{ji}|(\mathbf{a}_i^T, b_i)]$ instead of $[\mathbf{0}, |c_{ji}|(\mathbf{a}_i^T, b_i)]$. We claim that applying our method as if we had ReLU instead of leaky ReLU yields the same approximation bound. Indeed, consider the proof of Theorem 6. The leaky ReLU zonotopes are subsets of the ReLU zonotopes, and thus $\max_{\mathbf{v} \in V_{\tilde{P}_j}} \text{dist}(P_j, \mathbf{v})$ decreases. In addition, when bounding $\text{dist}(\mathbf{u}, \tilde{P}_j)$ for ReLU, the worst-case bound was obtained for $I'_{j+} = I_{j+}$. A vertex of a leaky ReLU zonotope corresponds to selecting for each $i \in I_{j+}$ either $\alpha|c_{ji}|(\mathbf{a}_i^T, b_i)$ or $|c_{ji}|(\mathbf{a}_i^T, b_i)$. Since $\alpha \in (0, 1)$, for $i$ of the first type we can scale $x_{ji}$ in the proof by $\alpha$ to bound their contribution in the leaky ReLU bound by $\alpha < 1$ times their contribution in the ReLU bound, meaning that the bound of $\text{dist}(\mathbf{u}, \tilde{P}_j)$ also decreases, and the final bound for ReLU holds also when the method is applied with leaky ReLU.

**Heuristic Improvements.** We obtain an advantage by dropping the bias from the clustering vectors – which leads to better upper hull approximations and prioritization of the generally more important regions extending to infinity – and normalizing the vectors used for clustering. These two heuristics are discussed in detail in the appendix.

**Connections to SVD.** Setting $\mathbf{M}_k = \sum_{i \in I_k} \mathbf{C}_{:,i}(\mathbf{a}_i^T, b_i)$, we can rewrite the optimization objective (1) and the weight update steps (2) (3) in matrix-form respectively as follows:

$$\text{minimize} \quad \sum_{j=1}^m \left\| \left( \widetilde{\mathbf{C}}_{:,k}(\tilde{\mathbf{a}}_k^T, \tilde{b}_k) - \sum_{i \in I_k} \mathbf{C}_{:,i}(\mathbf{a}_i^T, b_i) \right)_j \right\|^2 = \left\| \widetilde{\mathbf{C}}_{:,k}(\tilde{\mathbf{a}}_k^T, \tilde{b}_k) - \mathbf{M}_k \right\|_F^2,$$

$$\widetilde{\mathbf{C}}_{:,k} \leftarrow \frac{\mathbf{M}_k(\tilde{\mathbf{a}}_k, \tilde{b}_k)}{\left\| (\tilde{\mathbf{a}}_k^T, \tilde{b}_k) \right\|^2}, \quad (\tilde{\mathbf{a}}_k, \tilde{b}_k) \leftarrow \frac{\mathbf{M}_k^T \widetilde{\mathbf{C}}_{:,k}}{\left\| \widetilde{\mathbf{C}}_{:,k} \right\|^2}.$$

The objective is that of an optimal rank-1 approximation. Putting the update steps together (assuming convergence), we obtain

$$\widetilde{\mathbf{C}}_{:,k}^T \mathbf{M}_k(\tilde{\mathbf{a}}_\mathbf{k}, \tilde{b}) = \left\| (\tilde{\mathbf{a}}_k^T, \tilde{b}_k) \right\|^2 \|\widetilde{\mathbf{C}}_{:,k}\|^2.$$

This means that $\widetilde{\mathbf{M}}_k = \widetilde{\mathbf{C}}_{:,k}(\tilde{\mathbf{a}}_k^T, \tilde{b})$ is a rank-1 approximation of a singular value of $\mathbf{M}_k$. A good clustering (see Theorem 7) implies an initial solution better than the $\sigma_2(\mathbf{M}_k)$ rank-1 approximation. By monotonic improvement of alternating minimization, TropNNC converges to the best rank-1 approximation of $\mathbf{M}_k$.

In essence, TropNNC applies a power-iteration method to obtain the best rank-1 approximation of each $\mathbf{M}_k$. It relies on compact clustering and an additional assumption, in which case their addition is a good rank-$K$ approximation of the full non-linear operator $\mathbf{C} \cdot \text{ReLU} \cdot (\mathbf{A}, \mathbf{b})$. Hence, the whole procedure can be interpreted as an approximate SVD rank-$K$ reduction of $\mathbf{C} \cdot \text{ReLU} \cdot (\mathbf{A}, \mathbf{b})$. Applying SVD to $\mathbf{C}(\mathbf{A}, \mathbf{b})$ might lead to unfavorable cancellation in case of negative-positive alignment (see Theorem 7): with $\mathbf{C}_{:,i} = \mathbf{C}_{:,i'}$, $(\mathbf{a}_i^T, b_i) = -(\mathbf{a}_{i'}^T, b_{i'})$, neurons $i, i'$ give an "absolute value type" function. In $\mathbf{C}(\mathbf{A}, \mathbf{b})$, they cancel. In our method, we get 2 generators with Minkowski sum $[-|c_{ji}|(\mathbf{a}_i^T, b_i), |c_{ji}|(\mathbf{a}_i^T, b_i)] \neq \{\mathbf{0}\}$, i.e., no cancellation.

Even with no misalignment, SVD is not efficient as the dimensions become larger. Instead, we cluster neurons into compact clusters. Note the resemblance to spectral clustering, where to perform clustering on pairwise distances, SVD is used to convert them to an embedding space, whose rank/dimension is then reduced, followed by clustering. Here, we do the opposite: we are already given (via the network itself) the matrix $\mathbf{C}(\mathbf{A}, \mathbf{b})$ in a "sum of rank-1 matrices" form as $\sum_i \mathbf{C}_{:,i}(\mathbf{a}_i^T, b_i)$, and we avoid calculating the full SVD by first applying clustering. To the best of our knowledge, this problem is not extensively studied in the literature.

We position our method among other SVD-based compression methods, which in general rely on the following: They consider a single weight matrix $\mathbf{W} \in \mathbb{R}^{n \times d}$, and via a SVD rank-$r$ approximation, write $\widetilde{\mathbf{W}} = \mathbf{U}_r \mathbf{\Sigma}_r \mathbf{V}_r^T = (\mathbf{U}_r \mathbf{\Sigma}_r^{1/2})(\mathbf{\Sigma}_r^{1/2} \mathbf{V}_r^T)$, which splits the $d \to n$ transformation to a $d \to r \to n$ one, reducing

parameters and operations if $r \ll n$. However, this increases the depth of the network, and leaves the width of half the layers unchanged[1]. In contrast, our method handles the combined weights of two consecutive layers, i.e. $\mathbf{C}(\mathbf{A}, \mathbf{b})$. Of course, these have a non-linear ReLU activation between them. Tropical geometry, and in particular our refined Theorem 2, are what allow us to handle this and show with Theorem 6 a bound (in part) of the same order with that of SVD if ReLU was not present. To the best of our knowledge, we are the first to reduce consecutive matrices as one. Other works viewing matrices jointly, such as (Chen et al., 2023), still consider decompositions per single weight matrices, differentiating them from ours.

The following theorem analyzes the decomposition obtained via our clustering method compared to that of exact SVD. The proof's analysis, using perturbation theory, is more general than the statement itself.

**Theorem 7.** *Let* $\mathbf{c}_i = \mathbf{C}_{:,i}$, $\mathbf{M} = \sum_{i=1}^{n} \mathbf{c}_i(\mathbf{a}_i^T, b_i) = \mathbf{C}(\mathbf{A}, \mathbf{b})$, $\mathbf{M}_k = \sum_{i \in I_k} \mathbf{c}_i(\mathbf{a}_i^T, b_i)$. *Suppose that for each cluster the iterative algorithm reaches convergence to a solution* $(\tilde{\mathbf{a}}_k, \tilde{b}_k), \tilde{\mathbf{c}}_k$. *If the cluster centers are enough so that we have compact clusters and the network was trained with sufficient weight-decay (formal assumptions in the Appendix), then for* $\widetilde{\mathbf{M}} = \sum_k \tilde{\mathbf{c}}_k(\tilde{\mathbf{a}}_k^T, \tilde{b}_k) = \widetilde{\mathbf{C}}(\widetilde{\mathbf{A}}, \tilde{\mathbf{b}})$ *we have that*

$$\left\| \sum_k \tilde{\mathbf{c}}_k(\tilde{\mathbf{a}}_k^T, \tilde{b}_k) \right\|_2 \leq O(1) \|\mathbf{M}\|_2.$$

*In addition, there exists diagonal (rescaling) matrix* $\widetilde{\mathbf{D}}$ *such that*

$$\|\widetilde{\mathbf{C}}\widetilde{\mathbf{D}}^{-1}\|_2 \leq O(1)\|\mathbf{C}\|_2, \quad \|\widetilde{\mathbf{D}}(\widetilde{\mathbf{A}}, \tilde{\mathbf{b}})\|_2 \leq O(1)\|(\mathbf{A}, \mathbf{b})\|_2.$$

The assumptions are discussed in detail in the Appendix. Anti-parallel alignment leading to cancellation of the dominant singular direction and another becoming (uncontrollably) dominant, was problematic. In weight-decayed, ReLU networks, however, this is not a problem (also verified empirically). Because $a \cdot b < 0 \Rightarrow (a+b)^2 < a^2 + b^2$, weight-decay training does not allow for positive-negative alignment. Negative-positive alignment can lead to cancellation when ignoring ReLU, however taking ReLU into account such alignment creates "absolute value type" functions, which allow the maximum (in magnitude) of the two anti-parallel contributions to pass through. Although the last argument holds only for the spectral properties of the non-linear operator $\mathbf{C} \cdot \text{ReLU} \cdot (\mathbf{A}, \mathbf{b})$ (which is the one we care about), we assume WLOG that it also holds when ignoring ReLU to ease the exposition (i.e. all spectral norms are "modulo" a hidden ReLU). Also note that although weight-decay helps with alignment, because $a \cdot b > 0 \Rightarrow (a+b)^2 > a^2 + b^2$ it actually encourages redundancy.

As noticed in the Heuristics, similarity is determined by direction. The same was noticed in (Kim et al., 2020) with "direction, not absolute distance" determining similarity. The proof of Theorem 7 confirms this, but for a special "direction". Similar neurons should be considered those with big *rank-1* cosine similarity, i.e. $s(i, j) = \cos \angle(\mathbf{c}_i(\mathbf{a}_i^T, b_i), \mathbf{c}_j(\mathbf{a}_j^T, b_j)) = \frac{(\mathbf{c}_i^T \mathbf{c}_j)((\mathbf{a}_i, b_i)^T(\mathbf{a}_j, b_j))}{\|\mathbf{c}_i^T\|\|\mathbf{c}_j\|\|(\mathbf{a}_i, b_i)\|\|(\mathbf{a}_j, b_j)\|}$. This takes into account both input and output weights. If one does not consider rank-1 direction (e.g. if one takes only input weights into account), then this leads to worse results than simple $L^2$ distance. In the experiments, we simply used $L^2$.

**Deep and Convolutional Networks.** For deep networks, we can view each hidden layer of the network as an instance for our algorithm, having an input from the previous layer and providing an output to the next layer. Hence, we can recursively apply our multi-output algorithm to compress each hidden layer. This leads to the following approximation bound for the whole network:

**Theorem 8.** *Suppose we have a ReLU network with* $Q$ *layers, defined recursively as* $\mathbf{v}^{(q)}(\mathbf{x}) = \mathbf{A}^{(q)}\mathbf{x}^{(q-1)}, \mathbf{x}^{(q)} = ReLU(\mathbf{v}^{(q)} + \mathbf{b}^{(q)})$ *for all* $q \in [Q]$, *with input* $\mathbf{x}^{(0)} = \mathbf{x}$ *and output* $\mathbf{v} = \mathbf{v}^{(Q)}$. *Layer* $q$ *has* $n_q$ *neurons (i.e.* $\mathbf{A}^{(q)} \in \mathbb{R}^{n_q \times n_{q-1}}$), *and it is to be compressed to* $K_q$ *neurons. We compress the network layer-wise, from* $q = 1$ *to* $q = Q - 1$, *by applying iterative TropNNC to each hidden layer. We assume that throughout the execution, the assumptions of Theorem 7 hold. Suppose that for* $q_1 \leq q_2$,

---

[1]for layers with $\Theta(N)$ neurons, since we have $\Theta(Nr)$ parameters after SVD, $r$ has to be small. With $N \to \sqrt{Nr}, r \to \sqrt{Nr}$ instead, one gets the same number of parameters but with better balancing, i.e. less induced information bottleneck

$p_2(q_1, q_2) = \prod_{q=q_1}^{q_2} \|(\mathbf{A}^{(q)}, \mathbf{b}^{(q)})\|_2$ *and* $p_1(q_1, q_2) = \prod_{q=q_1}^{q_2} \|\mathbf{A}^{(q)}\|_1$. *Then, for the compressed network's output $\tilde{\mathbf{v}}$ it holds that*

$$\frac{1}{\rho} \max_{\mathbf{x} \in B} \|\mathbf{v}(\mathbf{x}) - \tilde{\mathbf{v}}(\mathbf{x})\|_1 \leq \sum_{q=1}^{Q-1} \left[ W(q) \left( \sqrt{n_{q+1}} \min \left\{ \sqrt{K_q} \|\mathbf{A}^{(q+1)}\|_F \|(\mathbf{A}^{(q)}, \mathbf{b}^{(q)})\|_F, \sum_{i=1}^{n_q} \frac{l_{k(i)}^{(q)}}{N_{min}} + \|\mathbf{E}_i^{(q)}\|_F \right\} \right. \right.$$
$$\left. \left. + \|(\mathbf{A}^{(q)}, \mathbf{b}^{(q)})\|_2 \sum_{j=1}^{n_{q+1}} \sum_{i \in N_j^{(q)}} |a_{ji}^{(q+1)}| \right) \right],$$

*where*

$$W(q) = p_1(q+2, Q) \left( p_2(1, q) + \sum_{q'=1}^{q} p_2(q'+1, q) \right) O(1)^q.$$

In the above bound, $W(q)$ is a form of weight/importance for layer $q$. The term $p_1(q+2, Q)$ captures how a signal scales as it goes from layer $q+1$ to the output. Any error induced by compressing layer $q$ will inevitably be scaled by the same factor, and hence this term cannot be improved. The term $\left( p_2(1, q) + \sum_{q'=1}^{q} p_2(q'+1, q) \right)$ captures how the domain of the signal values scales as the signal goes from the input to layer $q-1$. As the scale of the input to a layer proportionately influences the scale of the induced error, this term cannot be improved. The bound corresponds to a worst case that is quite unlikely because it assumes "worst-case alignment" for a lot of quantities that are in practice uncorrelated. For example, in the last step of the proof, the bound is tight only if i) there is a common input that achieves the worst-case error for every layer simultaneously, ii) for this input the signal through the network aligns closely with the dominant singular direction of every linear transformation, iii) the errors induced in every layer perfectly align. These conditions are very unlikely to hold in practice. The constant $O(1)$ is due to blow-ups in spectral norms, while the term $\sqrt{K_q}$ is due to blow up in Frobenius norm. For this to happen, we need adversarial alignment of the non-dominant singular directions of the linear transformations of the networks and of the clusters. We also note that the proof immediately gives us a way to develop a global sparsity allocation strategy: we weigh each layer $q$ based on $W(q)$. Notice the similarity of the first factor of $W(q)$ with the importance that is assigned in NISP (Yu et al., 2018). In the experiments, we did not implement such a global sparsity allocation strategy, and simply used uniform pruning (or non-uniform with hierarchical clustering (see below), which also does not capture information on $p_1$ and $p_2$). In the Appendix we empirically verify the soundness of the assumptions.

For convolutional networks, we unravel the kernels of the weight tensors row-wise for $\mathbf{A}$ and column-wise for $\mathbf{C}$, apply our algorithm, and reshape them back to 4D tensors. If we also have batch normalization, then this can be dealt with by fusing the operations of batch normalization into the preceding convolutional or linear layer, as batch normalization is itself a linear operation. Because its mean and variance corrections, but also its parameters, are the same for every neuron of a channel, fusing these into convolutional layers maintains them convolutional.

**Non-uniform Compression.** In Algorithm 2 we intentionally did not mention how clustering is performed. We can either use K-Means with a fixed per layer pruning ratio, or we can use hierarchical clustering with a global threshold parameter, like in the CUP framework. When used with hierarchical clustering, TropNNC differs from CUP in step 1, where we choose a different approach to build the clustering vectors (or filter features as Duggal et al. (2021) call them) of a convolutional layer, and more importantly in step 3, where we choose a different cluster representative based on tropical geometry. We also propose a modification to step 2 of CUP. Since the clustering vectors of different layers have varying dimensions, and vectors in higher-dimensional spaces tend to be more spread out, we introduce two variants for selecting the distance threshold for each layer:

- Variant 1: For each layer, take the distance threshold to be some global constant times the square root of the dimension of the clustering vectors of the layer.
- Variant 2: For each layer, take the distance threshold to be some global constant times the mean of the norms of the clustering vectors of the layer.

**Limitations.** Limitations are discussed in the appendix.

**Broader Discussion.** We used tropical geometry, and in particular Theorem 2, to cast the problem of a layer's compression as a simultaneous zonotope approximation problem, giving us a way to handle the non-linear intermediate ReLU-type activations and study network compression from a theoretical perspective. In particular, the ideas of the theoretical analysis can be summarized as follows:

- Theorem 2 refines the bound on tropical polynomial approximation. When working with zonotopes, it allows us to compare a vertex of one zonotope with its projection to the other, instead of another vertex. This allows us to extend the representative via a summation, and improve the error of "multiple term" vertices without worsening the error of "one term" vertices, which is what single-output TropNNC exploits.
- For multiple outputs, one has to approximate their zonotopes simultaneously. We approximate between the single-output representatives by choosing a common, "in-between" direction, and scaling it appropriately for each output. We further refine it via iterative minimization to minimize the error between the single-output representatives. Here, in each output the scale of the representative is matched with the scale of the summation of the cluster's generators after projection onto the common direction. This allows again the use of projections via Theorem 2 when deriving a bound.
- The iterative procedure connects with low-rank approximation of $\mathbf{C} \cdot \mathrm{ReLU} \cdot (\mathbf{A}, \mathbf{b})$. In Theorem 6 the terms $l_{k(i)}$ capture the quality of each cluster's SVD, while $\|\mathbf{E}_i\|_F$ capture the quality of the clustering. Together, they capture the total low-rank error. Tropical geometry handles the ReLU non-linearity.

Our theoretical analysis can also be used to gain insights about layer-wise network compression in plain (VGG-style) networks, which can be decomposed equivalently into finding a sparsity allocation among the layers, and compressing each layer so that the error of the whole network is minimized. In layer-wise compression, after determining sparsity allocation, one makes the approximation/decision to decompose the compression of the whole network into each layer sequentially. The general objective when compressing a plain layer can be expressed as:

$$\min_{(\widetilde{\mathbf{A}}, \tilde{\mathbf{b}}), \widetilde{\mathbf{C}}} \mathbb{E}_{\mathbf{x} \sim \mathcal{D}}[\|\tilde{f}_{out} \circ \widetilde{\mathbf{C}} \circ \mathrm{ReLU} \circ (\widetilde{\mathbf{A}}, \tilde{\mathbf{b}}) \circ \tilde{f}_{in}(\mathbf{x}) - f_{out} \circ \mathbf{C} \circ \mathrm{ReLU} \circ (\mathbf{A}, \mathbf{b}) \circ f_{in}(\mathbf{x})\|^2].$$

$\mathcal{D}$ is the input distribution. $f_{in}$ and $f_{out}$ are the network functions feeding in and out of the layer before any compression. $\tilde{f}_{in}$ and $\tilde{f}_{out}$ are the same functions right before the layer's compression. Usually, layers are compressed in increasing order from the input, meaning $\tilde{f}_{out} = f_{out}$. Also, methods usually minimize the incremental error, and approximate $f_{in} \leftarrow \tilde{f}_{in}$. In this case, the objective for each layer can be written as

$$\min_{(\widetilde{\mathbf{A}}, \tilde{\mathbf{b}}), \widetilde{\mathbf{C}}} \mathbb{E}_{\mathbf{x} \sim \tilde{f}_{in}(\mathcal{D})}[\|f_{out} \circ \widetilde{\mathbf{C}} \circ \mathrm{ReLU} \circ (\widetilde{\mathbf{A}}, \tilde{\mathbf{b}}) \circ \mathbf{x}_{in} - f_{out} \circ \mathbf{C} \circ \mathrm{ReLU} \circ (\mathbf{A}, \mathbf{b}) \circ \mathbf{x}_{in}\|^2],$$

where $\tilde{f}_{in}(\mathcal{D})$ denotes the push-forward distribution that the intermediate layer sees as input during compression. Thus, layer-wise compression can be considered as a form of low-rank approximation of the non-linear operator $\mathbf{C} \circ \mathrm{ReLU} \circ (\mathbf{A}, \mathbf{b})$ for inputs following $\tilde{f}_{in}(\mathcal{D})$ and outputs filtered through $f_{out}$. Then, an efficient layer-wise algorithm would aim to: i) take into account the distribution of the intermediate layer inputs, ii) take into account how the distribution of the error propagates to the output, iii) seek to achieve the best possible approximation of each non-linear layer with inputs following (i) and caring about outputs filtered through (ii), iv) weigh the importance of layers based on (i) and (ii). Ideally, after compression, v) the new feed-in function should not yield intermediate inputs that make subsequent layer compressions difficult.

- In Theorem 8, (i) is captured via $p_2$, (ii) via $p_1$, and (iv) via $W(q)$. (iii) is the aim of Theorem 6.
- Note that (v) partially falls outside layer-wise compression, since it requires foresight into subsequent compressions. Regardless, signal scales should be kept under control to avoid entering into high-error regions extending to infinity (see Heuristic 2, Appendix). Pruning methods, which can be viewed as applying a contractive projection, automatically satisfy this. For our method, we need Theorem 7 (for worst-case theoretical guarantees). Heuristic 2 also helps in practice.

Notice that pruning methods retaining $K$ neurons can be viewed implicitly as a special case of a clustering method using $K + 1$ clusters, placing all pruned neurons into the $(K + 1)$-st cluster and setting its representative weights to zero. Thus, we can interpret data-driven pruning methods through this lens.

- Data-driven importance-based methods typically focus on capturing the intermediate signal distributions $f_{in}(\mathcal{D})$ (i) and identifying and removing neurons with low individual contribution, aiming to fulfill condition (iii) indirectly via an accurate implicit clustering as the one above. On the other hand, Theorems 6 and 7 show that post-hoc neuron merging alone is insufficient; the clustering (explicit or implicit) must be compact, so that the merged rank-1 approximations $(|I_k|\bar{\mathbf{c}}_k)(\bar{\mathbf{a}}_k^T, \bar{b}_k)$ approximate each $\mathbf{M}_k$ well.
- In the data-free setting we have no distributional information. Thus, (i) and (ii) cannot be estimated beyond their scaling behavior captured by $p_2, p_1$ of $W(q)$. Consequently, only (iii) can be addressed in a principled manner under no/uninformative distributional assumptions. Methods based on low-rank approximations therefore focus on optimizing (iii).
- Empirically, data-driven methods tend to outperform data-free ones, suggesting that accurate modeling of intermediate signal distributions (i) plays a central role in achieving strong compression performance, even when objective (iii) is not optimized explicitly. This observation underscores the intrinsic difficulty of data-free pruning. Nevertheless, TropNNC remains competitive with data-driven approaches such as ThiNet, even under uniform sparsity allocation (i.e. the layer weight $W(q)$ of Theorem 8 is dropped).
- The above suggest that data-driven neuron similarity discovery could provide a principled direction for future methods addressing (i)–(iv) simultaneously. Note that with such method, our clustering power-iteration scheme may be useful for reasons beyond handling ReLU and efficiency.

Another possible research direction involves lifting the constraint of using zonotopes, or even polytopes, for the approximation. Note the relationship between the extended Newton polytope and Fenchel duality: For a tropical polynomial $p(\mathbf{x}) = \max_{i \in [n]} \left\{ \mathbf{a}_i^T \mathbf{x} + b_i \right\}$, the points $(\mathbf{a}^T, b)$ of the upper hull of $\mathrm{ENewt}(p)$ correspond exactly to all supporting hyperplanes $\mathbf{a}^T \mathbf{x} + b$ of $p(\mathbf{x}) \geq \mathbf{a}^T \mathbf{x} + b$. Equivalently, we have

$$\mathrm{upper}(\mathrm{ENewt}(p)) = \{(\mathbf{a}, b) \mid p^*(\mathbf{a}) = -b < \infty\} = \mathrm{upper}(\mathrm{hypo}(p^*)),$$

where $p^*(\mathbf{a}) = \sup_{\mathbf{x}}(\mathbf{a}^T \mathbf{x} - p(x))$ is the Fenchel conjugate of $p$. Conjugates are always concave, with convex hypographs, and their conjugation yields the original convex function. We may approximate the zonotopes with arbitrary convex sets, consider their upper hulls as the graphs of Fenchel conjugates, and obtain an approximate function after conjugation. Theorem 2 holds in this case as well. A natural choice is the Chebyshev sphere, or even inscribed ellipsoids, which correspond to hyperbolic-type functions. For example, consider two neurons $\max(\alpha x + 1, 0), \max(-\alpha x + 1, 0)$ with output $v = \max(2, \alpha x + 1, -\alpha x + 1)$, not reducible to a single ReLU neuron. If instead we approximate the rhomboid zonotope with its inscribed ellipsoid, then we get a hyperbolic approximation $1 + \sqrt{(1 + |\alpha x|)/2}$, which lower bounds the piecewise affine $v$, supporting it at its vertices. If the dimension is large, this is hit with the curse of dimensionality: the vertices of a unit square are at distance $\sqrt{n} - 1$ away from the Chebyshev sphere, and the approximation error in the middle of the affine pieces is large. Hence, theoretical research might find use in our work to reason about the fundamental hardness of network compression, since it is reduced to zonotopic approximation, a hard geometric problem regardless of the approximating "shapes"/types of functions used. Applied research may try to seek regularization techniques suppressing the complexity of the zonotopes during training.

## 6 Experiments

**Baselines.** We empirically evaluate TropNNC against baselines that perform structured pruning, without retraining. Specifically, we choose Neural Path K-Means by Misiakos et al. (2022), ThiNet, CUP, and the simple random and L1 structured pruning baselines. Neural Path K-Means is originally designed for linear layers only. To enable a fair comparison, we extended their approach using our proposed technique, making them applicable to convolutional layers as well. For the non-uniform variant of our framework, we compare it with CUP. As explained, our proposed algorithm enhances all three steps of CUP. In the presented experiments, we compare CUP exclusively with the fully enhanced version of TropNNC. To eliminate randomness, all random methods were performed 5 times on each model and the best performing compressed model was selected for testing. All metrics report the average over 5 repetitions of the same experiments with error bars indicating the standard deviation.

In terms of assumptions, ThiNet is data-driven and leverages access to data, while the remaining methods (including TropNNC) are strictly data-free. No retraining or fine-tuning is used for any method. Comparisons should be interpreted with this difference in mind.

Table 1: Comparison of Neural Path K-means and TropNNC on MNIST and Fashion-MNIST

| Percentage of Remaining Neurons | MNIST | | Fashion-MNIST | |
|---|---|---|---|---|
| | Neural Path K-means | TropNNC | Neural Path K-means | TropNNC |
| 100.0 | $98.54 \pm 0.16$ | $98.54 \pm 0.16$ | $89.16 \pm 0.21$ | $89.16 \pm 0.21$ |
| 50.0 | $97.85 \pm 0.39$ | $98.49 \pm 0.14$ | $88.17 \pm 0.46$ | $89.00 \pm 0.25$ |
| 25.0 | $96.69 \pm 1.06$ | $98.36 \pm 0.14$ | $86.33 \pm 0.87$ | $88.70 \pm 0.22$ |
| 10.0 | $96.25 \pm 1.39$ | $97.96 \pm 0.35$ | $84.91 \pm 1.28$ | $88.24 \pm 0.40$ |
| 5.0 | $95.17 \pm 2.36$ | $97.06 \pm 0.73$ | $81.48 \pm 3.90$ | $87.42 \pm 0.46$ |

**Datasets and networks.** We evaluate our framework on the MNIST (Deng, 2012) and CIFAR (Krizhevsky & Hinton, 2009) datasets, testing it across various models including simple multi-layer perceptrons (MLPs), convolutional neural networks (CNNs), AlexNet (Krizhevsky et al., 2012), and VGG (Simonyan & Zisserman, 2015). The non-uniform variant of our algorithm is tested on the CIFAR and ImageNet (Deng et al., 2009) datasets, with its performance evaluated across models such as VGG, and ResNet18 (He et al., 2016). The selected datasets and models are widely used benchmarks for neural network compression, and offer diversity in their complexity. We compare the methods based on test accuracies and FLOPs reduction, following the literature.

**Experimental setup.** One hyper-parameter needed tuning: ThiNet's training samples used. For each dataset, we gradually increased this number until no further improvement of ThiNet was observed. 5000 was found to suffice for both MNIST and CIFAR. The same number can be found by using the method presented in (Luo et al., 2017) for selecting the parameter. Computational overheads are discussed in the Appendix.

## 6.1   MNIST and Fashion-MNIST Datasets

The first experiment is performed on the MNIST and Fashion-MNIST datasets. Table 1 compares TropNNC with Neural Path K-means for the same CNN network and the same pruning ratios as in (Misiakos et al., 2022). We compress the final linear layer. As shown in the results, for both datasets, our algorithm outperforms Neural Path K-means.

To evaluate the performance of our algorithm in compressing linear layers of deeper networks, we applied TropNNC to "deepNN", a MLP with layer sizes $28 \times 28, 512, 256, 128$, and $10$. The performance plots are provided in Figures 4a and 4b. As illustrated, Neural Path K-Means achieves results comparable to L1-structured pruning. By contrast, TropNNC outperforms both Neural Path K-means and ThiNet.

To assess the performance of our algorithm in compressing convolutional layers, we applied TropNNC to "deepCNN2D", a LeNet-type convolutional neural network with ReLU activations. The performance plots are provided in Figures 4c and 4d. The results demonstrate that TropNNC significantly outperforms Neural Path K-means, whose effectiveness appears reduced. Moreover, TropNNC matches, or even surpasses, ThiNet.

## 6.2   CIFAR-10 and -100 Datasets

In this experiment, we compress AlexNet and VGG trained on the CIFAR-10 and CIFAR-100 datasets to assess the performance of each compression method. Figures 5a and 5b illustrate the compression of the linear layers of AlexNet on CIFAR-10 and CIFAR-100, respectively. Additionally, Figures 5c and 5d show the compression of VGG's convolutional layers for these datasets.

The results indicate that for larger datasets, TropNNC consistently outperforms Neural Path K-means, whose performance approaches that of random pruning. Furthermore, it matches or even surpasses the performance of ThiNet for the compression of linear layers. These findings highlight the effectiveness of TropNNC in handling more complex and larger-scale data scenarios. For the compression of convolutional layers of VGG, TropNNC matches ThiNet.

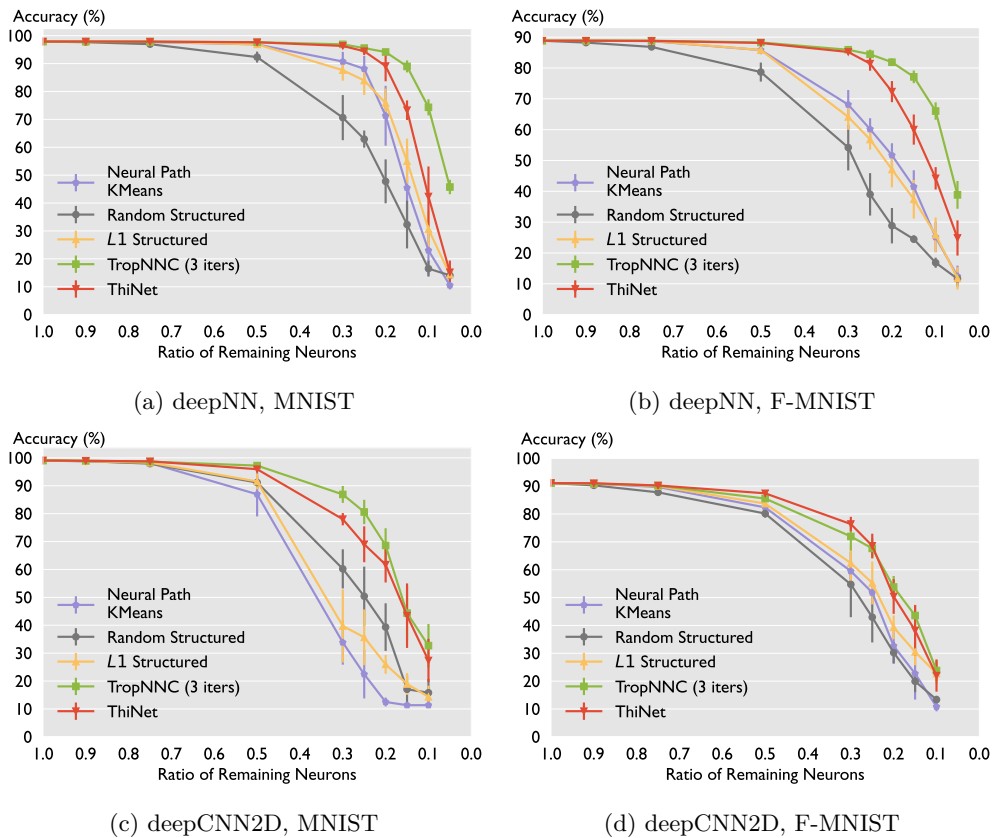

Figure 4: Compression of linear and convolutional layers of ReLU neural networks on MNIST datasets.

### 6.3 Non-uniform Pruning

We evaluate the effectiveness of the non-uniform variant of TropNNC. We compress various models, such as VGG on CIFAR-10, and ResNet18 on ImageNet. The results are summarized in Tables 2 and 3. We should emphasize that we did not apply any form of fine-tuning or re-training. Table 4 provides an ablation of the iterative variant of TropNNC.

Our findings indicate that our method demonstrates a clear advantage. Our approach shows a significant performance improvement for the VGG model, while the benefits are comparatively modest for ResNet18. Although this discrepancy remains somewhat unexplained we provide a plausible hypothesis (detailed in the Limitations, Appendix) explaining this phenomenon: The operator $\mathbf{I} + \mathbf{M}$ might deviate a lot from the approximation $\mathbf{I} + \widetilde{\mathbf{M}}$ (i.e., $\widetilde{\mathbf{I} + \mathbf{M}} \not\approx \mathbf{I} + \widetilde{\mathbf{M}}$).

We also found that the first variant of non-uniform TropNNC excelled at reducing the overall network size but was less effective at minimizing inference operations. In contrast, the second variant performed well in both tasks, outperforming CUP. Upon further analysis, we concluded that the first variant tends to focus more aggressively on the final layers, where parameter count is high due to the increased number of channels, but the number of operations is lower because of smaller image sizes. Meanwhile, the second variant, like CUP, also targets the initial layers, where fewer parameters are present, but a greater number of operations is required.

## 7 Conclusion

We proposed TropNNC, a tropical geometrical method for structured, data-free pruning of linear and convolutional layers of ReLU activated neural networks. Based on clustering similar neurons/channels, it selects

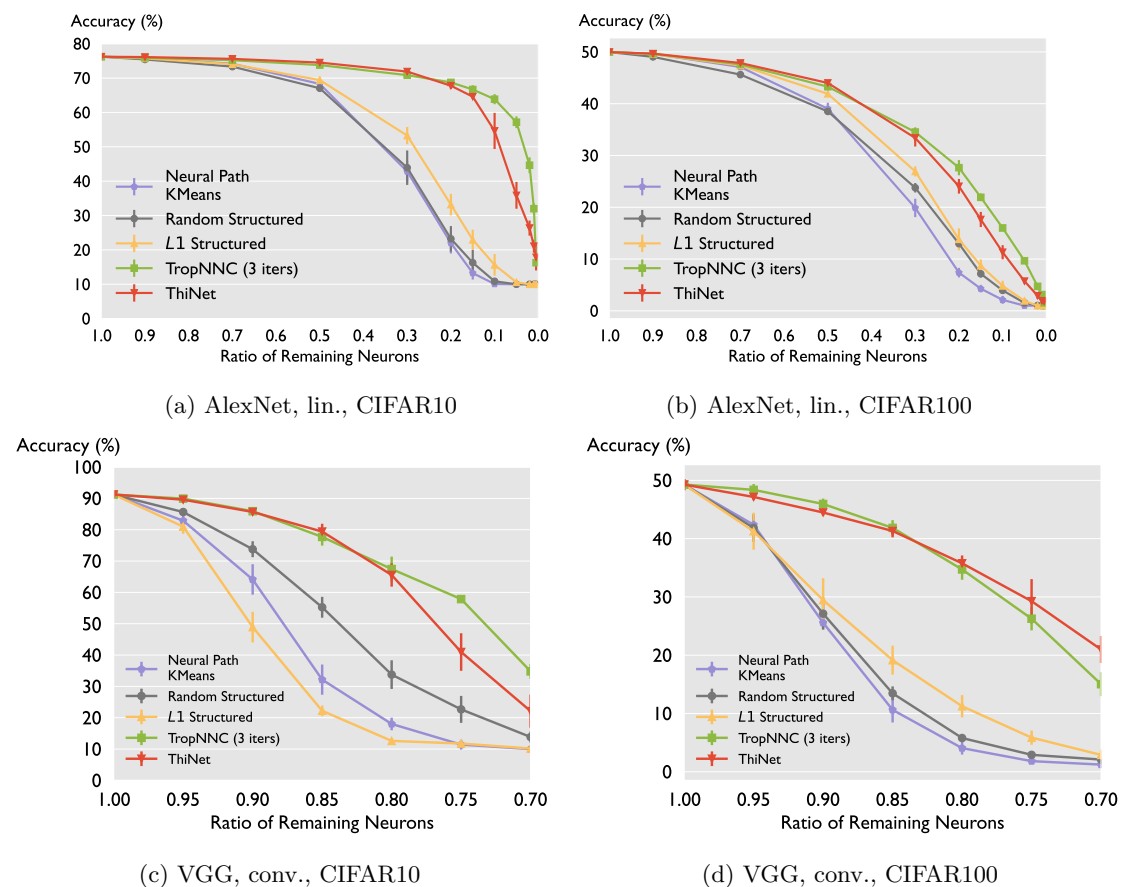

Figure 5: Compression of linear layers of AlexNet and convolutional layers of VGG on CIFAR datasets.

Table 2: Comparison of CUP and TropNNC (variant 1) accuracy across different pruning thresholds on CIFAR10, VGG.

| Method | Threshold | #params ↓ | FLOPS ↓ | Acc. ↑ |
|---|---|---|---|---|
| Original | - | 14.7M | 0.63G | 93.64 |
| CUP | 0.15 | 6.24M | 0.46G | 92.41 |
| TropNNC (3 iters) | 0.014 | 5.22M | 0.46G | 93.58 |
| CUP | 0.20 | 4.62M | 0.42G | 67.91 |
| TropNNC (3 iters) | 0.017 | 3.61M | 0.42G | 93.02 |
| CUP | 0.25 | 3.61M | 0.39G | 16.02 |
| TropNNC (3 iters) | 0.02 | 2.75M | 0.39G | 91.71 |

better cluster representatives than previous work by using tropical geometry and optimization, with provable compression guarantees. TropNNC significantly outperforms prior work on tropical geometric pruning and manages to match or even surpass the performance of the data-driven ThiNet. In non-uniform pruning it outperforms CUP, with significant improvement in the case of the VGG architecture. Our findings highlight the potential of tropical geometry in the realm of neural network compression.

Table 3: Comparison of CUP and TropNNC accuracy across different pruning thresholds on ImageNet, ResNet18.

| Method | Threshold | #params ↓ | FLOPS ↓ | Acc. ↑ |
|---|---|---|---|---|
| Original | - | 11.69M | 3.64G | 69.75 |
| CUP | 0.5 | 11.66M | 3.58G | 66.80 |
| TropNNC (v2) | 1.1 | 11.66M | 3.48G | 68.95 |
| CUP | 0.6 | 11.49M | 3.38G | 53.40 |
| TropNNC (v2) | 1.2 | 11.46M | 3.25G | 61.30 |
| CUP | 0.65 | 11.17M | 3.15G | 28.45 |
| TropNNC (v2) | 1.25 | 10.98M | 2.92G | 41.65 |
| TropNNC (v1) | 0.0187 | 9.84M | 3.46G | 59.15 |

Table 4: Comparison of non-uniform TropNNC and non-uniform iterative TropNNC (variant 1) accuracy on CIFAR10, VGG.

| Method | Threshold | #params ↓ | FLOPS ↓ | Acc. ↑ |
|---|---|---|---|---|
| Original | - | 14.7M | 0.63G | 93.64 |
| TropNNC | 0.02 | 2.75M | 0.39G | 86.2 |
| TropNNC (3 iters) | 0.021 | 2.52M | 0.38G | 91.2 |
| TropNNC | 0.025 | 1.80M | 0.33G | 34.89 |
| TropNNC (3 iters) | 0.0255 | 1.76M | 0.33G | 69.95 |

## Broader Impact Statement

This work does not introduce any new ethical concerns beyond those already present in prior research on network pruning and model compression. As with existing pruning methods, our approach aims to reduce model size and computational cost, and does not involve the use of sensitive data, human subjects, or deployment in high-risk applications.

## Acknowledgments

This project is funded by the European Union under Horizon Europe (grant No. 101136568 - HERON)

Funded by
the European Union
.

Early part of this research was also supported by the Hellenic Foundation for Research and Innovation (H.F.R.I.) under the "2nd Call for H.F.R.I. Research Projects to support Faculty Members & Researchers" (Project Number:2656, Acronym: TROGEMAL).

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

# A   Appendix

In this appendix, we first go into more detail regarding the heuristic improvements of our algorithm. Afterwards, we introduce the background of tropical algebra which was omitted from the main text. Finally, we provide the proofs for the theorems and propositions stated in the main text.

Before we proceed with the technical content of this appendix, we first provide a disclosure of the limitations of our method. We then also briefly discuss the computational overhead of the method.

**Limitations**

A limitation of our method is that it relies on the presence of consecutive convolutional or linear layers – that is, the existence of intermediate hidden layers. In addition, it only holds for networks with ReLU and other ReLU-type activations (as discussed in the relevant section in the main text). For other activations (e.g. tanh or sigmoid), the theory breaks. Finally, the method is not ideal for networks with residual connections, as detailed right below.

**Limitations of the method for networks with residual connections**

Here, we provide two hypotheses that may explain the reduced effectiveness of our method on residual architectures.

Suppose we have a layer with weights $(\mathbf{A}, \mathbf{b}, \mathbf{C})$, with matching number of inputs and outputs $d = m$ and $n$ hidden neurons, acting as a residual to a shortcut $\mathbf{x}$. This block can be written as

$$\mathbf{v}(\mathbf{x}) = \mathbf{C} \cdot \mathrm{ReLU}(\mathbf{A}\mathbf{x} + \mathbf{b}) + \mathbf{x} = ((\mathbf{I}, \mathbf{0}) + \mathbf{C} \cdot \mathrm{ReLU} \cdot (\mathbf{A}, \mathbf{b})) \begin{pmatrix} \mathbf{x} \\ 1 \end{pmatrix}.$$

Typically, a bias might then be added to $\mathbf{v}$, which is then passed through another ReLU and feed as input to the next residual block.

One principled way for our method to be applied in this case, is to take advantage of ReLU and write

$$x_j = \max(x_j, 0) + \min(x_j, 0) = \max(x_j, 0) - \max(-x_j, 0),$$

and fuse the residual path and the shortcut into one plain layer with $2d + n$ hidden neurons, with two new neurons per input with very sparse weights. This adds an additional generator to the positive and negative zonotope of each output $v_j$. In particular, the new zonotopes become $P'_j = P_j \oplus [\mathbf{0}, \mathbf{e}_j]$, $Q'_j = Q_j \oplus [\mathbf{0}, -\mathbf{e}_j]$. This, however, introduces a potential limitation: Initially, the shortcut $\mathbf{x}$ does not count toward the parameter count or inference cost. Equivalently, the $2d$ new neurons are so sparse that they are "given for free". After compression, however, their weights will change, and they will stop being sparse. This means that with the above scheme we have to remove at least $2d$ neurons before there is any advantage compared to the initial network (i.e., we need to remove $K + 2d$ neurons to have a $K$-compression of the original network). This suggests that, under this formulation, the method may be less effective for residual networks.

Alternatively (and as is commonly done), to compress the residual block, the main path $\mathbf{C} \cdot \mathrm{ReLU} \cdot (\mathbf{A}, \mathbf{b})$ is compressed, leaving the shortcut intact. If we ignore the ReLU non-linearity and view it via SVD, then our aim is to approximate the linear operator $(\mathbf{I} + \mathbf{C}(\mathbf{A}, \mathbf{b}))$ as $(\mathbf{I} + \widetilde{\mathbf{C}}(\widetilde{\mathbf{A}}, \tilde{\mathbf{b}}))$. (In the data-free setting) we want

$$\min_{\widetilde{\mathbf{C}}, (\widetilde{\mathbf{A}}, \tilde{\mathbf{b}})} \|(\mathbf{I} + \mathbf{C}(\mathbf{A}, \mathbf{b})) - (\mathbf{I} + \widetilde{\mathbf{C}}(\widetilde{\mathbf{A}}, \tilde{\mathbf{b}}))\|_F^2 = \min_{\widetilde{\mathbf{C}}, (\widetilde{\mathbf{A}}, \tilde{\mathbf{b}})} \|\mathbf{C}(\mathbf{A}, \mathbf{b}) - \widetilde{\mathbf{C}}(\widetilde{\mathbf{A}}, \tilde{\mathbf{b}})\|_F^2,$$

and $\widetilde{\mathbf{M}} = \widetilde{\mathbf{C}}(\widetilde{\mathbf{A}}, \tilde{\mathbf{b}})$ is the SVD low-rank approximation of $\mathbf{M} = \mathbf{C}(\mathbf{A}, \mathbf{b})$. However, after adding $\mathbf{I}$, this may lead to a suboptimal approximation of $\mathbf{I} + \mathbf{C}(\mathbf{A}, \mathbf{b})$ due to shift in spectral values. Take, for instance, $\mathbf{M} = \begin{pmatrix} 1/4 & 0 \\ 0 & -3/4 \end{pmatrix}$. $\mathbf{M}$ is full-rank, diagonalizable with eigenvalues $1/4, -3/4$. Its low-rank approximation

is $\widetilde{\mathbf{M}} = \begin{pmatrix} 0 & 0 \\ 0 & -3/4 \end{pmatrix}$. Then, we have

$$\mathbf{I} + \mathbf{M} = \begin{pmatrix} 5/4 & 0 \\ 0 & -1/4 \end{pmatrix}, \quad \widetilde{\mathbf{I} + \mathbf{M}} = \begin{pmatrix} 5/4 & 0 \\ 0 & 0 \end{pmatrix}, \quad \mathbf{I} + \widetilde{\mathbf{M}} = \begin{pmatrix} 1 & 0 \\ 0 & 1/4 \end{pmatrix}$$

and the approximation error of $\mathbf{I} + \widetilde{\mathbf{M}}$ is 3 times as large as that of $\widetilde{\mathbf{I} + \mathbf{M}}$. In the above, the redundant eigenvalue was the one closest to $-1$, not the one with the smallest magnitude. Hence, low-rank approximation of $\mathbf{M}$ got rid of the wrong eigenvalue.

The above situation, where there is a mismatch between the "small magnitude" redundant and the "close to $-1$" redundant eigenvalue, is not against theoretical intuition. Consider the ODE viewpoint of a simple residual block as a discretized first-order ODE step (Chen et al., 2018):

$$\mathbf{v}(\mathbf{x}) = \mathbf{x} + hF(\mathbf{x}) \longrightarrow \dot{\mathbf{x}} = F(\mathbf{x}).$$

For this ODE to be stable, the eigenvalues of the Jacobian of $F$ (which corresponds to $\mathbf{C}(\mathbf{A}, \mathbf{b})/h$) must have negative real parts. Then, it is reasonable to expect in the residual network the eigenvalues of $\mathbf{C}(\mathbf{A}, \mathbf{b})$ to be negatively biased. In addition, the residual path makes corrections to the input, and we do not reasonably expect its eigenvalues to be too large in magnitude (e.g., $> 1$), since this would vanish or explode the signal. Then, if the eigenvalues are symmetrically distributed, the above two conditions may create a discrepancy between the $\widetilde{\mathbf{M}}$-redundant and the truly redundant eigenvalues.

The two observations suggest that the reduced gains on residual architectures may be due to structural properties of residual connections requiring alternative handling, rather than a limitation of our method itself. We emphasize that these are hypotheses supported by theoretical considerations; a systematic empirical validation is an important direction for future work.

**Computational overhead**

The iterative update steps can be implemented efficiently using tensor operations in their matrix-form, as described in the "Connections to SVD" subsection of the main text. This reduces implementation overhead and results in efficient updates. In practice, the dominant cost arises from the clustering step. Empirically, we find that the overall compression procedure is efficient, comparable to our baselines, and does not introduce prohibitive overhead. While we do not include a systematic runtime benchmark, for networks at the scale considered in our experiments, the runtime of the method is practical.

### A.1 Heuristic Improvements

In the experiments comparing TropNNC and CUP, TropNNC was enhanced with 2 heuristic improvements. These improvements are not rigorously backed by theory; however, we try to provide intuitive explanations as to why they work.

**Heuristic 1.** First, let us focus on the single-output case, where the problem of network compression reduces to the problem of zonotope approximation. In this setting, Algorithm 1 suggests clustering similar generators based on their $L^2$ distances and replacing each cluster with the sum of its elements. The intuition is as follows: if the generators within a cluster are exactly equal, then through their Minkowski sum, these generators become equivalent to a single generator given by their sum. We can take this one step further: if we have generators that are parallel to each other, their Minkowksi sum reduces to a single equivalent generator without introducing any error. This suggest that generator similarity is determined by direction (i.e. cosine similarity) and not $L^2$ distance. Hence, an improvement to the single-output algorithm is to cluster the generators based on cosine similarity. In practice, this can be achieved by applying K-Means clustering to normalized generators, while still using the original generators in the summation step.

A similar heuristic can also be applied to the multi-output algorithm. By normalizing the input weights of the clustering vectors (i.e. the vectors used for clustering similar neurons/channels, not the weights used for calculating the representative of each cluster) we obtain an improvement to the algorithm.

**Heuristic 2.** The second heuristic is motivated by Theorem A.3, which states that two tropical polynomials are functionally identical if the upper hulls of their extended Newton polytopes coincide (i.e. only the vertices of the upper hull contribute to non-redundant terms). Indeed, this theorem also serves as motivation for Theorem 2, which shows that approximate equality of the extended Newton polytopes implies approximate functional equivalence of the corresponding tropical polynomials. Although we do not formally prove it, this theorem can be further refined by bounding the error of two tropical polynomials by the Hausdorff distance of the upper hulls of their extended Newton polytopes:

$$\frac{1}{\rho} \max_{\mathbf{x} \in B} |p(\mathbf{x}) - \tilde{p}(\mathbf{x})| \leq H(UF(P), UF(\tilde{P}))$$

Based on this refined result, it becomes clear that to compress networks, it suffices to approximate the upper hulls of the zonotopes of the network, rather than the zonotopes themselves.

We can take this idea one step further. Consider the upper hull of an extended Newton polytope, and project it by eliminating the bias term. Then, the points on the convex hull of the resulting projection (which coincides with the convex hull of the (non-extended) Newton polytope) correspond to the slopes of the outer linear regions of the polynomials, i.e. the regions which extend to infinity. We can view these points as being the "outer" points of the upper hull. When approximating a tropical polynomial, these unbounded regions are particularly important, because they have the potential to make the sup-norm of the approximation error infinite (something which is not captured by Theorem 2 because we divide by $\rho$ and take the $L^1$-norm error on a bounded set).

Based on this insight, our goal is to design an algorithm that prioritizes the approximation of the upper hulls of the zonotopes – particularly the vertices corresponding to the unbounded regions of the tropical polynomials. (i.e. the "outer" vertices of the upper hulls).

Let us first see how vanilla TropNNC behaves on a simple example. Consider the tropical polynomial $v(x) = \max(0, -x + 5) + \max(0, x + 5) + \max(0, x)$. Such a polynomial and its corresponding zonotope are depicted in Figure 6a. Notice that the upper hull has 4 points, each corresponding to a linear region of the polynomial. It also has 2 "outer" points, which correspond to the 2 outer regions of the polynomial extending to infinity.

We reduce the terms from 3 to $K = 2$. First, TropNNC performs clustering on the generators of the zonotope, forming the clusters $\{(-1, 5), (1, 5)\}$ and $\{(1, 0)\}$. It then replaces the first cluster with the representative $(0, 10)$ and the second cluster with the representative $(1, 0)$. The reduced tropical polynomial is $\tilde{v}(x) = \max(0, 10) + \max(0, x) = 10 + \max(0, x)$. The compression procedure and the corresponding reduced function is depicted in Figure 6b.

It is evident that this approximation is not ideal: while the reduced polynomial accurately captures the two central regions, it fails to preserve the unbounded regions. This causes the approximation error becomes unbounded.

Since we have $K = 2$, the best possible approximation using only 2 terms is $v'(x) = \max(0, -x + 5) + \max(0, 2x+5)$. This approximation captures both the important outer regions with zero error, and also one of the central regions. The question however remains: how could we have arrived at such an approximation? To answer this, notice that $v'(x) = \max(0, -x+5)+\max(0, (x+5)+(x))$. Thus, if TropNNC had initially formed the clusters $\{(-1, 5)\}$ and $\{(1, 5), (1, 0)\}$, then it would have been able to find the better approximation. Now, the modification we have to make becomes clear: instead of forming the clusters based on the generators themselves, we can form the clusters based on the reduced generators with the bias suppressed (and, of course, for the summation step the bias remains). The modified compression procedure is depicted in Figure 6c. The new approximation clearly provides a closer fit to the upper hull and captures the "outer" points exactly, resulting in zero error in the outer unbounded regions.

To conclude, another heuristic improvement comes from dropping the bias from the clustering vectors, seeing as this helps the algorithm achieve better upper hull approximations and prioritize the important "outer" vertices of the upper hull.

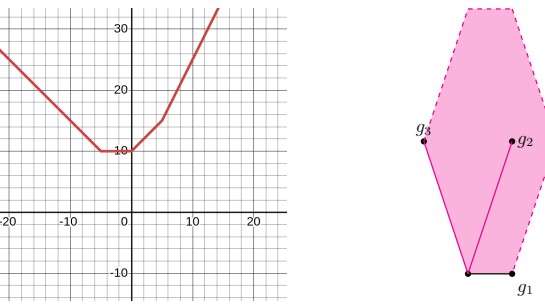

(a) Example tropical polynomial $v(x) = \max(0, -x + 5) + \max(0, x + 5) + \max(0, x)$ and corresponding zonotope.

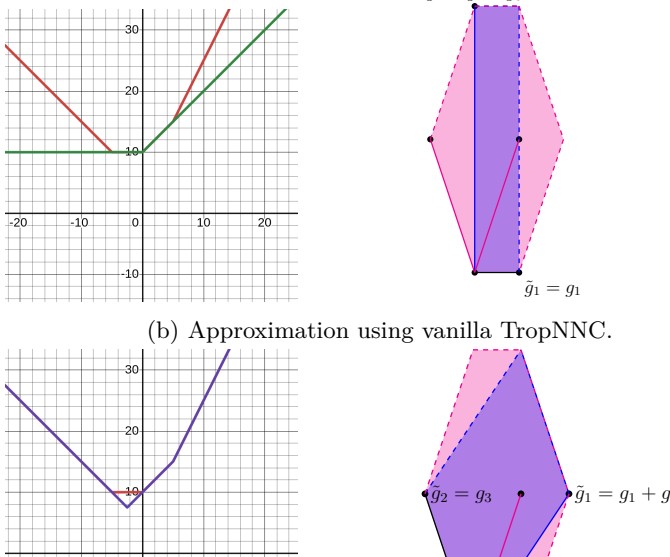

(b) Approximation using vanilla TropNNC.

(c) Approximation using Heuristic 2.

Figure 6: Example of reduction of tropical polynomial.

## A.2 Background on Tropical Algebra

We begin with the definitions of tropical polynomials, tropical rational functions, Newton polytopes, and zonotopes.

**Tropical Polynomials and Rational Functions.** Within the max-plus semiring, we can define polynomials. A *tropical polynomial* $f$ in $d$ variables $\mathbf{x} = (x_1, \ldots, x_d)$ is defined as the function:

$$f(\mathbf{x}) = \bigvee_{i=1}^{n} \left\{ \mathbf{a}_i^T \mathbf{x} + b_i \right\} = \max_{i \in [n]} \left\{ \mathbf{a}_i^T \mathbf{x} + b_i \right\}, \tag{4}$$

where $[n] := \{1, ..., n\}$. Here, $n$ represents the *rank* of the tropical polynomial. Each monomial term $\left\{ \mathbf{a}_i^T \mathbf{x} + b_i \right\}$ of the polynomial has an *exponent* or *slope* $\mathbf{a}_i \in \mathbb{R}^d$ and a *coefficient* or *bias* $b_i \in \mathbb{R}$. Each monomial term corresponds to a plane in $\mathbb{R}^{d+1}$. Consequently, tropical polynomials are piecewise linear convex functions. Specifically, every tropical polynomial is a continuous piecewise linear convex function, and every continuous piecewise linear convex function can be expressed (though not uniquely) as a tropical polynomial (Maclagan & Sturmfels, 2021). The set of tropical polynomials in $\mathbf{x}$ defines the semiring $\mathbb{R}_{\max}[\mathbf{x}]$. Figures 7a, 7b illustrate examples of tropical polynomials in one and multiple variables, respectively.

Tropical rational functions are defined as the difference of two tropical polynomials $p$ and $q$:

$$r(\mathbf{x}) = p(\mathbf{x}) - q(\mathbf{x})$$

Tropical rational functions correspond to general piecewise linear functions. Specifically, every tropical rational function is a continuous piecewise linear function, and every continuous piecewise linear function can be expressed (though not uniquely) as a tropical rational function. Figures 7c, 7d provide examples of tropical rational functions.

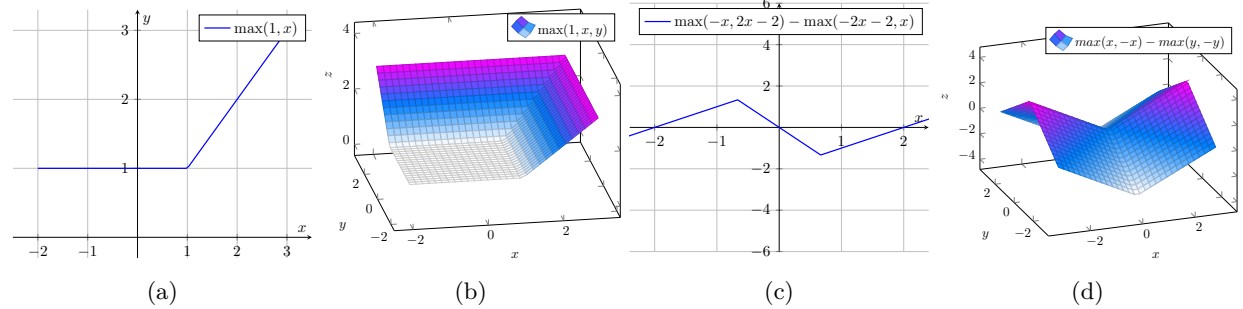

Figure 7: (a) depicts a single-variate tropical polynomial, (b) depicts a multi-variate tropical polynomial, (c) depicts a single-variate tropical rational function, (d) depicts a multi-variate tropical rational function

**Newton Polytopes.** For a tropical polynomial $f$ as defined in (4), we define its *Newton polytope* as the convex hull of the slopes $\mathbf{a}_i$ of $f$

$$\mathrm{Newt}(f) := \mathrm{conv}\left\{\mathbf{a}_i, i \in [n]\right\}.$$

Additionally, we define the *extended Newton polytope* of a tropical polynomial $f$ as the convex hull of the slopes $\mathbf{a}_i$ of $f$ extended in their last dimension by the coefficient $b_i$

$$\mathrm{ENewt}(f) := \mathrm{conv}\left\{(\mathbf{a}_i^T, b_i), i \in [n]\right\}.$$

The following proposition allows us to calculate the (extended) Newton polytope of expressions of tropical polynomials, and it was used in the main text to deduce that the extended Newton polytopes of the outputs of the network are zonotopes.

**Proposition A.1** (Zhang et al., 2018). *Let $f, g \in \mathbb{R}_{\max}[\mathbf{x}]$ be two tropical polynomials in $\mathbf{x}$. For the extended Newton polytope, the following holds:*

$$\mathrm{ENewt}\left(f \vee g\right) = \mathrm{conv}\left\{\mathrm{ENewt}(f) \cup \mathrm{ENewt}(g)\right\}$$
$$\mathrm{ENewt}(f + g) = \mathrm{ENewt}(f) \oplus \mathrm{ENewt}(g).$$

The Minkowski sum $\oplus$ of two polytopes (or more generally subsets of $\mathbb{R}^d$) $P$ and $Q$ is defined as:

$$P \oplus Q := \{p + q \mid p \in P, q \in Q\}.$$

Using the principle of induction, Proposition A.1 can be generalized to any finite tropical expression of tropical polynomials.

The *upper envelope* or *upper hull* $UF(P)$ of an extended Newton polytope is defined as the set of all points $(\mathbf{a}^T, b)$ of the polytope $P$ that are not "shadowed" by any other part of the polytope when viewed from above (last dimension). This means that there is no $b' > b$ such that $(\mathbf{a}^T, b')$ belongs to $P$. We have the following useful lemma.

**Lemma A.2.** *Let $p \in \mathbb{R}_{\max}[\mathbf{x}]$ a tropical polynomial in $d$ variables with extended Newton polytope $P = \mathrm{ENewt}(p)$. If $(\mathbf{a}^T, b)$ lies below the upper envelope of $P$, then $\forall \mathbf{x} \in \mathbb{R}^d, \mathbf{a}^T\mathbf{x} + b \leq p(\mathbf{x})$. The inequality is strict if $(\mathbf{a}^T, b)$ lies strictly below the upper envelope.*

*Proof.* Since $(\mathbf{a}^T, b)$ lies below the upper envelope of $P$, there exists a point

$$(\mathbf{a}^T, b') = \sum_{i=1}^{k} \lambda_i \mathbf{v}_i, \quad \sum_{i=1}^{k} \lambda_i = 1$$

on a face of the upper envelope of $P$ defined by the points $\mathbf{v}_1, \ldots, \mathbf{v}_k \in V_{UF(P)}$ such that $b' \geq b$. Therefore, we have:

$$\mathbf{a}^T \mathbf{x} + b \leq \mathbf{a}^T \mathbf{x} + b' = \langle (\mathbf{a}^T, b'), (\mathbf{x}, 1) \rangle$$

$$= \sum_{i=1}^{k} \lambda_i \langle \mathbf{v}_i, (\mathbf{x}, 1) \rangle \leq \max_{i=1,\ldots,k} \langle \mathbf{v}_i, (\mathbf{x}, 1) \rangle \leq p(\mathbf{x}).$$

If $(\mathbf{a}_i^T, b_i)$ lies strictly below the upper envelope of $P$, then $b' > b$, and the inequality is strict. $\qquad \square$

The extended Newton polytope provides a geometrical interpretation for studying tropical polynomials. For instance, the following theorem holds.

**Theorem A.3.** *For any two tropical polynomials $f, g \in \mathbb{R}_{\max}[\mathbf{x}]$, the following holds:*

$$f = g \Leftrightarrow UF(\text{ENewt}(f)) = UF(\text{ENewt}(g))$$

*This implies that two tropical polynomials are functionally identical if and only if their extended Newton polytopes have the same upper envelope.*

The above theorems indicate that a tropical polynomial is fully functionally determined by the upper envelope of its extended Newton polytope, as shown by the following example.

**Example 2.** *Consider the polynomials corresponding to Figure 8.*

$$f(x, y) = \max\{0, -y + 1, y + 1\}$$

$$g(x, y) = \max\{x + 1, -x + 1\}.$$

*We have that*

$$(f \vee g)(x, y) = \max\{0, x + 1, y + 1, -y + 1, -x + 1\}$$

$$(f + g)(x, y) = \max\{x + 1, -x + 1, x - y + 2,$$

$$-x - y + 2, x + y + 2, -x + y + 2\}.$$

*The extended Newton polytopes of $f, g, f \vee g, f + g$ are shown in Figure 8. The polynomial $f \vee g$ can be reduced as follows:*

$$(f \vee g)(x, y) = \max\{x + 1, y + 1, -y + 1, -x + 1\},$$

*which corresponds to the vertices of the upper envelope of $\text{ENewt}(f \vee g)$.*

**Zonotopes.** In polytope theory, *zonotopes* are a special class of convex polytopes that can be defined as the Minkowski sum of a finite set of line segments (or edges). Formally, given a set of line segments $g_1, \ldots, g_n$, the zonotope is defined as

$$Z := \bigoplus_{i \in [n]} g_i$$

The line segments $g_i$ are referred to as the zonotope's *generators*.

Alternatively, a zonotope can be expressed equivalently by a set of vectors $\mathbf{v}_1, \ldots, \mathbf{v}_n \in \mathbb{R}^d$ and a starting point $\mathbf{s} \in \mathbb{R}^d$. By taking the generators to be the segments $[\mathbf{0}, \mathbf{v}_1], \ldots, [\mathbf{0}, \mathbf{v}_n]$ and translating the first segment by $\mathbf{s}$, we obtain the equivalent form:

$$Z = \left\{ \mathbf{s} + \sum_{i=1}^{n} \lambda_i \mathbf{v}_i \mid 0 \leq \lambda_i \leq 1 \right\}$$

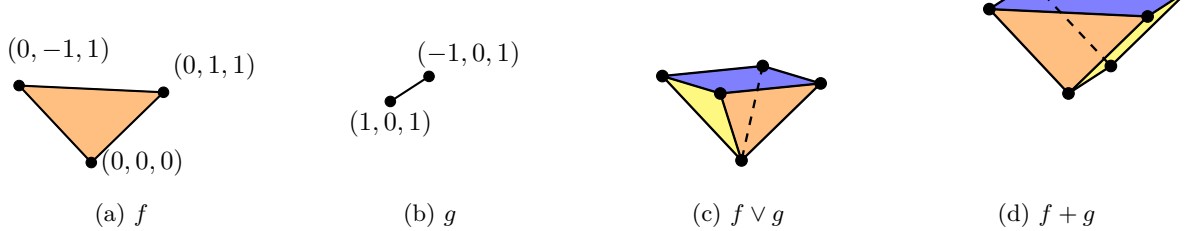

$(0, -1, 1)$   $(0, 1, 1)$   $(-1, 0, 1)$
$(1, 0, 1)$
$(0, 0, 0)$

(a) $f$       (b) $g$       (c) $f \vee g$       (d) $f + g$

Figure 8: Operations on tropical polynomials. $\mathrm{ENewt}(f \vee g)$ corresponds to the convex hull of the union of the vertices of the polytopes $\mathrm{ENewt}(f), \mathrm{ENewt}(g)$. $\mathrm{ENewt}(f + g)$ corresponds to the Minkowski sum of $\mathrm{ENewt}(f), \mathrm{ENewt}(g)$. For the polytope $\mathrm{ENewt}(f \vee g)$ we illustrate with blue the upper envelope, which consists of a single face. The vertices of the upper envelope are the only non reduntant terms of the polynomial $f \vee g$.

In this context, the vectors $\mathbf{v}_i$ are sometimes referred to as the zonotope's generators, meaning the segments $[\mathbf{0}, \mathbf{v}_i]$. When the starting point $\mathbf{s}$ is not mentioned, it is assumed to be the origin $\mathbf{0}$.

For a zonotope with a starting point $\mathbf{s} \in \mathbb{R}^d$ and generators $\mathbf{v}_1, \ldots, \mathbf{v}_n \in \mathbb{R}^d$, a vertex $\mathbf{u} \in V_Z$ corresponds to points where $\lambda_i = 0$ or $1$. The vertex $\mathbf{u}$ can be expressed as:

$$\mathbf{u} = \mathbf{s} + \sum_{i \in I} \mathbf{v}_i,$$

where $I \subseteq [n]$.

### A.3 Tropical polynomial approximation based on Hausdorff distance

Before we proceed with the proof of our refined Theorem 2, we first provide an auxiliary lemma, stated in the main text, which is necessary to see how our bound is indeed tighter than the bound of Misiakos et al. (2022). This lemma is also used for the proofs of Theorem 2, Theorem 6, and Proposition 6.

**Lemma A.4.** *Due to the convexity and compactness of polytopes, we have that*

$$H(P, \tilde{P}) = \max \left\{ \max_{\mathbf{u} \in V_P} \mathrm{dist}(\mathbf{u}, \tilde{P}), \max_{\mathbf{v} \in V_{\tilde{P}}} \mathrm{dist}(P, \mathbf{v}) \right\}$$

*Proof.* We will prove that

$$\sup_{\mathbf{u} \in P} \mathrm{dist}(\mathbf{u}, \tilde{P}) = \max_{\mathbf{u} \in V_P} \mathrm{dist}(\mathbf{u}, \tilde{P}),$$

i.e., the supremum is attained at some vertex of $P$.

The polytope $P$, the domain of the supremum, is convex and compact. Thus, it suffices to prove that the function

$$f(\mathbf{u}) = \mathrm{dist}(\mathbf{u}, \tilde{P}) = \inf_{\tilde{\mathbf{u}} \in \tilde{P}} \|\mathbf{u} - \tilde{\mathbf{u}}\|$$

is convex in terms of $\mathbf{u}$.

Let $\mathbf{u}_1, \mathbf{u}_2 \in P$. By the compactness of $\tilde{P}$, there exist points $\tilde{\mathbf{u}}_1, \tilde{\mathbf{u}}_2 \in \tilde{P}$ such that $f(\mathbf{u}_1) = \mathrm{dist}(\mathbf{u}_1, \tilde{P}) = \|\mathbf{u}_1 - \tilde{\mathbf{u}}_1\|$ and $f(\mathbf{u}_2) = \mathrm{dist}(\mathbf{u}_2, \tilde{P}) = \|\mathbf{u}_2 - \tilde{\mathbf{u}}_2\|$.

For every $\lambda \in [0, 1]$, we have that

$$\lambda f(\mathbf{u}_1) + (1 - \lambda) f(\mathbf{u}_2) = \lambda \|\mathbf{u}_1 - \tilde{\mathbf{u}}_1\| + (1 - \lambda)\|\mathbf{u}_2 - \tilde{\mathbf{u}}_2\|$$
$$\geq \|\lambda(\mathbf{u}_1 - \tilde{\mathbf{u}}_1) + (1 - \lambda)(\mathbf{u}_2 - \tilde{\mathbf{u}}_2)\|$$
$$= \|\lambda \mathbf{u}_1 + (1 - \lambda)\mathbf{u}_2 - \lambda \tilde{\mathbf{u}}_1 - (1 - \lambda)\tilde{\mathbf{u}}_2\|.$$

By the convexity of $\tilde{P}$, $\tilde{\mathbf{u}} = \lambda\tilde{\mathbf{u}}_1 + (1-\lambda)\tilde{\mathbf{u}}_2 \in \tilde{P}$. Hence,

$$\lambda f(\mathbf{u}_1) + (1-\lambda)f(\mathbf{u}_2) \geq \|\lambda\mathbf{u}_1 + (1-\lambda)\mathbf{u}_2 - \tilde{\mathbf{u}}\| \geq f(\lambda\mathbf{u}_1 + (1-\lambda)\mathbf{u}_2),$$

which concludes the proof. □

We continue with the proof of Theorem 2.

*Proof (Theorem 2).* Consider a point $\mathbf{x} \in B$ and assume that $p(\mathbf{x}) = \mathbf{a}^T\mathbf{x} + b$ and $\tilde{p}(\mathbf{x}) = \mathbf{c}^T\mathbf{x} + d$. Take an arbitrary $(\mathbf{u}^T, v) \in \tilde{P}$. This point lies below the upper envelope of $\tilde{P}$. Thus, by Lemma A.2, we have that $\tilde{p}(\mathbf{x}) \geq \mathbf{u}^T\mathbf{x} + v$. Choose $(\mathbf{u}^T, v)$ to be the closest point to $(\mathbf{a}^T, b)$. Then,

$$
\begin{aligned}
p(\mathbf{x}) - \tilde{p}(\mathbf{x}) &\leq p(\mathbf{x}) - (\mathbf{u}^T, v)\begin{pmatrix}\mathbf{x}\\1\end{pmatrix} \\
&= ((\mathbf{a}^T, b) - (\mathbf{u}^T, v))\begin{pmatrix}\mathbf{x}\\1\end{pmatrix} \\
&\leq \left\|(\mathbf{a}^T, b) - (\mathbf{u}^T, v)\right\|\left\|\begin{pmatrix}\mathbf{x}\\1\end{pmatrix}\right\| \\
&\leq \text{dist}((\mathbf{a}^T, b), \tilde{P}) \cdot \rho \\
&\leq \max_{(\mathbf{a}^T, b) \in V_P} \text{dist}((\mathbf{a}^T, b), \tilde{P}) \cdot \rho,
\end{aligned}
$$

where the second inequality is due to the Cauchy-Schwarz inequality.

In a similar manner, take an arbitrary $(\mathbf{r}^T, s) \in P$. This point lies below the upper envelope of $P$. Thus, by Lemma A.2, we have that $p(\mathbf{x}) \geq \mathbf{r}^T\mathbf{x} + s$. Choose $(\mathbf{r}^T, s)$ to be the closest point to $(\mathbf{c}^T, d)$. Then,

$$
\begin{aligned}
p(\mathbf{x}) - \tilde{p}(\mathbf{x}) &\geq (\mathbf{r}^T, s)\begin{pmatrix}\mathbf{x}\\1\end{pmatrix} - \tilde{p}(\mathbf{x}) \\
&= ((\mathbf{r}^T, s) - (\mathbf{c}^T, d))\begin{pmatrix}\mathbf{x}\\1\end{pmatrix} \\
&\geq -\left\|(\mathbf{r}^T, s) - (\mathbf{c}^T, d)\right\|\left\|\begin{pmatrix}\mathbf{x}\\1\end{pmatrix}\right\| \\
&\geq -\text{dist}((\mathbf{r}^T, s), \tilde{P}) \cdot \rho \\
&\geq -\max_{(\mathbf{c}^T, d) \in V_{\tilde{P}}} \text{dist}((\mathbf{c}^T, d), P) \cdot \rho.
\end{aligned}
$$

Finally, we obtain that

$$-\max_{(\mathbf{c}^T, d) \in V_{\tilde{P}}} \text{dist}((\mathbf{c}^T, d), P) \cdot \rho \leq p(\mathbf{x}) - \tilde{p}(\mathbf{x}) \leq \max_{(\mathbf{a}^T, b) \in V_P} \text{dist}((\mathbf{a}^T, b), \tilde{P}) \cdot \rho,$$

which implies

$$\frac{1}{\rho}|p(\mathbf{x}) - \tilde{p}(\mathbf{x})| \leq \max\left\{\max_{(\mathbf{a}^T, b) \in V_P} \text{dist}((\mathbf{a}^T, b), \tilde{P}), \max_{(\mathbf{c}^T, d) \in V_{\tilde{P}}} \text{dist}((\mathbf{c}^T, d), P)\right\}, \quad \forall\mathbf{x} \in B.$$

Therefore, by Lemma A.4, we have,

$$\frac{1}{\rho}\max_{\mathbf{x} \in B}|p(\mathbf{x}) - \tilde{p}(\mathbf{x})| \leq H(P, \tilde{P}),$$

□

### A.4 Proofs of Theorem 4 and Corollary 5

We continue with the proofs of Theorem 4 and Corollary 5.

*Proof (Thm. 4).* For the output function, it holds that

$$v(\mathbf{x}) = p(\mathbf{x}) - q(\mathbf{x}), \quad \tilde{v}(\mathbf{x}) = \tilde{p}(\mathbf{x}) - \tilde{q}(\mathbf{x}).$$

From the triangle inequality, we deduce

$$\frac{1}{\rho} |v(\mathbf{x}) - \tilde{v}(\mathbf{x})| \leq H(P, \tilde{P}) + H(Q, \tilde{Q}).$$

Thus, it suffices to get a bound on $H(P, \tilde{P})$ and $H(Q, \tilde{Q})$.

Let $I_+ \subseteq [n]$ and $I_- \subseteq [n]$ be the sets of positive and negative generators, respectively. First, we deal with $H(P, \tilde{P})$. Notice that $\forall \mathbf{v} \in V_{\tilde{P}}, \mathbf{v}$ is the sum of generators of some clusters of $P$. Thus $\forall \mathbf{v} \in V_{\tilde{P}}, \mathbf{v}$ is a vertex of $P$. Hence,

$$\text{dist}(P, \mathbf{v}) = 0, \quad \forall \mathbf{v} \in V_{\tilde{P}}.$$

Let $I_k \subseteq [n]$ be the set of generators that belong to cluster $k$. Let $C_+ \subseteq [K]$ be the set of clusters of positive generators, and $C_- \subseteq [K]$ the set of clusters of negative generators.

Consider any vertex $\mathbf{u}$ of $P$. This vertex can be written as the sum of generators $c_i(\mathbf{a}_i^T, b_i)$, for some subset $I'_+ \subseteq I_+$. Thus,

$$\mathbf{u} = \sum_{i \in I'_+} c_i(\mathbf{a}_i^T, b_i).$$

For every positive generator $c_i(\mathbf{a}_i^T, b_i)$ that belongs to cluster $k$, define

$$x_i = \underset{x \in [0, 1/|I_k|]}{\arg \min} \left\| c_i(\mathbf{a}_i^T, b_i) - x \tilde{c}_k(\tilde{\mathbf{a}}_k^T, \tilde{b}_k) \right\| \in \left[ 0, \frac{1}{|I_k|} \right],$$

i.e., project the generator onto its cluster representative (under a restriction).

For every cluster $k \in C_+$ define $I'_k = I_k \cap I'_+$ and

$$\tilde{x}_k = \sum_{i \in I'_k} x_i.$$

Since $x_i \in [0, 1/|I_k|]$ and $I'_k \subseteq I_k$, we have

$$\tilde{x}_k = \sum_{i \in I'_k} x_i \leq \frac{|I'_k|}{|I_k|} \leq 1.$$

Thus, for every cluster $k \in C_+$, the point $\tilde{x}_k \tilde{c}_k(\tilde{\mathbf{a}}_k^T, \tilde{b}_k)$ lies inside the segment $[\mathbf{0}, \tilde{c}_k(\tilde{\mathbf{a}}_k^T, \tilde{b}_k)]$ and thus belongs to $\tilde{P}$.

For the vertex $\mathbf{u}$, we choose to compare it with the point

$$\sum_{k \in C_+} \tilde{x}_k \tilde{c}_k(\tilde{\mathbf{a}}_k^T, \tilde{b}_k) \in \tilde{P}.$$

Thus, we have that

$$
\begin{aligned}
\operatorname{dist}(\mathbf{u}, \tilde{P}) &\leq \left\| \sum_{i \in I'_+} c_i(\mathbf{a}_i^T, b_i) - \sum_{k \in C_+} \tilde{x}_k \tilde{c}_k(\tilde{\mathbf{a}}_k^T, \tilde{b}_k) \right\| \\
&\leq \sum_{k \in C_+} \left\| \sum_{i \in I'_k} c_i(\mathbf{a}_i^T, b_i) - \tilde{x}_k \tilde{c}_k(\tilde{\mathbf{a}}_k^T, \tilde{b}_k) \right\| \\
&\leq \sum_{k \in C_+} \left\| \sum_{i \in I'_k} \left[ c_i(\mathbf{a}_i^T, b_i) - x_i \tilde{c}_k(\tilde{\mathbf{a}}_k^T, \tilde{b}_k) \right] \right\| \\
&\leq \sum_{k \in C_+} \sum_{i \in I'_k} \left\| c_i(\mathbf{a}_i^T, b_i) - x_i \tilde{c}_k(\tilde{\mathbf{a}}_k^T, \tilde{b}_k) \right\| \\
&= \sum_{k \in C_+} \sum_{i \in I'_k} \min_{x \in [0, 1/|I_k|]} \left\| c_i(\mathbf{a}_i^T, b_i) - x \tilde{c}_k(\tilde{\mathbf{a}}_k^T, \tilde{b}_k) \right\| \\
&= \sum_{k \in C_+} \sum_{i \in I'_k} \min_{x \in [0, 1/|I_k|]} \left\| c_i(\mathbf{a}_i^T, b_i) - x |I_k| (c_i(\mathbf{a}_i^T, b_i) + \boldsymbol{\epsilon}_i) \right\| \\
&\leq \sum_{k \in C_+} \sum_{i \in I'_k} \min_{x \in [0, 1/|I_k|]} \left\{ |1 - x|I_k|| \cdot \left\| c_i(\mathbf{a}_i^T, b_i) \right\| + |x|I_k|| \cdot \|\boldsymbol{\epsilon}_i\| \right\} \\
&= \sum_{k \in C_+} \sum_{i \in I'_k} \min \left\{ \left\| c_i(\mathbf{a}_i^T, b_i) \right\|, \|\boldsymbol{\epsilon}_i\| \right\} \\
&\leq \sum_{k \in C_+} \sum_{i \in I'_k} \min \left\{ \left\| c_i(\mathbf{a}_i^T, b_i) \right\|, \delta_{\max} \right\} \\
&= \sum_{i \in I'_+} \min \left\{ \left\| c_i(\mathbf{a}_i^T, b_i) \right\|, \delta_{\max} \right\},
\end{aligned}
$$

where $\boldsymbol{\epsilon}_i$ is the error between generator $i$ and the cluster center/mean of K-means.

The maximum value of the upper bound occurs when $I'_+ = I_+$. Thus,

$$
\max_{\mathbf{u} \in V_P} \operatorname{dist}(\mathbf{u}, \tilde{P}) \leq \sum_{i \in I_+} \min \left\{ \left\| c_i(\mathbf{a}_i^T, b_i) \right\|, \delta_{\max} \right\}.
$$

Finally, we have

$$
\begin{aligned}
H(P, \tilde{P}) &= \max \left\{ \max_{\mathbf{u} \in V_P} \operatorname{dist}(\mathbf{u}, \tilde{P}), \max_{\mathbf{v} \in V_{\tilde{P}}} \operatorname{dist}(P, \mathbf{v}) \right\} \\
&\leq \max \left\{ \sum_{i \in I_+} \min \left\{ \left\| c_i(\mathbf{a}_i^T, b_i) \right\|, \delta_{\max} \right\}, 0 \right\} \\
&= \sum_{i \in I_+} \min \left\{ \left\| c_i(\mathbf{a}_i^T, b_i) \right\|, \delta_{\max} \right\}.
\end{aligned}
$$

Similarly, for $H(Q, \tilde{Q})$, we have

$$
H(Q, \tilde{Q}) \leq \sum_{i \in I_-} \min \left\{ \left\| c_i(\mathbf{a}_i^T, b_i) \right\|, \delta_{\max} \right\}.
$$

Combining, we get

$$
\frac{1}{\rho} |v(\mathbf{x}) - \tilde{v}(\mathbf{x})| \leq \sum_{i \in I} \min \left\{ \left\| c_i(\mathbf{a}_i^T, b_i) \right\|, \delta_{\max} \right\},
$$

which concludes the proof of Theorem 4. $\qquad \square$

*Proof (Corollary 5).* The bound of Misiakos et al. (2022) is the following:

$$\frac{1}{\rho} \max_{\mathbf{x} \in B} |v(\mathbf{x}) - \tilde{v}(\mathbf{x})| \le K\delta_{\max} + \left(1 - \frac{1}{N_{\max}}\right) \sum_{i=1}^{n} |c_i| \|(\mathbf{a}_i^T, b_i)\|.$$

We will show that

$$K\delta_{\max} + \left(1 - \frac{1}{N_{\max}}\right) \sum_{i=1}^{n} |c_i| \|(\mathbf{a}_i^T, b_i)\| \ge \sum_{i \in I} \min\left\{\left\|c_i(\mathbf{a}_i^T, b_i)\right\|, \delta_{\max}\right\}.$$

This can be rewritten as

$$K\delta_{\max} + \left(1 - \frac{1}{N_{\max}}\right) \sum_{i=1}^{n} |c_i| \|(\mathbf{a}_i^T, b_i)\| \ge \sum_{i \in I} \left\|c_i(\mathbf{a}_i^T, b_i)\right\| + \sum_{i \in I} \min\left\{0, \delta_{\max} - \left\|c_i(\mathbf{a}_i^T, b_i)\right\|\right\}.$$

Further simplifying, we get

$$K\delta_{\max} \ge \frac{1}{N_{\max}} \sum_{i=1}^{n} |c_i| \|(\mathbf{a}_i^T, b_i)\| + \sum_{i=1}^{n} \min\left\{0, \delta_{\max} - \left\|c_i(\mathbf{a}_i^T, b_i)\right\|\right\}.$$

It suffices to show that for every cluster $k$, we have:

$$\delta_{\max} \ge \frac{1}{|I_k|} \sum_{i \in I_k} |c_i| \|(\mathbf{a}_i^T, b_i)\| + \sum_{i \in I_k} \min\left\{0, \delta_{\max} - |c_i| \|(\mathbf{a}_i^T, b_i)\|\right\}.$$

However, it holds that

$$\sum_{i \in I_k} \min\left\{0, \delta_{\max} - |c_i| \|(\mathbf{a}_i^T, b_i)\|\right\} \le \delta_{\max} - \max_{i \in I_k} |c_i| \|(\mathbf{a}_i^T, b_i)\|,$$

and

$$\max_{i \in I_k} |c_i| \|(\mathbf{a}_i^T, b_i)\| \ge \frac{1}{|I_k|} \sum_{i \in I_k} |c_i| \|(\mathbf{a}_i^T, b_i)\|.$$

Hence, we have

$$\sum_{i \in I_k} \min\left\{0, \delta_{\max} - |c_i| \|(\mathbf{a}_i^T, b_i)\|\right\} + \frac{1}{|I_k|} \sum_{i \in I_k} |c_i| \|(\mathbf{a}_i^T, b_i)\|$$
$$\le \delta_{\max} - \max_{i \in I_k} |c_i| \|(\mathbf{a}_i^T, b_i)\| + \frac{1}{|I_k|} \sum_{i \in I_k} |c_i| \|(\mathbf{a}_i^T, b_i)\| \le \delta_{\max},$$

which concludes the proof. $\square$

## A.5 Proof of Theorem 6

Before we proceed with the proof of Theorem 6, we first give the definition of null neurons and generators.

**Definition A.5** (Null neuron/generator). *A neuron/generator $i \in [n]$ that belongs to cluster $k \in [K]$ is a null neuron/generator with respect to output $j \in [m]$ if $\tilde{c}_{jk} c_{ji} \le 0$. $N_j$ is the set of all null neurons with respect to output $j$.*

*Proof (Thm. 6).* Assume the algorithm's iterative scheme has reached a stationary point (otherwise, assume the last step is an output weight update and the proof works fine). First, we focus on a single output, say $j$-th output. We will bound $H(P_j, \tilde{P}_j), H(Q_j, \tilde{Q}_j)$ for all $j \in [m]$.

Let $I_{j+}, I_{j-}$ be the sets of positive and negative generators of output $j$. Let $I_k$ be the set of neurons that belong to cluster $k$. Let $C_{j+}$ be the set of positive clusters for output $j$ (i.e. clusters for which $\tilde{c}_{jk} > 0$), and $C_{j-}$ be the set of negative clusters for output $j$. Let $I_{jk} = I_k \cap I_{j+}$ if $k$ is a positive cluster, else $I_{jk} = I_k \cap I_{j-}$

Consider any vertex $\mathbf{u}$ of $P_j$. This vertex can be written as the sum of generators $c_{ji}(\mathbf{a}_i^T, b_i)$, for some subset $I'_{j+} \subseteq I_{j+}$. Thus,

$$\mathbf{u} = \sum_{i \in I'_{j+}} c_{ji}(\mathbf{a}_i^T, b_i).$$

For every generator $c_{ji}(\mathbf{a}_i^T, b_i), i \in I_{j+}$ that belongs to positive cluster $k \in C_{j+}$, define

$$x_{ji} = \arg\min_{x \in [0, 1/|I_{jk}|]} \left\| c_{ji}(\mathbf{a}_i^T, b_i) - x\tilde{c}_{jk}(\tilde{\mathbf{a}}_k^T, \tilde{b}_k) \right\| \in \left[0, \frac{1}{|I_{jk}|}\right],$$

i.e., project the generator onto $\tilde{c}_{jk}(\tilde{\mathbf{a}}_k^T, \tilde{b}_k)$ of its cluster $k$ (under a restriction).

Assume a variant of the algorithm, where the optimization criterion is the following:

$$\sum_{j=1}^m \left\| |\tilde{c}_{jk}|(\tilde{\mathbf{a}}_k^T, \tilde{b}_k) - \sum_{i \in I_{jk}} |c_{ji}|(\mathbf{a}_i^T, b_i) \right\|^2.$$

The sign of $\tilde{c}_{jk}$ never changes and gets fixed based on the initial solution. The set $I_{jk}$ of the non-null generators of cluster $k$ in terms of output $j$ depends on the sign of $\tilde{c}_{jk}$, and it is determined by the initial solution. The output weight update rule changes: We update the absolute value of the weight $|\tilde{c}_{jk}|$. The update is performed as normal if the result is positive, otherwise we set $|\tilde{c}_{jk}| = 0$. We deduce that throughout the execution of the algorithm, after every output weight update step the following holds:

- If $|\tilde{c}_{jk}| = 0$, then every generator of cluster $k$ is a null generator by definition.

- If $|\tilde{c}_{jk}| > 0$, then the following argument holds.

For every cluster $k \in C_{j+}$ define $I'_{jk} = I_k \cap I'_{j+}$ and

$$\tilde{x}_{jk} = \sum_{i \in I'_{jk}} x_{ji}.$$

Since $x_{ji} \in [0, 1/|I_{jk}|]$ and $I'_{jk} \subseteq I_{jk}$, we have

$$\tilde{x}_{jk} = \sum_{i \in I'_{jk}} x_{ji} \leq \frac{|I'_{jk}|}{|I_{jk}|} \leq 1.$$

Thus, for every cluster $k \in C_{j+}$, the point $\tilde{x}_{jk}\tilde{c}_{jk}(\tilde{\mathbf{a}}_k^T, \tilde{b}_k)$ lies inside the segment $[\mathbf{0}, \tilde{c}_{jk}(\tilde{\mathbf{a}}_k^T, \tilde{b}_k)]$, and thus belongs to $\tilde{P}_j$.

For the vertex $\mathbf{u}$, we choose to compare it with the point

$$\sum_{k \in C_{j+}} \tilde{x}_{jk}\tilde{c}_{jk}(\tilde{\mathbf{a}}_k^T, \tilde{b}_k) \in \tilde{P}_j.$$

In addition, we can also consider the unrestricted projection $x'_{ji}$ over $\mathbb{R}$ instead of $[0, 1/|I_{jk}|]$. Then, by the definition of the output weight update step, it will hold that

$$\tilde{x}'_{jk} = \sum_{i \in I_{jk}} x'_{ji} = 1, \quad \text{and} \quad \tilde{c}_{jk}(\tilde{\mathbf{a}}_k^T, \tilde{b}_k) = \sum_{i \in I_{jk}} x'_{ji}\tilde{c}_{jk}(\tilde{\mathbf{a}}_k^T, \tilde{b}_k).$$

We have that

$$\tilde{c}_{jk}(\tilde{\mathbf{a}}_k^T, \tilde{b}_k) = \sum_{i \in I_{jk}} c_{ji}(\mathbf{a}_i^T, b_i) + l_{jk} = |I_{jk}| \frac{\sum_{i \in I_{jk}} c_{ji}(\mathbf{a}_t^T, b_i)}{|I_{jk}|} + l_{jk} = |I_{jk}|(c_{ji}(\mathbf{a}_i^T, b_i) + \boldsymbol{\epsilon}_{ji}) + l_{jk},$$

where $\sum_{j=1}^{m} \|l_{jk}\|^2 = l_k^2$ the optimization criterion loss, and $\epsilon_{ji}$ is the error between $c_{ji}(\mathbf{a}_i^T, b_i)$ and the mean $\frac{\sum_{i \in I_{jk}} c_{ji}(\mathbf{a}_t^T, b_i)}{|I_{jk}|}$. It is easy to verify that $\epsilon_{ji}$ tends to zero as $\delta_{max}$ tends to 0.

Thus, we have that

$$
\begin{aligned}
\text{dist}(\mathbf{u}, \tilde{P}_j) &\leq \left\| \sum_{i \in I'_{j+}} c_{ji}(\mathbf{a}_i^T, b_i) - \sum_{k \in C_{j+}} \tilde{x}_{jk} \tilde{c}_{jk}(\tilde{\mathbf{a}}_k^T, \tilde{b}_k) \right\| \\
&\leq \sum_{k \in C_{j+}} \left\| \sum_{i \in I'_{jk}} c_{ji}(\mathbf{a}_i^T, b_i) - \tilde{x}_{jk} \tilde{c}_{jk}(\tilde{\mathbf{a}}_k^T, \tilde{b}_k) \right\| + \sum_{i \in N_{j+}} |c_{ji}| \|(\mathbf{a}_i^T, b_i)\| \\
&\leq \sum_{k \in C_{j+}} \left\| \sum_{i \in I'_{jk}} [c_{ji}(\mathbf{a}_i^T, b_i) - x_{ji} \tilde{c}_{jk}(\tilde{\mathbf{a}}_k^T, \tilde{b}_k)] \right\| + \sum_{i \in N_{j+}} |c_{ji}| \|(\mathbf{a}_i^T, b_i)\| \\
&\leq \sum_{k \in C_{j+}} \sum_{i \in I'_{jk}} \left\| c_{ji}(\mathbf{a}_i^T, b_i) - x_{ji} \tilde{c}_{jk}(\tilde{\mathbf{a}}_k^T, \tilde{b}_k) \right\| + \sum_{i \in N_{j+}} |c_{ji}| \|(\mathbf{a}_i^T, b_i)\| \\
&\leq \sum_{k \in C_{j+}} \sum_{i \in I'_{jk}} \min_{x \in [0, 1/|I_{jk}|]} \left\| c_{ji}(\mathbf{a}_i^T, b_i) - x \left( |I_{jk}|(c_{ji}(\mathbf{a}_i^T, b_i) + \epsilon_{ji}) + l_{jk} \right) \right\| + \sum_{i \in N_{j+}} |c_{ji}| \|(\mathbf{a}_i^T, b_i)\| \\
&\leq \sum_{k \in C_{j+}} \sum_{i \in I'_{jk}} \min_{x \in [0, 1/|I_{jk}|]} \left\| (1 - x|I_{jk}|) c_{ji}(\mathbf{a}_i^T, b_i) - x l_{jk} - x|I_{jk}|\epsilon_{ji} \right\| + \sum_{i \in N_{j+}} |c_{ji}| \|(\mathbf{a}_i^T, b_i)\| \\
&\leq \sum_{k \in C_{j+}} \sum_{i \in I'_{jk}} \min_{x \in [0, 1/|I_{jk}|]} \left\{ |1 - x|I_{jk}|| \|c_{ji}(\mathbf{a}_i^T, b_i)\| + |x|I_{jk}|| \left\| \frac{l_{jk}}{|I_{jk}|} + \epsilon_{ji} \right\| \right\} + \sum_{i \in N_{j+}} |c_{ji}| \|(\mathbf{a}_i^T, b_i)\| \\
&\leq \sum_{k \in C_{j+}} \sum_{i \in I'_{jk}} \min \left\{ \|c_{ji}(\mathbf{a}_i^T, b_i)\|, \left\| \frac{l_{jk}}{|I_{jk}|} + \epsilon_{ji} \right\| \right\} + \sum_{i \in N_{j+}} |c_{ji}| \|(\mathbf{a}_i^T, b_i)\| \\
&\leq \sum_{k \in C_{j+}} \sum_{i \in I'_{jk}} \min \left\{ \|c_{ji}(\mathbf{a}_i^T, b_i)\|, \frac{\|l_{jk}\|}{|I_{jk}|} + \|\epsilon_{ji}\| \right\} + \sum_{i \in N_{j+}} |c_{ji}| \|(\mathbf{a}_i^T, b_i)\|.
\end{aligned}
$$

The maximum value of the upper bound occurs when $I'_{j+} = I_{j+}$. Thus, we have

$$
\max_{\mathbf{u} \in V_{P_j}} \text{dist}(\mathbf{u}, \tilde{P}_j) \leq \sum_{k \in C_{j+}} \sum_{i \in I_{jk}} \min \left\{ \|c_{ji}(\mathbf{a}_i^T, b_i)\|, \frac{\|l_{jk}\|}{|I_{jk}|} + \|\epsilon_{ji}\| \right\} + \sum_{i \in N_{j+}} |c_{ji}| \|(\mathbf{a}_i^T, b_i)\|.
$$

To obtain a bound for $\max_{\mathbf{v} \in V_{\tilde{P}_j}} \text{dist}(P_j, \mathbf{v})$, we write $\mathbf{v} = \sum_{k \in C'_{j+}} \tilde{c}_{jk}(\tilde{\mathbf{a}}_k^T, \tilde{b}_k) \in \tilde{P}_j$ and choose vertex $\mathbf{u} = \sum_{i \in I'_{j+}} c_{ji}(\mathbf{a}_i^T, b_i)$, with $I'_{j+} = \{i \in I_{j+} | i \in I_k, k \in C'_{j+}\}$. This set contains $I_{jk}$, and thus $\tilde{x}'_{jk} = 1$. We can repeat the above steps with unrestricted projections. We will then get minima over $\mathbb{R}$ instead of $[0, 1/|I_{jk}|]$, which are smaller. Hence, this distance has already been taken into account in the calculation of the bound of $\max_{\mathbf{u} \in V_{P_j}} \text{dist}(\mathbf{u}, \tilde{P}_j)$.

At last, we have

$$
H(P_j, \tilde{P}_j) \leq \sum_{k \in C_{j+}} \sum_{i \in I_{jk}} \min \left\{ \|c_{ji}(\mathbf{a}_i^T, b_i)\|, \frac{\|l_{jk}\|}{|I_{jk}|} + \|\epsilon_{ji}\| \right\} + \sum_{i \in N_{j+}} |c_{ji}| \|(\mathbf{a}_i^T, b_i)\|.
$$

Similarly, for $H(Q_j, \tilde{Q}_j)$ we have

$$
H(Q_j, \tilde{Q}_j) \leq \sum_{k \in C_{j-}} \sum_{i \in I_{jk}} \min \left\{ \|c_{ji}(\mathbf{a}_i^T, b_i)\|, \frac{\|l_{jk}\|}{|I_{jk}|} + \|\epsilon_{ji}\| \right\} + \sum_{i \in N_{j-}} |c_{ji}| \|(\mathbf{a}_i^T, b_i)\|.
$$

Combining, we obtain

$$\frac{1}{\rho} \max_{\mathbf{x} \in B} |v_j(\mathbf{x}) - \tilde{v}_j(\mathbf{x})| \le \sum_{k=1}^{m} \sum_{i \in I_{jk}} \min\left\{ \|c_{ji}(\mathbf{a}_i^T, b_i)\|, \frac{\|l_{jk}\|}{|I_{jk}|} + \|\boldsymbol{\epsilon}_{ji}\| \right\} + \sum_{i \in N_j} |c_{ji}| \|(\mathbf{a}_i^T, b_i)\|.$$

Using the fact that $I_{jk} \subseteq I_k$ and $N_{min} \le |I_{jk}|, \forall j, k$ we have

$$\frac{1}{\rho} \max_{\mathbf{x} \in B} |v_j(\mathbf{x}) - \tilde{v}_j(\mathbf{x})| \le \sum_{k=1}^{m} \sum_{i \in I_k} \min\left\{ \|c_{ji}(\mathbf{a}_i^T, b_i)\|, \frac{\|l_{jk}\|}{N_{min}} + \|\boldsymbol{\epsilon}_{ji}\| \right\} + \sum_{i \in N_j} |c_{ji}| \|(\mathbf{a}_i^T, b_i)\|.$$

We make use of the following inequality, which is a direct consequence of Cauchy-Schwartz Inequality

$$\sum_{j=1}^{m} |u_j| \le \sqrt{m} \sqrt{\sum_{j=1}^{m} |u_j|^2} = \sqrt{m} \|(u_1, \dots, u_m)\|.$$

We have

$$\sum_{j=1}^{m} |c_{ji}| \le \sqrt{m} \|\mathbf{C}_{:,i}\|,$$

$$\sum_{j=1}^{m} \|l_{jk}\| \le \sqrt{m} \sqrt{\sum_{j=1}^{m} \|l_{jk}\|^2} = \sqrt{m} \cdot l_k,$$

$$\sum_{j=1}^{m} \|\boldsymbol{\epsilon}_{ji}\| \le \sqrt{m} \sqrt{\sum_{j=1}^{m} \|\boldsymbol{\epsilon}_{ji}\|^2} = \sqrt{m} \|\mathbf{E}_i\|_F.$$

Using the above inequalities, the fact that $\sum \min \le \min \sum$, and the fact that $\max \sum \le \sum \max$ we get

$$\begin{aligned}
\frac{1}{\rho} \max_{\mathbf{x} \in B} \|\mathbf{v}(\mathbf{x}) - \tilde{\mathbf{v}}(\mathbf{x})\|_1 &\le \sum_{k=1}^{m} \sum_{i \in I_k} \min\left\{ \sum_{j=1}^{m} |c_{ji}| \|(\mathbf{a}_i^T, b_i)\|, \frac{\sum_{j=1}^{m} \|l_{jk}\|}{N_{min}} + \sum_{j=1}^{m} \|\boldsymbol{\epsilon}_{ji}\| \right\} \\
&\quad + \sum_{j=1}^{m} \sum_{i \in N_j} |c_{ji}| \|(\mathbf{a}_i^T, b_i)\| \\
&\le \sqrt{m} \sum_{i=1}^{n} \min\left\{ \|\mathbf{C}_{:,i}\| \|(\mathbf{a}_i^T, b_i)\|, \frac{l_{k(i)}}{N_{min}} + \|\mathbf{E}_i\|_F \right\} \\
&\quad + \sum_{j=1}^{m} \sum_{i \in N_j} |c_{ji}| \|(\mathbf{a}_i^T, b_i)\|,
\end{aligned}$$

as desired. $\qquad\square$

## A.6 Assumptions and Proof of Theorem 7

For this and the next subsection throughout, we assume convergence of the iterative process; in particular the update rules hold with equality for the final weights.

**Assumption A.6** (Assumptions (Compact clusters, Weight-decay, ReLU))**.** Here, we formalize and verify the assumptions needed by Theorem 7. Note that Theorem 7 has two parts. For the second part some assumptions hold automatically. We will formalize and verify the assumptions of each part.

We start with the first part. Let

$$\mathbf{M} = \sum_{i=1}^{n} \mathbf{c}_i (\mathbf{a}_i^T, b_i),$$

where, as mentioned, we hide the ReLU activation. For this reason, if $\langle (\mathbf{a}_i^T, b_i), (\mathbf{a}_{i'}^T, b_{i'}) \rangle < 0$, then $\mathrm{ReLU}(\langle (\mathbf{a}_i^T, b_i), (\mathbf{a}_{i'}^T, b_{i'}) \rangle) = 0$, and this contribution does not pass through. Hence,

$$\frac{\mathbf{c}_i^T}{\|\mathbf{c}_i\|} \mathbf{M} \frac{(\mathbf{a}_i, b_i)}{\|(\mathbf{a}_i, b_i)\|} = \|\mathbf{c}_i\| \|(\mathbf{a}_i, b_i)\| + \sum_{\substack{i' \neq i \\ \langle (\mathbf{a}_i^T, b_i), (\mathbf{a}_{i'}^T, b_{i'}) \rangle \geq 0}} \frac{\langle (\mathbf{a}_i^T, b_i), (\mathbf{a}_{i'}^T, b_{i'}) \rangle}{\|(\mathbf{a}_i^T, b_i)\| \|(\mathbf{a}_{i'}^T, b_{i'})\|} \frac{\langle \mathbf{c}_i, \mathbf{c}_{i'} \rangle}{\|\mathbf{c}_i\| \|\mathbf{c}_{i'}\|}.$$

What this means is that if we have negative alignment of input weights $\langle (\mathbf{a}_i^T, b_i), (\mathbf{a}_{i'}^T, b_{i'}) \rangle < 0$, then ReLU will prevent cancellation and allow the maximum of the two contributions to pass through.

Now, to control cancellation due to negative output weight alignment, we formalize our weight-decay assumption. First, consider two neurons that have positive-positive input alignment and positive-negative output (mis)alignment, say $(\mathbf{a}_1, b_1) = C_1(\mathbf{a}_2, b_2)$, $\mathbf{c}_1 = -C_2\mathbf{c}_2$, $C_1, C_2 \geq 0$. Then, $c_{j1}\mathrm{ReLU}(\mathbf{a}_1^T\mathbf{x} + b_1) = -C_1 C_2 c_{j2}\mathrm{ReLU}(\mathbf{a}_2^T\mathbf{x} + b_2)$, and their contribution cancels. Using instead weights $\mathbf{c}_1' = (1 - C_1 C_2)\mathbf{c}_1$, $\mathbf{c}_2' = \mathbf{0}$, we would get a network with the same output function, but with a smaller $\ell_2$ magnitude of weights. Thus, weight-decay would not allow such a misalignment in the output weights.

If, for simplicity, we take $C_1 = 1$, then note that replacing the output weights with $\mathbf{c}_1' = \mathbf{c}_2' = \frac{(1-C_2)}{2}\mathbf{c}_1$ (which are positively aligned) would yield an even smaller weight-decay loss while still maintaining the same function. Thus, although weight-decay discourages output misalignment, it actually encourages uniformity in the output weights and redundancy.

In the above example, we have perfect input alignment (i.e., $\langle (\mathbf{a}_i^T, b_i), (\mathbf{a}_{i'}^T, b_{i'}) \rangle = 1$), and weight-decay prevents output weight misalignment (i.e., $\langle \mathbf{c}_i, \mathbf{c}_{i'} \rangle \not< 0$). If, instead, we have almost perfect input alignment, then replacing the output weights as done above causes a very small error in the output function. We assume that, in practice, the network can "learn to work around" this small error, to allow for reduction in the weight-decay loss.

As input weight alignment reduces, output weight misalignment increases. Our weight-decay assumption is an assumption on how these two inverse rates relate to each other via their product. More precisely, we assume that there exists at least one direction for which the total cancellation leaves this direction's contribution unchanged up to a constant factor (for example, in the above expression, the sum of products does not cancel the first term by a factor greater than some constant $C < 1$). We further require that it holds after merging the neurons into their clusters $k$. We relax this by assuming it holds after partitioning the clusters into compact classes $[k^\star]$. Formally, using quantities to be defined later in the proof, "hiding" the ReLU argument for easy of exposition, our weight-decay assumption gives that there exists at least one $[k^\star]$ such that, in the following, when (1) is less than 1, then (2) is sufficiently greater than $-1$ so that it holds

$$\sum_{k \in [k^\star]} \left( |I_k| \|\bar{\mathbf{c}}_k\| \|(\bar{\mathbf{a}}_k^T, \bar{b}_k)\| s_1(k) s_2(k) \right) \|\bar{\mathbf{c}}_{[k^\star]}\| \|(\bar{\mathbf{a}}_{[k^\star]}, \bar{b}_{[k^\star]})\|$$

$$+ \sum_{[l^\star] \neq [k^\star]} \sum_{l \in [l^\star]} \left( |I_l| \|\bar{\mathbf{c}}_l\| \|(\bar{\mathbf{a}}_l^T, \bar{b}_l)\| s_1(l) s_2(l) \right) \|\bar{\mathbf{c}}_{[l^\star]}\| \|(\bar{\mathbf{a}}_{[l^\star]}, \bar{b}_{[l^\star]})\| \underbrace{\frac{\langle (\bar{\mathbf{a}}_{[k^\star]}, \bar{b}_{[k^\star]}), (\bar{\mathbf{a}}_{[l^\star]}, \bar{b}_{[l^\star]}) \rangle}{\|(\bar{\mathbf{a}}_{[k^\star]}, \bar{b}_{[k^\star]})\| \|(\bar{\mathbf{a}}_{[l^\star]}, \bar{b}_{[l^\star]})\|}}_{(1)} \underbrace{\frac{\langle \bar{\mathbf{c}}_{[k^\star]}, \bar{\mathbf{c}}_{[l^\star]} \rangle}{\|\bar{\mathbf{c}}_{[k^\star]}\| \|\bar{\mathbf{c}}_{[l^\star]}\|}}_{(2)}$$

$$\geq \Omega(1) \sum_{k \in [k^\star]} \left( |I_k| \|\bar{\mathbf{c}}_k\| \|(\bar{\mathbf{a}}_k^T, \bar{b}_k)\| \right) \|\bar{\mathbf{c}}_{[k^\star]}\| \|(\bar{\mathbf{a}}_{[k^\star]}, \bar{b}_{[k^\star]})\|. \tag{5}$$

We verify empirically that this assumption holds in practice. In Figure 9 we notice that this inequality holds with $\Omega(1) \geq 1$ for $[k] = \{k\}$ and all different pruning ratios used.

The second assumption that we make is that of compact clusters (formal definition in the proof of Theorem 7). We assume that the cluster centers $K$ are enough so that the clusters are compact. This assumption is actually related more to the compressibility of the network itself: the fewer the necessary centers for compactness, the more compressible the network. We verify this empirically in Figure 9. The compactness of the clusters follows a reverse relation to the performance after compression.

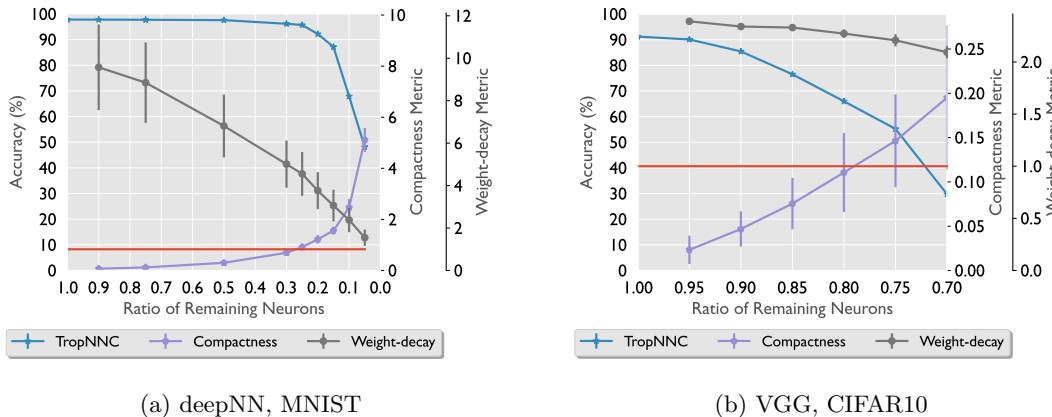

(a) deepNN, MNIST            (b) VGG, CIFAR10

Figure 9: Empirical verification of assumptions on a feedforward deepNN (left) and on a convolutional VGG (right) network. For the weight-decay assumption, we plot the greatest value $\Omega(1)$ such that the bound (5) holds. For all pruning ratios, this constant exceeds 1 (red threshold), and the assumption holds (even in our favor). Note that for VGG, $\Omega(1)$ was not uniform among the different layers, with deeper layers giving better values than earlier ones. The plot presents $\Omega(1)$ for an earlier layer giving worse/smaller $\Omega(1)$. Still, it holds $\Omega(1) > 1$. For cluster compactness, we plot the average of the smallest $\epsilon_k$ satisfying the definition. It follows a reverse relation to the accuracy of the compressed model, indicating it is linked to the compressibility of the network. **Note** that the quantities follow a different scale (TropNNC $\rightarrow$ "Accuracy (%)" scale; Compactness $\rightarrow$ "Compactness Metric" scale; Weight-decay + red threshold $\rightarrow$ "Weight-decay Metric" scale.)

Finally, we discuss the assumptions of the second part of Theorem 7. For its proof, two new instances of the first part of the proof are constructed, for which either the input weights of the clusters are orthogonal, or the output weights of the clusters are orthogonal. Hence, with $[k] = \{k\}$, the cross terms are all 0 and the weight-decay assumption holds automatically (an $\Omega(1) \geq 1/2$ factor is still needed by the ReLU argument).

*Proof (Thm. 7).* We assume that the clustering satisfies Assumption A.6. First, notice that two clusters can be approximately parallel or antiparallel. If this is the case, we cannot isolate just one of them, because the others will contribute to approximately the same singular directions. For this reason, we will merge the clusters into compact classes satisfying Assumption A.6.

Assume that each cluster $k$ has cluster center $(\bar{\mathbf{a}}_k, \bar{b}_k, \bar{\mathbf{c}}_k)$, and consider the lines that go through the vectors $(\bar{\mathbf{a}}_k, \bar{b}_k)$ and $\bar{\mathbf{c}}_k$. Choose an angle tolerance $\theta$. Build a graph with an edge $(k, k')$ if and only if their lines have angle less than $\theta$ for both types of vectors, which is undirected. The connected components of this graph induce a partition of the clusters. Let $[k]$ be the connected component/class of cluster $k$. Define the merged cluster $I_{[k]} = \bigcup_{k' \in [k]} I_k$. For every class $[k]$ fix a representative $k^{\star} \in [k]$, and define $s_1(k), s_2(k) \in \{-1, 1\}$ to indicate whether $(\bar{\mathbf{a}}_k, \bar{b}_k)$ and $\bar{\mathbf{c}}_k$ are co-directional with the respective vectors of $k^{\star}$. Define the center of the class $[k^*]$ to be

$$(\bar{\mathbf{a}}_{[k^\star]}, \bar{b}_{[k^\star]}) = \frac{\sum_{k \in [k^\star]} s_2(k)|I_k|\|\bar{\mathbf{c}}_k\|(\bar{\mathbf{a}}_k, \bar{b}_k)}{\sum_{k \in [k^\star]} |I_k|\|(\bar{\mathbf{a}}_k, \bar{b}_k)\|\|\bar{\mathbf{c}}_k\|s_1(k)s_2(k)}, \quad \bar{\mathbf{c}}_{[k^\star]} = \frac{\sum_{k \in [k^\star]} s_1(k)|I_k|\|(\bar{\mathbf{a}}_k, \bar{b}_k)\|\bar{\mathbf{c}}_k}{\sum_{k \in [k^\star]} |I_k|\|(\bar{\mathbf{a}}_k, \bar{b}_k)\|\|\bar{\mathbf{c}}_k\|s_1(k)s_2(k)}.$$

Note that the class center differs from the weighted sum of the merged points. It includes a form of normalization.

We now formalize the assumptions. We require i) cluster compactness:

$$\forall i \in I_k, \quad \|(\mathbf{a}_i, b_i) - (\bar{\mathbf{a}}_k, \bar{b}_k)\| \leq \|(\bar{\mathbf{a}}_k, \bar{b}_k)\|\epsilon_{k1}/2, \quad \|\mathbf{c}_i - \bar{\mathbf{c}}_k\| \leq \|\bar{\mathbf{c}}_k\|\epsilon_{k2}/2,$$

$$\forall k \in [k^\star], \quad \left\| s_1(k)\frac{(\bar{\mathbf{a}}_k, \bar{b}_k)}{\|(\bar{\mathbf{a}}_k, \bar{b}_k)\|} - (\bar{\mathbf{a}}_{[k^\star]}, \bar{b}_{[k^\star]}) \right\| \leq \epsilon_{k1}/2 \quad \text{and} \quad \left\| s_2(k)\frac{\bar{\mathbf{c}}_k}{\|\bar{\mathbf{c}}_k\|} - \bar{\mathbf{c}}_{[k^\star]} \right\| \leq \epsilon_{k2}/2.$$

and ii) that the weight-decay induced bound of Assumption A.6 holds for the clusters and their partitioning into classes.

Now, define $\mathbf{M}_{[k^\star]} = \sum_{k \in [k^\star]} \mathbf{M}_k$, and similarly for the residual operators. We control the second singular value of each $\mathbf{M}_{[k^\star]}$. For a fixed $k$, write

$$(\mathbf{a}_i, b_i) = (\bar{\mathbf{a}}_k, \bar{b}_k) + \delta(\mathbf{a}_i, b_i), \quad \mathbf{c}_i = \bar{\mathbf{c}}_k + \delta\mathbf{c}_i$$

with

$$\|\delta(\mathbf{a}_i, b_i)\| \leq \|(\bar{\mathbf{a}}_k, \bar{b}_k)\| \epsilon_{k1}/2, \quad \|\delta\mathbf{c}_i\| \leq \|\bar{\mathbf{c}}_k\| \epsilon_{k2}/2.$$

Then, by expanding, we have

$$\mathbf{M}_k = |I_k| \bar{\mathbf{c}}_k (\bar{\mathbf{a}}_k^T, \bar{b}_k) + \mathbf{E}_k,$$

where, after using $\sum_i \delta(\mathbf{a}_i, b_i) = \sum_i \delta\mathbf{c}_i = 0$,

$$\mathbf{E}_k = \sum_{i \in I_k} (\delta\mathbf{c}_i(\bar{\mathbf{a}}_k^T, \bar{b}_k) + \bar{\mathbf{c}}_k\delta(\mathbf{a}_i^T, b_i) + \delta\mathbf{c}_i\delta(\mathbf{a}_i^T, b_i)) = \sum_{i \in I_k} \delta\mathbf{c}_i\delta(\mathbf{a}_i^T, b_i).$$

By properties of norms, we obtain

$$\|\mathbf{E}_k\|_2 \leq \sum_{i \in I_k} \|\delta\mathbf{c}_i\|\|\delta(\mathbf{a}_i, b_i)\| \leq |I_k|\|\bar{\mathbf{c}}_k\|\|(\bar{\mathbf{a}}_k, \bar{b}_k)\|\epsilon_{k1}\epsilon_{k2}/4.$$

Now consider the whole class $[k^\star]$. We have,

$$\mathbf{M}_{[k^\star]} = \sum_{k \in [k^\star]} \mathbf{M}_k$$

$$= \sum_{k \in [k^\star]} |I_k| \bar{\mathbf{c}}_k (\bar{\mathbf{a}}_k^T, \bar{b}_k) + \sum_{k \in [k^\star]} \mathbf{E}_k.$$

$$\sum_{k \in [k^\star]} |I_k|\|\bar{\mathbf{c}}_k\|\|(\bar{\mathbf{a}}_k^T, \bar{b}_k)\| \frac{\bar{\mathbf{c}}_k}{\|\bar{\mathbf{c}}_k\|} \frac{(\bar{\mathbf{a}}_k^T, \bar{b}_k)}{\|(\bar{\mathbf{a}}_k^T, \bar{b}_k)\|} + \sum_{k \in [k^\star]} \mathbf{E}_k.$$

Now write

$$\frac{(\bar{\mathbf{a}}_k, \bar{b}_k)}{\|(\bar{\mathbf{a}}_k, \bar{b}_k)\|} = s_1(k)(\bar{\mathbf{a}}_{[k^\star]}, \bar{b}_{[k^\star]}) + \delta(\bar{\mathbf{a}}_k, \bar{b}_k), \quad \frac{\bar{\mathbf{c}}_k}{\|\bar{\mathbf{c}}_k\|} = s_2(k)\bar{\mathbf{c}}_{[k^\star]} + \delta\bar{\mathbf{c}}_k \quad (6)$$

with

$$\|\delta(\bar{\mathbf{a}}_k, \bar{b}_k)\| \leq \epsilon_{k1}/2, \quad \|\delta\bar{\mathbf{c}}_k\| \leq \epsilon_{k2}/2.$$

Then, again by expanding, we get

$$\mathbf{M}_{[k^\star]} = \sum_{k \in [k^\star]} \left(|I_k|\|\bar{\mathbf{c}}_k\|\|(\bar{\mathbf{a}}_k^T, \bar{b}_k)\| s_1(k)s_2(k)\right) \bar{\mathbf{c}}_{[k^\star]}(\bar{\mathbf{a}}_{[k^\star]}^T, \bar{b}_{[k^\star]}) + \mathbf{E}'_{[k^\star]} + \sum_{k \in [k^\star]} \mathbf{E}_k,$$

where,

$$\mathbf{E}'_{[k^\star]} = \sum_{k \in [k^\star]} \left(|I_k|\|\bar{\mathbf{c}}_k\|\|(\bar{\mathbf{a}}_k^T, \bar{b}_k)\| s_1(k)\delta\bar{\mathbf{c}}_k\right) (\bar{\mathbf{a}}_{[k^\star]}^T, \bar{b}_{[k^\star]})$$

$$+ \bar{\mathbf{c}}_{[k^\star]} \sum_{k \in [k^\star]} \left(|I_k|\|\bar{\mathbf{c}}_k\|\|(\bar{\mathbf{a}}_k^T, \bar{b}_k)\| s_2(k)\delta(\bar{\mathbf{a}}_k^T, \bar{b}_k)\right).$$

$$+ \sum_{k \in [k^\star]} \left(|I_k|\|\bar{\mathbf{c}}_k\|\|(\bar{\mathbf{a}}_k^T, \bar{b}_k)\| \delta\bar{\mathbf{c}}_k\delta(\bar{\mathbf{a}}_k^T, \bar{b}_k)\right).$$

If we plug in the approximations in (6) into the definition of the class centers (which are a normalized mean), we obtain that the first two sums above are equal to 0. Thus, we get

$$\mathbf{E}'_{[k^\star]} = \sum_{k \in [k^\star]} \left(|I_k|\|\bar{\mathbf{c}}_k\|\|(\bar{\mathbf{a}}_k^T, \bar{b}_k)\| \delta\bar{\mathbf{c}}_k\delta(\bar{\mathbf{a}}_k^T, \bar{b}_k)\right).$$

By properties of norms, we obtain,

$$\|\mathbf{E}'_{[k^\star]}\|_2 \leq \sum_{k \in [k^\star]} \left( |I_k| \|\bar{\mathbf{c}}_k\| \|(\bar{\mathbf{a}}_k, \bar{b}_k)\| \epsilon_{k1}\epsilon_{k2}/4 \right)$$

Thus, we have

$$\mathbf{M}_{[k^\star]} = \mathbf{M}^{(0)}_{[k^\star]} + \mathbf{E}_{[k^\star]},$$

with

$$\mathbf{M}^{(0)}_{[k^\star]} = \sum_{k \in [k^\star]} \left( |I_k| \|\bar{\mathbf{c}}_k\| \|(\bar{\mathbf{a}}_k^T, \bar{b}_k)\| s_1(k)s_2(k) \right) \bar{\mathbf{c}}_{[k^\star]}(\bar{\mathbf{a}}_{[k^\star]}^T, \bar{b}_{[k^\star]})$$

and

$$\mathbf{E}_{[k^\star]} = \mathbf{E}'_{[k^\star]} + \sum_{k \in [k^\star]} \mathbf{E}_k,$$

$$\|\mathbf{E}_{[k^\star]}\|_2 \leq 2 \sum_{k \in [k^\star]} \left( |I_k| \|\bar{\mathbf{c}}_k\| \|(\bar{\mathbf{a}}_k, \bar{b}_k)\| \epsilon_{k1}\epsilon_{k2}/4 \right) \leq \sum_{k \in [k^\star]} \left( |I_k| \|\bar{\mathbf{c}}_k\| \|(\bar{\mathbf{a}}_k, \bar{b}_k)\| \right) \epsilon_{k1}\epsilon_{k2}.$$

Setting

$$\eta_k = \epsilon_{k1}\epsilon_{k2}$$

we get

$$\|\mathbf{E}_{[k^\star]}\|_2 \leq \sum_{k \in [k^\star]} \left( |I_k| \|\bar{\mathbf{c}}_k\| \|(\bar{\mathbf{a}}_k, \bar{b}_k)\| \right) \eta_k.$$

In the expression of $\mathbf{M}_{[k^\star]}$, the term $\mathbf{M}^{(0)}_{[k^\star]}$ has rank 1. Applying Weyl's inequality,

$$\sigma_2(\mathbf{M}_{[k^\star]}) \leq \|\mathbf{E}_{[k^\star]}\|_2 \leq \sum_{k \in [k^\star]} \left( |I_k| \|\bar{\mathbf{c}}_k\| \|(\bar{\mathbf{a}}_k, \bar{b}_k)\| \right) \eta_k.$$

Now we lower bound the spectral norm of $\mathbf{M}$. We have

$$\mathbf{M} = \sum_{[k^\star]} \mathbf{M}_{[k^\star]} = \mathbf{M}^{(0)} + \sum_{[k^\star]} \mathbf{E}_{[k^\star]},$$

with

$$\mathbf{M}^{(0)} = \sum_{[k^\star]} \mathbf{M}^{(0)}_{[k^\star]} = \sum_{[k^\star]} \sum_{k \in [k^\star]} \left( |I_k| \|\bar{\mathbf{c}}_k\| \|(\bar{\mathbf{a}}_k^T, \bar{b}_k)\| s_1(k)s_2(k) \right) \bar{\mathbf{c}}_{[k^\star]}(\bar{\mathbf{a}}_{[k^\star]}^T, \bar{b}_{[k^\star]}).$$

After expanding, for every $[k^\star]$ we have that

$$\frac{\bar{\mathbf{c}}_{[k^\star]}^T}{\|\bar{\mathbf{c}}_{[k^\star]}\|} \mathbf{M}^{(0)} \frac{(\bar{\mathbf{a}}_{[k^\star]}, \bar{b}_{[k^\star]})}{\|(\bar{\mathbf{a}}_{[k^\star]}, \bar{b}_{[k^\star]})\|} = \sum_{k \in [k^\star]} \left( |I_k| \|\bar{\mathbf{c}}_k\| \|(\bar{\mathbf{a}}_k^T, \bar{b}_k)\| s_1(k)s_2(k) \right) \|\bar{\mathbf{c}}_{[k^\star]}\| \|(\bar{\mathbf{a}}_{[k^\star]}, \bar{b}_{[k^\star]})\|$$

$$+ \sum_{[l^\star] \neq [k^\star]} \sum_{l \in [l^\star]} \left( |I_l| \|\bar{\mathbf{c}}_l\| \|(\bar{\mathbf{a}}_l^T, \bar{b}_l)\| s_1(l)s_2(l) \right) \|\bar{\mathbf{c}}_{[l^\star]}\| \|(\bar{\mathbf{a}}_{[l^\star]}, \bar{b}_{[l^\star]})\| \frac{\langle (\bar{\mathbf{a}}_{[k^\star]}, \bar{b}_{[k^\star]}), (\bar{\mathbf{a}}_{[l^\star]}, \bar{b}_{[l^\star]}) \rangle}{\|(\bar{\mathbf{a}}_{[k^\star]}, \bar{b}_{[k^\star]})\| \|(\bar{\mathbf{a}}_{[l^\star]}, \bar{b}_{[l^\star]})\|} \frac{\langle \bar{\mathbf{c}}_{[k^\star]}, \bar{\mathbf{c}}_{[l^\star]} \rangle}{\|\bar{\mathbf{c}}_{[k^\star]}\| \|\bar{\mathbf{c}}_{[l^\star]}\|}.$$

Taking maximum over all $[k^*]$ and using Assumption A.6, and since $\|M\|_2 = \sup_{\|x\|, \|y\|=1} |y^T M x|$, we obtain that

$$\|\mathbf{M}^{(0)}\|_2 \geq \Omega(1) \max_{[k^\star]} \sum_{k \in [k^\star]} \left( |I_k| \|\bar{\mathbf{c}}_k\| \|(\bar{\mathbf{a}}_k^T, \bar{b}_k)\| \right) \|\bar{\mathbf{c}}_{[k^\star]}\| \|(\bar{\mathbf{a}}_{[k^\star]}, \bar{b}_{[k^\star]})\|$$

Now consider again the definition of the class centers. Simple manipulation (divide and multiply by the norm of the vectors in the sum) shows that these are effectively weighted averages of normalized vectors. In addition, the second condition of cluster compactness implies that they are ($\epsilon/2$)-close to any of these

normalized vectors, and hence they will be $(\epsilon/2)$-close to the unit sphere. This implies that $1 - \epsilon/2 \leq \|\bar{\mathbf{c}}_{[k^\star]}\|, \|(\bar{\mathbf{a}}_{[k^\star]}, \bar{b}_{[k^\star]})\| \leq 1 + \epsilon/2$. For small $\epsilon$, we have

$$\|\mathbf{M}^{(0)}\|_2 \geq \Omega(1) \max_{[k^\star]} \sum_{k \in [k^\star]} \left( |I_k| \|\bar{\mathbf{c}}_k\| \|(\bar{\mathbf{a}}_k^T, \bar{b}_k)\| \right).$$

In addition, we have

$$\|\mathbf{M} - \mathbf{M}^{(0)}\|_2 = \|\sum_{[k^\star]} \mathbf{E}_{[k^\star]}\|_2 \leq \sum_{[k^\star]} \sum_{k \in [k^\star]} \left( |I_k| \|\bar{\mathbf{c}}_k\| \|(\bar{\mathbf{a}}_k, \bar{b}_k)\| \right) \eta_k = \sum_k \left( |I_k| \|\bar{\mathbf{c}}_k\| \|(\bar{\mathbf{a}}_k, \bar{b}_k)\| \right) \eta_k$$

Choosing $\eta = \sum_k \left( |I_k| \|\bar{\mathbf{c}}_k\| \|(\bar{\mathbf{a}}_k, \bar{b}_k)\| \right) \eta_k / \sum_k \left( |I_k| \|\bar{\mathbf{c}}_k\| \|(\bar{\mathbf{a}}_k, \bar{b}_k)\| \right)$, $\eta_k = \epsilon_{k1} \epsilon_{k2}$ as

$$\eta \leq O(1) \frac{\max_{[k^\star]} (\sum_{k \in [k^\star]} (|I_k| \|\bar{\mathbf{c}}_k\| \|(\bar{\mathbf{a}}_k^T, \bar{b}_k)\|))}{\sum_k (|I_k| \|\bar{\mathbf{c}}_k\| \|(\bar{\mathbf{a}}_k, \bar{b}_k)\|)}$$

we obtain that

$$\|\mathbf{M}\|_2 \geq \|\mathbf{M}^{(0)}\|_2 - \|\mathbf{M} - \mathbf{M}^{(0)}\|_2 \geq \Omega(1) \max_{[k^\star]} \sum_{k \in [k^\star]} \left( |I_k| \|\bar{\mathbf{c}}_k\| \|(\bar{\mathbf{a}}_k^T, \bar{b}_k)\| \right).$$

Now, consider the best rank-1 approximations $\widetilde{M_k}$ of each cluster, and the corresponding residual errors. Define the residual

$$\mathbf{R}_k = \mathbf{M}_k - \widetilde{\mathbf{M}}_k, \quad \|\mathbf{R}\|_2 = \sigma_2(\mathbf{M}_k).$$

Using the obtained expression for $\mathbf{M}_k$,

$$\|\mathbf{R}_k\|_2 = \sigma_2(\mathbf{M}_k) \leq \|\mathbf{E}_k\|_2.$$

Set

$$\mathbf{R} = \sum_k \mathbf{R}_k.$$

Then we have that,

$$\mathbf{M} = \sum_k \mathbf{M}_k = \sum_k \widetilde{\mathbf{M}}_k + \sum_k \mathbf{R}_k = \widetilde{\mathbf{M}} + \mathbf{R}$$

$$\Rightarrow \|\widetilde{\mathbf{M}}\|_2 \leq \|\mathbf{M}\|_2 + \|\mathbf{R}\| \leq \|\mathbf{M}\|_2 + \sum_k \|\mathbf{R}_k\|_2,$$

By the bound on the spectral norm of $\mathbf{E}_k$ we have

$$\sum_k \|\mathbf{R}_k\|_2 \leq \sum_k |I_k| \|\bar{\mathbf{c}}_k\| \|(\bar{\mathbf{a}}_k, \bar{b}_k)\| \eta_k / 4 = \sum_{[k^\star]} \sum_{k \in [k^\star]} |I_k| \|\bar{\mathbf{c}}_k\| \|(\bar{\mathbf{a}}_k, \bar{b}_k)\| \eta_k / 4.$$

Therefore for the chosen (asymptotic) upper bound for $\eta$ chosen above, we have

$$\frac{\|\mathbf{R}\|_2}{\|\mathbf{M}\|_2} \leq O(1) \frac{\sum_{[k^\star]} \sum_{k \in [k^\star]} |I_k| \|\bar{\mathbf{c}}_k\| \|(\bar{\mathbf{a}}_k, \bar{b}_k)\|}{\max_{[k^\star]} \sum_{k \in [k^\star]} \left( |I_k| \|\bar{\mathbf{c}}_k\| \|(\bar{\mathbf{a}}_k^T, \bar{b}_k)\| \right)} \eta \leq O(1).$$

This implies that

$$\|\widetilde{\mathbf{M}}\|_2 \leq \|\mathbf{M}\|_2 + O(1) \|\mathbf{M}\|_2 = O(1) \|\mathbf{M}\|_2 \leq O(1) \|\mathbf{C}\|_2 \|(\mathbf{A}, \mathbf{b})\|_2.$$

To complete the theorem, we prove the second pair of inequalities. Writing the rank-1 matrix $\widetilde{\mathbf{M}}_k = \tilde{\mathbf{c}}_k(\tilde{\mathbf{a}}_k^T, \tilde{b}_k)$, we have that

$$\mathbf{M}_k = (|I_k|\bar{\mathbf{c}}_k)(\bar{\mathbf{a}}_k^T, \bar{b}_k) + \mathbf{E}_k = \tilde{\mathbf{c}}_k(\tilde{\mathbf{a}}_k^T, \tilde{b}_k) + \mathbf{R}_k.$$

$$\Rightarrow (|I_k|\bar{\mathbf{c}}_k)(\bar{\mathbf{a}}_k^T, \bar{b}_k) = \tilde{\mathbf{c}}_k(\tilde{\mathbf{a}}_k^T, \tilde{b}_k) + \mathbf{R}_k - \mathbf{E}_k.$$

We have two rank-1 matrices perturbed by a controlled error. We can apply Weyl's inequality and an eigengap theorem to control the error in magnitude and angle between the singular vectors. We have

$$|\|(|I_k|\bar{\mathbf{c}}_k)\|_2\|(\bar{\mathbf{a}}_k^T, \bar{b}_k)\|_2 - \|\tilde{\mathbf{c}}_k\|_2\|(\tilde{\mathbf{a}}_k^T, \tilde{b}_k)\|_2| \leq \|\mathbf{R}_k\|_2 + \|\mathbf{E}_k\|_2 \leq 2\|\mathbf{E}_k\|_2$$

$$\Rightarrow \left| 1 - \frac{\|\tilde{\mathbf{c}}_k\|_2\|(\tilde{\mathbf{a}}_k^T, \tilde{b}_k)\|_2}{\|(|I_k|\bar{\mathbf{c}}_k)\|_2\|(\bar{\mathbf{a}}_k^T, \bar{b}_k)\|_2} \right| \leq \frac{2\|\mathbf{E}_k\|_2}{|I_k|\|\bar{\mathbf{c}}_k\|_2\|(\bar{\mathbf{a}}_k^T, \bar{b}_k)\|_2} \leq \eta_k/2,$$

$$\sin \angle(\bar{\mathbf{c}}_k, \tilde{\mathbf{c}}_k) \leq \frac{2\|\mathbf{E}\|_2}{|I_k|\|\bar{\mathbf{c}}_k\|_2\|(\bar{\mathbf{a}}_k^T, \bar{b}_k)\|_2} \leq \eta_k/2,$$

$$\sin \angle((\bar{\mathbf{a}}_k^T, \bar{b}_k), (\tilde{\mathbf{a}}_k^T, \tilde{b}_k)) \leq \frac{2\|\mathbf{E}\|_2}{|I_k|\|\bar{\mathbf{c}}_k\|_2\|(\bar{\mathbf{a}}_k^T, \bar{b}_k)\|_2} \leq \eta_k/2.$$

Setting $\widetilde{\mathbf{D}} = \mathrm{diag}_k(d_k)$ with $d_k = \sqrt{\frac{\|\tilde{\mathbf{c}}_k\|_2\|(\tilde{\mathbf{a}}_k^T, \tilde{b}_k)\|_2}{\|\bar{\mathbf{c}}_k\|_2\|(\bar{\mathbf{a}}_k^T, \bar{b}_k)\|_2}}$, for $\eta/2 \leq 1$ the first inequality implies that

$$\Rightarrow \left| 1 - \frac{\|\tilde{\mathbf{c}}_k\|_2 d_k^{-1}}{\sqrt{|I_k|}\|\bar{\mathbf{c}}_k\|_2} \right| = \left| 1 - \frac{d_k\|(\tilde{\mathbf{a}}_k^T, \tilde{b}_k)\|_2}{\sqrt{|I_k|}\|(\bar{\mathbf{a}}_k^T, \bar{b}_k)\|_2} \right| = 1 - \sqrt{\frac{\|\tilde{\mathbf{c}}_k\|_2\|(\tilde{\mathbf{a}}_k^T, \tilde{b}_k)\|_2}{\|(|I_k|\bar{\mathbf{c}}_k)\|_2\|(\bar{\mathbf{a}}_k^T, \bar{b}_k)\|_2}} \leq 1 - \sqrt{1 - \eta_k^2/4} \leq \eta_k/2.$$

The perturbation inequalities theorems can be applied in reverse direction to give two rank-1 matrices with small error. In particular, we make the following replacements in the problem instance:

$$(\tilde{\mathbf{a}}_k, \tilde{b}_k) \rightarrow \mathbf{e}_k, \quad (\mathbf{a}_i, b_i) \rightarrow \frac{\mathbf{e}_k}{\sqrt{|I_k|}}, \quad \forall i \in I_k,$$

where $\mathbf{e}_k$ is the $k$-th unit vector of the canonical basis of $\mathbb{R}^K$ (i.e. in the new instance we also change the dimension of the input to $d = K$). Notice that for the new problem instance, we have

$$(\bar{\mathbf{a}}_k, \bar{b}_k) \rightarrow \frac{\mathbf{e}_k}{\sqrt{|I_k|}}, \quad \|(\bar{\mathbf{a}}_k, \bar{b}_k)\|_2 \rightarrow \frac{1}{\sqrt{|I_k|}}.$$

Thus, we can write

$$\left| 1 - \frac{\|\tilde{\mathbf{c}}_k\|_2 d_k^{-1}\|(\tilde{\mathbf{a}}_k, \tilde{b}_k)\|_2}{|I_k|\|\bar{\mathbf{c}}_k\|_2\|(\bar{\mathbf{a}}_k, \bar{b}_k)\|_2} \right| \leq \eta_k/2.$$

The output weights did not change direction, while for the input weights the sin-angle becomes zero. Hence, the perturbation inequalities hold for the new instance, and we obtain that

$$\Rightarrow (|I_k|\bar{\mathbf{c}}_k)\mathbf{e}_k^T = d_k^{-1}\tilde{\mathbf{c}}_k\mathbf{e}_k^T + \text{error},$$

where the error is of the order of $|I_k|\|\bar{\mathbf{c}}_k\|_2\|(\bar{\mathbf{a}}_k, \bar{b}_k)\|_2\eta \rightarrow \sqrt{|I_k|}\|\bar{\mathbf{c}}_k\|_2\eta$. Notice that in the new instance, the per-cluster means $(\bar{\mathbf{a}}_k, \bar{b}_k) \parallel \mathbf{e}_k$ are orthogonal, and hence by setting $[k] = \{k\}$ all assumptions remain intact and the problem instance is easier (in particular, the weight-decay bound holds with $\Omega(1) = 1$). Also notice that for the new instance, we have $(\widetilde{\mathbf{A}}, \tilde{\mathbf{b}}) = I$ and $\|(\mathbf{A}, \mathbf{b})\|_2 \leq 1$ (as each neuron has a unique input which is compensated by $\frac{1}{\sqrt{|I_k|}}$). Thus, by using the first part of the proof on the new problem instance, we obtain that

$$\|\widetilde{\mathbf{C}}\widetilde{\mathbf{D}}^{-1}\|_2 = \|\widetilde{\mathbf{C}}\widetilde{\mathbf{D}}^{-1}(\widetilde{\mathbf{A}}, \tilde{\mathbf{b}})\|_2 \leq O(1)\|\mathbf{C}(\mathbf{A}, \mathbf{b})\|_2 \leq O(1)\|\mathbf{C}\|_2\|(\mathbf{A}, \mathbf{b})\|_2 \leq O(1)\|\mathbf{C}\|_2,$$

provided that we have

$$\eta \leq O(1)\frac{\max_k(\sqrt{|I_k|}\|\bar{\mathbf{c}}_k\|_2)}{\sum_k(\sqrt{|I_k|}\|\bar{\mathbf{c}}_k\|_2)}.$$

We do the same for the input weights, making the following replacements:

$$\tilde{\mathbf{c}}_k \rightarrow \mathbf{e}_k, \quad \mathbf{c}_i \rightarrow \frac{\mathbf{e}_k}{\sqrt{|I_k|}}, \forall i \in I_k,$$

this time changing the output weights of the instance (and also the output dimension to $m = K$). With similar arguments as above, we obtain that

$$\|\widetilde{\mathbf{D}}(\widetilde{\mathbf{A}}, \widetilde{\mathbf{b}})\|_2 = \|\widetilde{\mathbf{C}}\widetilde{\mathbf{D}}(\widetilde{\mathbf{A}}, \widetilde{\mathbf{b}})\|_2 \le O(1)\|\mathbf{C}(\mathbf{A}, \mathbf{b})\|_2 \le O(1)\|\mathbf{C}\|_2\|(\mathbf{A}, \mathbf{b})\|_2 \le O(1)\|(\mathbf{A}, \mathbf{b})\|_2,$$

provided that we have

$$\eta \le O(1)\frac{\max_k(\sqrt{|I_k|}\|(\bar{\mathbf{a}}_k, \bar{b}_k)\|_2)}{\sum_k(\sqrt{|I_k|}\|(\bar{\mathbf{a}}_k, \bar{b}_k)\|_2)}.$$

Finally, we relate the above result with our algorithm. We restate the optimization objective and weight update rules:

$$\text{minimize} \quad \sum_{j=1}^{m}\left\|\tilde{c}_{jk}(\tilde{\mathbf{a}}_k^T, \tilde{b}_k) - \sum_{i \in I_k} c_{ji}(\mathbf{a}_i^T, b_i)\right\|^2,$$

$$\tilde{c}_{jk} = \frac{\left\langle \sum_{i \in I_k} c_{ji}(\mathbf{a}_i^T, b_i), (\tilde{\mathbf{a}}_k^T, \tilde{b}_k)\right\rangle}{\left\|(\tilde{\mathbf{a}}_k^T, \tilde{b}_k)\right\|^2},$$

$$(\tilde{\mathbf{a}}_k^T, \tilde{b}_k) = \frac{\sum_{j=1}^{m}\sum_{i \in I_k} c_{ji}(\mathbf{a}_i^T, b_i)\widetilde{c}_{jk}}{\sum_{j=1}^{m}\widetilde{c}_{jk}^2}.$$

We can rewrite these in matrix-form respectively as follows:

$$\text{minimize} \quad \sum_{j=1}^{m}\left\|\left(\tilde{\mathbf{c}}_k(\tilde{\mathbf{a}}_k^T, \tilde{b}_k) - \sum_{i \in I_k} \mathbf{c}_i(\mathbf{a}_i^T, b_i)\right)_j\right\|^2 = \left\|\tilde{\mathbf{c}}_k(\tilde{\mathbf{a}}_k^T, \tilde{b}_k) - \mathbf{M}_k\right\|_F^2,$$

$$\tilde{\mathbf{c}}_k = \frac{\mathbf{M}_k(\tilde{\mathbf{a}}_k, \tilde{b}_k)}{\left\|(\tilde{\mathbf{a}}_k^T, \tilde{b}_k)\right\|^2},$$

$$(\tilde{\mathbf{a}}_k, \tilde{b}_k) = \frac{\mathbf{M}_k^T \tilde{\mathbf{c}}_k}{\|\tilde{\mathbf{c}}_k\|^2}.$$

Putting these together (assuming convergence), we obtain

$$\tilde{\mathbf{c}}_k^T \mathbf{M}_k(\tilde{\mathbf{a}}_\mathbf{k}, \tilde{b}) = \left\|(\tilde{\mathbf{a}}_k^T, \tilde{b}_k)\right\|^2\|\tilde{\mathbf{c}}_k\|^2.$$

In particular, this means that $\widetilde{\mathbf{M}}_k = \tilde{\mathbf{c}}_k(\tilde{\mathbf{a}}_k^T, \tilde{b})$ is a rank-1 approximation of a singular value of $\mathbf{M}_k$. In addition, in the above derived expression $\mathbf{M}_k = |I_k|\bar{\mathbf{c}}_k(\bar{\mathbf{a}}_k^T, \bar{b}_k) + \mathbf{E}_k$, we have that $(\bar{\mathbf{a}}_k^T, \bar{b}_k)$ and $|I_k|\bar{\mathbf{c}}_k$ are exactly the weights returned by the non-iterative algorithm, and thus for the small $\eta$ that we assumed above, the initial solution will be better than a $\sigma_2(\mathbf{M}_k)$ rank-1 approximation. Since the iterative method strictly decreases the objective, it follows that the solutions of the iterative method will converge to one that corresponds to a best rank-1 approximation of $\mathbf{M}_k$. $\qquad\square$

## A.7 Proof of Theorem 8

We continue with the proof of Theorem 8.

*Proof (Thm. 8).* First of all, ReLU is 1-Lipschitz in $\ell_2$, i.e. for every $\mathbf{x}$ we have

$$\|\text{ReLU}(\mathbf{x})\| \le \|\mathbf{x}\|.$$

This implies that

$$\|\mathbf{x}^{(q)}\| = \|\text{ReLU}(\mathbf{v}^{(\mathbf{q})} + \mathbf{b}^{(q)})\| \le \|\mathbf{v}^{(q)} + \mathbf{b}^{(q)}\| = \|\mathbf{A}^{(q)}\mathbf{x}^{(q-1)} + \mathbf{b}^{(q)}\|.$$

By the definition of spectral norm, it follows that

$$\|\mathbf{x}^{(q)}\| \leq \|(\mathbf{A}^{(q)}, \mathbf{b}^{(q)})\|\|(\mathbf{x}^{(q-1)}, 1)\|$$

$$\Rightarrow \|\mathbf{x}^{(q)}\|^2 \leq \|(\mathbf{A}^{(q)}, \mathbf{b}^{(q)})\|^2(\|\mathbf{x}^{(q-1)}\|^2 + 1)$$

$$\Rightarrow \|\mathbf{x}^{(q)}\|^2 + 1 \leq 1 + \|(\mathbf{A}^{(q)}, \mathbf{b}^{(q)})\|^2(\|\mathbf{x}^{(q-1)}\|^2 + 1).$$

By expanding this recurrence (with empty sums and products being equal to 0 and 1 respectively), we obtain

$$\|\mathbf{x}^{(q)}\|^2 + 1 \leq \left(\prod_{q'=1}^{q} \|(\mathbf{A}^{(q')}, \mathbf{b}^{(q')})\|^2\right)(\|\mathbf{x}\|^2 + 1) + \sum_{q'=1}^{q}\left(\prod_{q''=q'+1}^{q} \|(\mathbf{A}^{(q'')}, \mathbf{b}^{(q'')})\|^2\right).$$

Taking square roots, by subadditivity of square roots (for positive summands),

$$\sqrt{\|\mathbf{x}^{(q)}\|^2 + 1} \leq \left(\prod_{q'=1}^{q} \|(\mathbf{A}^{(q')}, \mathbf{b}^{(q')})\|\right)\sqrt{\|\mathbf{x}\|^2 + 1} + \sum_{q'=1}^{q}\left(\prod_{q''=q'+1}^{q} \|(\mathbf{A}^{(q'')}, \mathbf{b}^{(q'')})\|\right)$$

$$= p_2(1, q)\sqrt{\|\mathbf{x}\|^2 + 1} + \sum_{q'=1}^{q} p_2(q'+1, q).$$

Hence, for $\mathbf{x} \in B = \{\mathbf{x} : \sqrt{\|\mathbf{x}\|^2 + 1} \leq \rho\}$, we have

$$\mathbf{x}^{(q)} \in B^{(q)} = \{\mathbf{x} : \sqrt{\|\mathbf{x}\|^2 + 1} \leq \rho^{(q)} = p_2(1, q)\rho + \sum_{q'=1}^{q} p_2(q'+1, q)\}$$

$$\Rightarrow \frac{\rho^{(q)}}{\rho} \leq p_2(1, q) + \sum_{q'=1}^{q} p_2(q'+1, q)/\rho \leq p_2(1, q) + \sum_{q'=1}^{q} p_2(q'+1, q).$$

Now apply the previous lemma to some layer to obtain that

$$\|\widetilde{\mathbf{C}}\widetilde{\mathbf{D}}^{-1}\|_2 \leq O(1)\|\mathbf{C}\|_2, \quad \|\widetilde{\mathbf{D}}(\widetilde{\mathbf{A}}, \tilde{\mathbf{b}})\|_2 \leq O(1)\|(\mathbf{A}, \mathbf{b})\|_2,$$

for some known diagonal (rescaling) matrix $\widetilde{\mathbf{D}}$. Henceforth, we assume that this rescaling is applied as a final step of the compression algorithm, i.e. we set $\widetilde{\mathbf{C}} \leftarrow \widetilde{\mathbf{C}}\widetilde{\mathbf{D}}^{-1}$ and $(\widetilde{\mathbf{A}}, \tilde{\mathbf{b}}) \leftarrow \widetilde{\mathbf{D}}(\widetilde{\mathbf{A}}, \tilde{\mathbf{b}})$. We emphasize that this rescaling of the algorithm's returned weights, although important for the theoretical analysis, was not used in the implementation of the algorithm in the experiments (note the difference between scaling the final weights, and scaling the original weights that are used for clustering as stated in Heuristic 1).

We return back to the notation of the full network. We compress all hidden layers in increasing order of depth. Each linear transformation matrix $\mathbf{A}^{(q)}$ is affected by the compression of exactly 2 layers, first as the output weights of the compressed layer, and then as the input weights of the very next compressed layer. We denote by $\widetilde{\mathbf{A}}^{(q)}$ and $\widetilde{\widetilde{\mathbf{A}}}^{(q)}$ the weight matrix after each layer compression respectively. $\mathbf{A}^{(1)}$ is an exception; it is affected only by one compression step. To simplify the resulting expressions, we set $\mathbf{A}^{(1)} = \widetilde{\mathbf{A}}^{(1)}$, i.e. we pretend that the first weight matrix has taken part in the hypothetical compression of layer 0. With this notation in place, when compressing layer $q$, setting $(\mathbf{A}, \mathbf{b}) \leftarrow (\widetilde{\mathbf{A}}^{(q)}, \mathbf{b}^{(q)})$ and $\mathbf{C} \leftarrow \mathbf{A}^{(q+1)}$ in the above inequalities, we obtain

$$\|\widetilde{\mathbf{A}}^{(q+1)}\|_2 \leq O(1)\|\mathbf{A}^{(q+1)}\|_2, \quad \|(\widetilde{\widetilde{\mathbf{A}}}^{(q)}, \tilde{\mathbf{b}}^{(q)})\|_2 \leq O(1)\|(\widetilde{\mathbf{A}}^{(q)}, \mathbf{b}^{(q)})\|_2. \tag{7}$$

Thus, we also have that

$$\|\widetilde{\mathbf{A}}^{(q+1)}\|_F \leq \sqrt{K}\|\widetilde{\mathbf{A}}^{(q+1)}\|_2 \leq O(1)\sqrt{K}\|\mathbf{A}^{(q+1)}\|_2 \leq O(1)\sqrt{K}\|\mathbf{A}^{(q+1)}\|_F$$

$$\Rightarrow \|(\widetilde{\mathbf{A}}^{(q+1)}, \mathbf{b}^{(q+1)})\|_F \le O(1)\sqrt{K}\|(\mathbf{A}^{(q+1)}, \mathbf{b}^{(q+1)})\|_F.$$

Finally, for general matrices observe that

$$\|(\mathbf{A}, \mathbf{b})\|_2^2 \ge \max(\|\mathbf{A}\|^2, \|\mathbf{b}\|^2) \ge (\|\mathbf{A}\|^2 + \|\mathbf{b}\|^2)/2$$

$$\Rightarrow \|(\mathbf{A}, \mathbf{b})\|_2 \le \|\mathbf{A}\| + \|\mathbf{b}\| \le \sqrt{2(\|\mathbf{A}\|^2 + \|\mathbf{b}\|^2)} \le 2\|(\mathbf{A}, \mathbf{b})\|_2.$$

Hence, the first inequality of (7) implies that

$$\|(\widetilde{\mathbf{A}}^{(q+1)}, \mathbf{b}^{(q+1)})\|_2 \le \|\widetilde{\mathbf{A}}^{(q+1)}\|_2 + \|\mathbf{b}^{(q+1)}\| \le O(1)\|\mathbf{A}^{(q+1)}\|_2 + \|\mathbf{b}^{(q+1)}\| \le O(1)\|(\mathbf{A}^{(q+1)}, \mathbf{b}^{(q+1)})\|_2.$$

We introduce the last bit of notation. Let $^{(q')}\mathbf{v}^{(q)}$ and $^{(q')}\mathbf{x}^{(q)}$ denote the intermediate outputs at layer $q$ (pre-bias and post-ReLU) after the first $q'$ layers have been compressed (and use similar notation for all other quantities). Notice that the above inequalities imply that throughout the execution of the algorithm, compressing layers expands the intermediate "input domain" of each layer by not too much. In particular, for every $q'$, we have the upper bound

$$^{(q')}\rho^{(q)} \le \left(O(1)^{\min(q'+1, q)}\right) \cdot \rho^{(q)}.$$

In the per-layer bound of Theorem 6, notice that

$$\sum_{i=1}^n \|\mathbf{C}_{:,i}\|\|(\mathbf{a}_i^T, b_i)\| \le \sqrt{\sum_{i=1}^n \|\mathbf{C}_{:,i}\|^2} \sqrt{\sum_{i=1}^n \|(\mathbf{a}_i^T, b_i)\|^2} = \|\mathbf{C}\|_F \|(\mathbf{A}, \mathbf{b})\|_F,$$

$$\|(\mathbf{a}_i^T, b_i)\| \le \|(\mathbf{A}, \mathbf{b})\|_2 \Rightarrow \sum_{j=1}^m \sum_{i \in N_j} |c_{ji}|\|(\mathbf{a}_i^T, b_i)\| \le \|(\mathbf{A}, \mathbf{b})\|_2 \sum_{j=1}^m \sum_{i \in N_j} |c_{ji}|.$$

Applying Theorem 6 with $(\mathbf{A}, \mathbf{b}) \leftarrow (\widetilde{\mathbf{A}}^{(q)}, \mathbf{b}^{(q)})$, $\mathbf{C} \leftarrow \mathbf{A}^{(q+1)}$, and $\mathbf{x} \in {}^{(q-1)}B^{(q-1)}$ on the input of the $q$ hidden layer, and using $\sum \min \le \min \sum$ and the above bounds, we get

$$\frac{1}{{}^{(q-1)}\rho^{(q-1)}} \max_{\mathbf{x} \in B} \|{}^{(q-1)}\mathbf{v}^{(q+1)}(\mathbf{x}) - {}^{(q)}\mathbf{v}^{(q+1)}(\mathbf{x})\|_1 \le$$

$$\sqrt{n_{q+1}} \min \left\{ \|\mathbf{A}^{(q+1)}\|_F \|(\widetilde{\mathbf{A}}^{(q)}, \mathbf{b}^{(q)})\|_F, \sum_{i=1}^{n_q} \frac{l_{k(i)}^{(q)}}{N_{min}} + \|\mathbf{E}_i^{(q)}\|_F \right\} + \|(\widetilde{\mathbf{A}}^{(q)}, \mathbf{b}^{(q)})\|_2 \sum_{j=1}^{n_{q+1}} \sum_{i \in N_j^{(q)}} |a_{ji}^{(q+1)}|.$$

Using the bounds on spectral norms and $^{(q')}\rho^{(q)}$, we get

$$\frac{1}{\rho^{(q-1)}} \max_{\mathbf{x} \in {}^{(q-1)}B^{(q-1)}} \|{}^{(q-1)}\mathbf{v}^{(q+1)}(\mathbf{x}) - {}^{(q)}\mathbf{v}^{(q+1)}(\mathbf{x})\|_1 \le$$

$$O(1)^q \left( \sqrt{n_{q+1}} \min \left\{ \sqrt{K_q}\|\mathbf{A}^{(q+1)}\|_F \|(\mathbf{A}^{(q)}, \mathbf{b}^{(q)})\|_F, \sum_{i=1}^{n_q} \frac{l_{k(i)}^{(q)}}{N_{min}} + \|\mathbf{E}_i^{(q)}\|_F \right\} + \|(\mathbf{A}^{(q)}, \mathbf{b}^{(q)})\|_2 \sum_{j=1}^{n_{q+1}} \sum_{i \in N_j^{(q)}} |a_{ji}^{(q+1)}| \right).$$

Now, notice that ReLU is also 1-Lipschitz in $\ell_1$, and thus

$$\|{}^{(q-1)}\mathbf{v}^{(q+2)}(\mathbf{x}) - {}^{(q)}\mathbf{v}^{(q+2)}(\mathbf{x})\|_1 = \|\mathbf{A}^{(q+2)}({}^{(q-1)}\mathbf{x}^{(q+1)}(\mathbf{x}) - {}^{(q)}\mathbf{x}^{(q+1)}(\mathbf{x}))\|_1$$

$$\le \|\mathbf{A}^{(q+2)}\|_1 \|{}^{(q-1)}\mathbf{x}^{(q+1)}(\mathbf{x}) - {}^{(q)}\mathbf{x}^{(q+1)}(\mathbf{x})\|_1$$

$$= \|\mathbf{A}^{(q+2)}\|_1 \|\text{ReLU}({}^{(q-1)}\mathbf{v}^{(q+1)}(\mathbf{x}) + \mathbf{b}^{(\mathbf{q+1})}) - \text{ReLU}({}^{(q)}\mathbf{v}^{(q+1)}(\mathbf{x}) + \mathbf{b}^{(\mathbf{q+1})})\|_1$$

$$\le \|\mathbf{A}^{(q+2)}\|_1 \|({}^{(q-1)}\mathbf{v}^{(q+1)}(\mathbf{x}) + \mathbf{b}^{(\mathbf{q+1})}) - ({}^{(q)}\mathbf{v}^{(q+1)}(\mathbf{x}) + \mathbf{b}^{(\mathbf{q+1})})\|_1$$

$$= \|\mathbf{A}^{(q+2)}\|_1 \|{}^{(q-1)}\mathbf{v}^{(q+1)}(\mathbf{x}) - {}^{(q)}\mathbf{v}^{(q+1)}(\mathbf{x})\|_1.$$

Thus, for the propagation of the error to the final layer, by induction it holds that

$$\|^{(q-1)}\mathbf{v}(\mathbf{x}) - {}^{(q)}\mathbf{v}(\mathbf{x})\|_1 \leq \prod_{q'=q+2}^{Q} \|\mathbf{A}^{(q')}\|_1 \|^{(q-1)}\mathbf{v}^{(q+1)}(\mathbf{x}) - {}^{(q)}\mathbf{v}^{(q+1)}(\mathbf{x})\|_1$$

$$= p_1(q+2, Q)\|^{(q-1)}\mathbf{v}^{(q+1)}(\mathbf{x}) - {}^{(q)}\mathbf{v}^{(q+1)}(\mathbf{x})\|_1.$$

Finally, using the triangle inequality and $\max \sum \leq \sum \max$, and combining everything,

$$\frac{1}{\rho} \max_{\mathbf{x} \in B} \|\mathbf{v}(\mathbf{x}) - \tilde{\mathbf{v}}(\mathbf{x})\|_1$$

$$\leq \frac{1}{\rho} \max_{\mathbf{x} \in B} \sum_{q=1}^{Q-1} \|^{(q-1)}\mathbf{v}(\mathbf{x}) - {}^{(q)}\mathbf{v}(\mathbf{x})\|_1$$

$$\leq \sum_{q=1}^{Q-1} \frac{1}{\rho} \max_{\mathbf{x} \in B} \|^{(q-1)}\mathbf{v}(\mathbf{x}) - {}^{(q)}\mathbf{v}(\mathbf{x})\|_1$$

$$\leq \sum_{q=1}^{Q-1} p_1(q+2, Q) \frac{1}{\rho} \max_{\mathbf{x} \in B} \|^{(q-1)}\mathbf{v}^{(q+1)}(\mathbf{x}) - {}^{(q)}\mathbf{v}^{(q+1)}(\mathbf{x})\|_1$$

$$\leq \sum_{q=1}^{Q-1} \left[ p_1(q+2, Q) \frac{\rho^{(q-1)}}{\rho} O(1)^q \left( \sqrt{n_{q+1}} \min \left\{ \sqrt{K_q} \|\mathbf{A}^{(q+1)}\|_F \|(\mathbf{A}^{(q)}, \mathbf{b}^{(q)})\|_F, \sum_{i=1}^{n_q} \frac{l_{k(i)}^{(q)}}{N_{min}} + \|\mathbf{E}_i^{(q)}\|_F \right\} \right. \right.$$
$$\left. \left. + \|(\mathbf{A}^{(q)}, \mathbf{b}^{(q)})\|_2 \sum_{j=1}^{n_{q+1}} \sum_{i \in N_j^{(q)}} |a_{ji}^{(q+1)}| \right) \right].$$

With

$$W(q) = p_1(q+2, Q) \left( p_2(1, q) + \sum_{q'=1}^{q} p_2(q'+1, q) \right) O(1)^q$$

we get

$$\frac{1}{\rho} \max_{\mathbf{x} \in B} \|\mathbf{v}(\mathbf{x}) - \tilde{\mathbf{v}}(\mathbf{x})\|_1 \leq \sum_{q=1}^{Q-1} \left[ W(q) \left( \sqrt{n_{q+1}} \min \left\{ \sqrt{K_q} \|\mathbf{A}^{(q+1)}\|_F \|(\mathbf{A}^{(q)}, \mathbf{b}^{(q)})\|_F, \sum_{i=1}^{n_q} \frac{l_{k(i)}^{(q)}}{N_{min}} + \|\mathbf{E}_i^{(q)}\|_F \right\} \right. \right.$$
$$\left. \left. + \|(\mathbf{A}^{(q)}, \mathbf{b}^{(q)})\|_2 \sum_{j=1}^{n_{q+1}} \sum_{i \in N_j^{(q)}} |a_{ji}^{(q+1)}| \right) \right],$$

which completes the proof.

As a last note, notice that Theorem 6 assumes a variant of TropNNC to obtain the bound. However, we have that

$$\sum_{j=1}^{m} \left\| \tilde{c}_{jk}(\tilde{\mathbf{a}}_k^T, \tilde{b}_k) - \sum_{i \in I_k} c_{ji}(\mathbf{a}_i^T, b_i) \right\|^2 \leq \sum_{j=1}^{m} \left\| |\tilde{c}_{jk}|(\tilde{\mathbf{a}}_k^T, \tilde{b}_k) - \sum_{i \in I_{jk}} |c_{ji}|(\mathbf{a}_i^T, b_i) \right\|^2 + \sum_{j=1}^{m} \left\| \sum_{i \in N_j} |c_{ji}|(\mathbf{a}_i^T, b_i) \right\|$$

$$\leq \sum_{j=1}^{m} \left\| |\tilde{c}_{jk}|(\tilde{\mathbf{a}}_k^T, \tilde{b}_k) - \sum_{i \in I_{jk}} |c_{ji}|(\mathbf{a}_i^T, b_i) \right\|^2 + \sum_{j=1}^{m} \sum_{i \in N_j} |c_{ji}| \left\| (\mathbf{a}_i^T, b_i) \right\|,$$

where the last term already appears in the bound of Theorem 6, and can be charged there with an additional $O(1) = 2$ factor. $\square$

