# OpenReview forum: "TropNNC: Structured Neural Network Compression Using Tropical Geometry"
_TMLR — Accepted by TMLR_

### Review · Reviewer_sY6m · 2026-03-15

**Summary Of Contributions:**

This paper considers the problem of compressing neural networks so that the neural networks contain fewer parameters, with the hope of accelerating the neural network training process. The main idea is that the paper finds that the output similarity depends on the Hausdorff distance in the field of tropical geometry, so that if we want to substitute the trained network with a new one that has a much smaller number of parameters, we need to minimize the Hausdorff distance between their sets of parameters. Based on this, the paper presents new compression methods using K-means clustering and proves the effectiveness of the method by characterizing the relationship between output similarity and the distance, under certain conditions on the clustering outcome. The paper also validates the proposed methods empirically through experiments on well-known benchmarks and demonstrates that the proposed methods outperform the baselines.

Strength

* The paper connects the neural network compression problem with the field of tropical geometry, deriving a new perspective.

* The findings of this paper, using Hausdorff distance to approximate the compression objective, which is also closely related to the parameter pruning problem, are new.

* The paper provides comprehensive theory to prove the effectiveness of using clustering as a tool for parameter compression.

* The paper also offers a suite of experimental results to support the empirical effectiveness of the method.

* The paper proposes new data-free algorithms given the observations of the connection between output similarity and Hausdorff distance.

**Additional Comments:**

NA

**Audience:**

Yes

**Audience Explanation:**

Neural network compression is a very emergent and relevant topic in the machine learning community, and thus there will definitely be audience who is interest in this paper.

**Claims And Evidence:**

Yes

**Claims Explanation:**

Based on my understanding, the main claims are supported by evidence, though I have doubts in the experiment part.

**Requested Changes:**

* The number of parameters is smaller, but what is the overhead of incorporating K-means clustering in these iterations? There does not seem to be a comparison of different methods with respect to their computation time or computational overhead, both in theory and in practice. Without this, it is hard to see whether this new algorithm actually improves training efficiency.

* The empirical performance does not seem to improve the benchmarks much, and the number of benchmarks may be limited to demonstrate the superiority of this proposed method.

* It is clear that the findings are new, but it would be helpful to explicitly lay out the novel proof techniques, which may better distinguish this work from the existing theory in tropical geometry. This will help the audience better appreciate the contribution of this work.

* How hard or easy is it in practice to choose the parameters in the K-means clustering?

* The novelty of the algorithm is relatively limited, as it essentially adds a clustering algorithm.

* The assumptions used in the theorems (e.g., Theorem 4) can be somewhat stringent. It would be great if the paper could first discuss how to verify these assumptions in a convenient way, and second, how broad or limited these assumptions are.

---

> ### Author Response · Authors · 2026-04-25
> **Response to Reviewer sY6m**
>
> We thank the reviewer for their careful reading and for highlighting several aspects of the work. We are glad that the novelty of the tropical geometric perspective and the theoretical contributions were recognized. In the revision, we have clarified the scope, strengthened the theoretical presentation, and improved empirical justification.
>
> Responses to the requested changes:
>
> **1. Computational overhead**
>
> We thank the reviewer for this question.
>
> We clarify that our method is a post-training, data-free compression method, and does not aim to accelerate training itself. Instead, its goal is to reduce model size and inference cost after training, without requiring access to data or retraining.
>
> Regarding computational overhead, the iterative update steps can be implemented efficiently using tensor operations (as described in the revised “Computational overhead” subsection). In practice, the dominant cost arises from the clustering step. Empirically, we find that the overall compression procedure is efficient, comparable to our baselines, and does not introduce prohibitive overhead. While we do not include a systematic runtime benchmark, we have added a dedicated discussion of computational cost in the appendix to clarify this point.
>
> **2. Benchmarks**
>
> We respectfully clarify that the goal of this work is not to achieve state-of-the-art compression performance, but to introduce a principled, theoretically grounded framework for data-free compression.
>
> Despite operating in a more constrained setting (no data), it remains competitive with stronger data-driven baselines. It consistently improves over prior tropical pruning approaches. It also positions tropical geometry as a potential tool for theoretically grounded compression techniques.
>
> We have adjusted the discussion to better reflect the intended scope of the method.
>
> **3. Proof techniques**
>
> We have expanded the discussion of the theoretical contributions to better highlight the key ideas and distinguish them from prior work in tropical geometry. In particular, we added brief intuitions for key results, and a concise summary of the main ideas in the discussion section.
>
> **4. Parameters of K-means**
>
> We thank the reviewer for raising this point.
>
> In practice, we found the method to be robust to K-means hyperparameters. In the end, we used standard settings to avoid introducing tuning bias. We also got comparable performance when using different implementations of K-means (e.g., a "hand-crafted" PyTorch implementation using just random initialization instead of kmeans++). These observations suggest that the method is robust to the clustering algorithm, and that performance is primarily driven by the tropical geometric formulation rather than specific implementation details.
>
> **5. Clustering algorithm novelty**
>
> We respectfully clarify that clustering is used as a tool, rather than being the primary contribution.
>
> The novelty of the work lies in the tropical geometric formulation of neural network compression as a simultaneous zonotope approximation problem, the use of Hausdorff distance to relate parameter approximation to output similarity, and the resulting theoretical framework and guarantees for compression.
>
> Clustering arises naturally within this framework as a mechanism for approximating the corresponding geometric objects.
>
> **6. Assumptions**
>
> We thank the reviewer for raising concerns about the assumptions.
>
> In the revision:
>
> * We show that the “no obtuse angle” assumption is in fact not required, and was an artifact of the original analysis. We have refined the proofs of Theorems 4 and 6 by removing this assumption entirely, without affecting the final bounds.
>
> * We also refine the assumptions in Theorem 7, clarifying their interpretation and providing empirical validation. We add a dedicated section discussing these assumptions, their practical relevance, and evaluate them in trained networks.
>
> We thank the reviewer again for the constructive feedback.

---

### Review · Reviewer_G1iC · 2026-03-29

**Summary Of Contributions:**

This paper introduces TropNNC, a novel structured neural network compression framework grounded in tropical geometry. Key contributions include:

1. Extending the tropical geometry pruning approach to convolutional layers and evaluating it on benchmark datasets (MNIST, CIFAR, ImageNet), where it demonstrates competitive performance against strong baselines like ThiNet and CUP.

2. Refining previous theoretical bounds by utilizing the continuous Hausdorff distance, yielding a tighter theoretical guarantee.

3. Providing a theoretical connection between the proposed iterative approximation method and Singular Value Decomposition (SVD) / spectral clustering.

The authors explicitly point out some weaknesses of this paper, such as

1. The empirical evaluation shows a significant performance leap on VGG but only modest benefits on ResNet18, a discrepancy the authors admit "remains somewhat unexplained".

2. "Heuristic 2" drops the bias term during clustering to prioritize unbounded regions extending to infinity. The paper lacks theoretical justification for why optimizing for infinite domains does not degrade performance on strictly bounded image data, explicitly admitting the heuristics are "not rigorously backed by theory".

3. Theorem 8 derives a highly informative layer-wise weight term $W(q)$ intended for a global sparsity allocation strategy. However, the authors explicitly state they did not implement this strategy, relying instead on uniform pruning or hierarchical clustering.

Since the authors proactively discussed these limitations, they should not constitute grounds for rejection.

**Audience:**

Yes

**Audience Explanation:**

The intersection of pure mathematics (tropical geometry) and practical deep learning optimization (model compression) is highly relevant to TMLR's audience. Researchers focusing on the theoretical understanding of neural networks will find the rigorous formulation of network layers as zonotopes and the associated Hausdorff distance bounds very insightful.

**Broader Impact Concerns:**

No further ethical concerns are raised.

**Claims And Evidence:**

Yes

**Claims Explanation:**

The claims made in the submission are largely supported by a rigorous theoretical foundation and substantial empirical evidence, though there are a few noticeable gaps connecting the theory to the experimental results.

**Requested Changes:**

Critical to securing recommendation for acceptance:

1. The theoretical results (Theorems 4 and 6) rely on the strong assumption that no two generators in a cluster form an obtuse angle. The paper does not empirically verify how often this condition holds in real, trained networks or how its violation impacts the practical error bounds. This assumption should be empirically analyzed or its necessity should be relaxed.

2. Theorem 7 heavily relies on the assumptions of "compact and well-separated clusters" and that "weight decay" successfully prevents anti-parallel neuron alignment. However, modern overparameterized networks often exhibit highly overlapping filter distributions. The authors should provide empirical metrics (e.g., Silhouette scores or angular distribution histograms of the filters) from a trained VGG or ResNet to prove that these strict geometric assumptions actually hold in practice.

Would Strengthen the Work:

1. Provide a hypothesis or ablation study exploring why TropNNC yields significant gains on VGG but only modest improvements on ResNet18  (e.g., how residual connections affect zonotope approximations).

2. Conduct experiments to investigate whether the proposed method remains feasible on deeper networks, such as ResNet-50.

---

> ### Author Response · Authors · 2026-04-25
> **Response to Reviewer G1iC**
>
> We thank the reviewer for their careful reading and constructive feedback. We appreciate the positive evaluation of the theoretical contributions and the recognition that the limitations were clearly acknowledged. In the revision, we have significantly strengthened the theoretical analysis, clarified assumptions, and added empirical validation where appropriate.
>
> Responses to the requested changes:
>
> **1. Assumptions (Theorems 4 and 6)**
>
> We thank the reviewer for highlighting the strength of this assumption.
>
> In the revised version, we show that the “no obtuse angle” assumption is in fact not required, and was an artifact of the original analysis. We have refined the proofs of Theorems 4 and 6 by removing this assumption entirely, without affecting the final bounds.
>
> **2. Assumptions (Theorem 7)**
>
> We agree that these assumptions require clarification and empirical grounding. In the revision, we refine these assumptions in the following ways:
>
>   * Regarding the "well-separated" and "weight-decay" assumptions, we weakened the "well-separated clusters", strengthened the "weight-decay", and merged them into one unified assumption. We have added a new section explaining, motivating, and verifying the empirical soundness of this assumption. Empirically, we observe that it is satisfied for different networks across all pruning ratios.
>
>   * Regarding the "compact clusters" assumption, we note that this assumption is best interpreted as a condition on the compressibility of the network. Higher cluster compactness corresponds to more compressible networks. Empirically, we observe a clear relationship between cluster compactness and the performance of the compressed model across architectures.
>
> Also note that we refined the discussion of the connection to SVD to reflect the revised assumptions. In the original version, the interpretation relied on approximate orthogonality/"well-separated clusters". With the revised assumptions, we instead interpret the method as a low-rank approximation of the non-linear operator C ReLU (A,b).
>
> **3. VGG vs ResNets**
>
> We agree that understanding this behavior is important.
>
> Due to resource constraints, we do not include additional ablations in this revision. Instead, we provide two theoretically grounded hypotheses, supported by our analysis' viewpoint and by prior work, to explain the observed behavior. These are detailed in the Appendix, together with the Limitations.
>
> The first hypothesis provides a principled way for our method to be applied to residual blocks by fusing the shortcut with the residual path. We explain how the corresponding zonotopes change with this step. However, this fusion requires increasing the number of hidden neurons by 2 times the size of the input. These new neurons have very sparse weights, and they do not count toward the total parameter count (i.e., the shortcut is given "for free"). After compression, their weights change and stop being sparse. This means that we have to compensate by pruning these additional neurons before we start having true reduction with respect to the initial network, suggesting that, under this formulation, the method may be less effective for residual architectures.
>
> The second hypothesis considers the approach of leaving the shortcut as is. With an SVD viewpoint, we demonstrate how there could exist a discrepancy between the redundant eigenvalues of the residual path alone, and the redundant eigenvalues of the block as a whole. Using the ODE perspective of prior work for residual blocks, we reason that the conditions for this discrepancy to hold are not unlikely.
>
> We note that direct ablation is nontrivial. For example, measuring “compressibility” is itself closely tied to the outcome of compression, and may not isolate the underlying cause. We therefore consider these hypotheses as a first step toward understanding this behavior, and leave empirical validation for future work.
>
> We have adjusted both the "Limitations" paragraph, and the discussion section, to reflect this architectural limitation.
>
> **4. Deeper models**
>
> We agree that evaluating on deeper architectures would further strengthen the empirical study.
>
> Due to computational constraints, we did not include experiments on models of size comparable to ResNet-50 in this revision. Instead, we focused on expanding the discussion of the effects of residual paths. We consider evaluation on larger models an important direction for future work.
>
> As a last note, in the discussion we also addressed Weakness 2. Regions extending to infinity are high-error. Although in trained networks the signals are bounded, this might not be the case after compression. Pruning methods can be considered as applying a contractive projection, and thus signal scales do not explode. In our method, we need Theorem 7 to ensure this. This is still a worst-case guarantee, however, and Heuristic 2 also helps in practice.
>
> We thank the reviewer again for the constructive feedback.

---

### Review · Reviewer_Ucng · 2026-04-10

**Summary Of Contributions:**

This paper studies neural network compression through a very unusual and interesting lens, namely tropical geometry. The authors propose TropNNC, a structured and data-free compression method that views ReLU-type networks in terms of tropical rational functions and then compresses them by approximating the corresponding tropical geometric objects. Beyond the core method itself, the paper also develops stronger approximation results than prior tropical pruning work, introduces improved cluster representatives and an iterative refinement procedure, and extends the framework to settings that matter more in practice, especially convolutional layers and batch-normalization-fused networks.

What I liked most about the paper is that it does not feel like just another pruning heuristic. The authors are clearly trying to build a principled geometric and mathematical story for compression, and I appreciated that ambition. To me, one of the main strengths is precisely that the paper combines a fairly novel theoretical perspective with a method that is still evaluated seriously on standard benchmarks like MNIST, CIFAR, and ImageNet. I also liked that the paper goes beyond fully connected toy settings and makes a genuine effort to show relevance for more practical architectures.

The main strengths, from my perspective, are the originality of the viewpoint, the amount of theory provided relative to most compression papers, and the fact that the method seems to improve meaningfully over earlier tropical pruning approaches while sometimes being competitive with stronger baselines. The main weaknesses are that some parts of the theory rely on assumptions that feel fairly strong, and the empirical story is a bit uneven across architectures. In particular, the paper seems strongest on VGG-style models, while the gains on ResNet are more mixed, and I think that part still needs a clearer explanation. Overall though, I found the paper interesting, thoughtful, and genuinely distinct from a lot of more routine work in this area.

**Audience:**

Yes

**Audience Explanation:**

To me, one of the paper’s main appeals is that it feels genuinely different from a lot of routine pruning work. Rather than introducing another mostly empirical heuristic, the authors build the method around a tropical-geometric perspective and use that perspective to motivate both the algorithm and its approximation guarantees. I think that makes the paper interesting not only as a compression result, but also as a conceptual contribution.

I also liked that the paper tries to make this perspective matter in more realistic settings. Compared with earlier tropical pruning work, this version feels more practical, especially through the extension to convolutional layers and the experimental comparisons on standard datasets. As a result, I think the paper has relevance both for people who care about principled ML methodology and for those interested in concrete compression methods.

**Claims And Evidence:**

Yes

**Claims Explanation:**

On the theoretical side, the paper contains substantial technical development rather than just high-level intuition. It provides refined approximation results, develops both single-output and multi-output compression procedures, and gives explicit error bounds for the approximations it constructs. That theoretical component is clearly one of the paper’s central strengths, and I appreciated that the authors try to connect the geometric framework back to practical compression.

On the empirical side, I think the evidence supports the paper’s more measured practical claims. In particular, the experiments show that TropNNC improves clearly over prior tropical pruning methods and can be competitive with stronger baselines even in a retraining-free, data-free setting. The results on MNIST, CIFAR, and the non-uniform pruning experiments suggest that this is not just a theoretically interesting idea, but a method with genuine practical value.

At the same time, I was less convinced by some of the broader interpretive claims. Parts of the discussion, such as the connection to SVD or the broader lessons the paper draws for pruning more generally, felt more suggestive than fully established by the current evidence. I also think the architecture-dependent behavior deserves more explanation, since the method appears especially strong on VGG-style networks but more mixed on ResNet. So overall, I do think the main technical and empirical claims are supported, but some of the broader conclusions would benefit from more qualification and stronger ablation or analysis.

**Requested Changes:**

I have a few suggestions that I think would improve the paper.

First, I would encourage the authors to calibrate some of the broader claims a bit more carefully. The core method and results are interesting, but in a few places the discussion moves quickly from what is directly established to broader conclusions about pruning or compression more generally. I think the paper would benefit from a clearer separation between what is rigorously shown, what is empirically demonstrated, and what is more of a conceptual interpretation.

Second, I would like to see a clearer discussion of the assumptions underlying the theory and how they relate to practice. The theoretical development is one of the strongest aspects of the paper, but some of the assumptions seem fairly strong, and it would help readers if the authors were more explicit about when these assumptions are expected to hold and how much they matter in realistic settings.

Third, I think the paper should say more about the architecture-dependent behavior of the method. The results appear especially strong on VGG-style models, while the gains on ResNet are more mixed. Since this affects how broadly one should interpret the practical impact of the approach, some additional analysis or discussion here would make the paper stronger.

Fourth, the empirical section would benefit from stronger ablations. There are several components and design choices in the method, and it would be useful to understand more clearly which parts are driving the gains.

Fifth, I think the positioning of the baselines could be clarified further. Since the paper emphasizes that the method is data-free and retraining-free, it would help to be very explicit about which comparison methods rely on data, retraining, or different assumptions, so that readers can interpret the comparisons more easily.

Finally, I think the presentation could be improved in some of the denser theoretical sections. The ideas are interesting and substantial, but a somewhat cleaner exposition and a more direct explanation of the practical meaning of the main results would make the paper easier to follow.

---

> ### Author Response · Authors · 2026-04-25
> **Response to Reviewer Ucng (Part 1/2)**
>
> (Part 1/2)
>
> We thank the reviewer for their thoughtful and detailed feedback. We appreciate the positive assessment of the paper’s originality, theoretical depth, and the effort to connect theoretical tropical geometry with practical network compression. We have addressed the requested changes, and believe the revision improves the positioning, clarity, and connection between theory and practice.
>
> Responses to the requested changes:
>
> **1. Calibration of broader claims**
>
> We agree that clearer calibration improves the paper. In the revision, we are more explicit about our conceptual interpretations. In particular, the discussion section now focuses more closely on interpretations grounded in our method, and statements relating to broader implications for pruning are presented more cautiously.
>
> **2. Assumptions**
>
> We thank the reviewer for highlighting this important point. We refined our assumptions to make them more broadly applicable in practice.
>
> In Theorems 4 and 6, we show that the “no obtuse angle” assumption was in fact an artifact of the original analysis. We remove it entirely in the revision, without affecting the final bounds.
>
> For the assumptions of Theorem 7 (clustering of good quality, weight-decay), we aligned them with what holds in practice:
>
>   * Regarding the "well-separated" and "weight-decay" assumptions, we weakened the "well-separated clusters", strengthened the "weight-decay", and merged them into one unified assumption. We have added a new section explaining, motivating, and verifying the empirical soundness of this assumption. Empirically, we observe that it is satisfied for different networks all pruning ratios. Following these refinements, we updated the proof of Theorem 7 accordingly. The overall structure of the argument remains the same.
>
>   * Regarding the "compact clusters" assumption, we note that this is best interpreted as a condition on the compressibility of the network. Higher cluster compactness corresponds to more compressible networks. Empirically, we observe a clear relationship between cluster compactness and the performance of the compressed model across architectures.
>
> Also note that we refined the discussion of the connection to SVD. In the original version, the interpretation relied on "well-separated clusters". With the revised assumptions, we instead interpret the method as a low-rank approximation of the non-linear operator $C \cdot ReLU \cdot (A,b)$. When misalignment is absent, this reduces to a setting analogous to approximate SVD directly on the linear $C(A,b)$. However, more generally, linear SVD may perform suboptimally due to cancellation, whereas our method avoids it by maintaining 2 different generators not leading to cancellation. We have updated the corresponding section to reflect this more general and accurate interpretation.
>
> **3. Architecture-dependent behavior**
>
> We agree that understanding the architecture-dependent behavior (e.g., VGG vs. ResNet) is important.
>
> Due to resource constraints, we do not include additional ablations in this revision. Instead, we provide two theoretically grounded hypotheses, supported by our analysis' viewpoint and by prior work, to explain the observed behavior. These are detailed in the Appendix, together with the Limitations.
>
> The first hypothesis provides a principled way for our method to be applied to residual blocks by fusing the shortcut with the residual path. We explain how the corresponding zonotopes change with this step. However, this fusion requires increasing the number of hidden neurons by 2 times the size of the input. These new neurons have very sparse weights, and they do not count toward the total parameter count (i.e., the shortcut is given "for free"). After compression, their weights change and stop being sparse. This means that we have to compensate by pruning these additional neurons before we start having true reduction with respect to the initial network, suggesting that, under this formulation, the method may be less effective for residual architectures.
>
> The second hypothesis considers the approach of leaving the shortcut as is. With an SVD viewpoint, we demonstrate how there could exist a discrepancy between the redundant eigenvalues of the residual path alone, and the redundant eigenvalues of the block as a whole. Using the ODE perspective of prior work for residual blocks, we reason that the conditions for this discrepancy to hold are not unlikely.
>
> We note that direct ablation is nontrivial. For example, measuring “compressibility” is itself closely tied to the outcome of compression, and may not isolate the underlying cause. We therefore consider these hypotheses as a first step toward understanding this behavior, and leave empirical validation for future work.
>
> We have adjusted both the "Limitations" paragraph, and the discussion section, to reflect this architectural limitation.
>
> (End of part 1/2)

---

> > ### Author Response · Authors · 2026-04-25
> > **Response to Reviewer Ucng (Part 2/2)**
> >
> > (Part 2/2)
> >
> > **4. Ablations**
> >
> > We agree that ablations are valuable for understanding the contribution of individual components.
> >
> > Due to time and computational constraints, we did not include a full ablation study in this revision. However, during development we observed the following:
> >
> >  * The method was in general robust to variations of K-means parameters. In the end, we used standard settings to avoid introducing additional tuning bias.
> >  * We also got comparable performance when using different implementations of K-means (e.g., a "hand-crafted" PyTorch implementation using just random initialization instead of kmeans++).
> >
> > These observations suggest that the method is robust to the clustering algorithm, and that performance is primarily driven by the tropical geometric formulation rather than specific implementation details.
> >
> > **5. Positioning of the baselines**
> >
> > We clarify that ThiNet is data-driven and leverages access to data, while the remaining methods (including TropNNC) are strictly data-free. No retraining or fine-tuning is used for any method. Comparisons should therefore be interpreted with this difference in mind. We have added an explicit statement clarifying this in the Experiments section.
> >
> > **6. Presentation**
> >
> > We improved the exposition by adding brief intuitions for key results, and by including a concise summary of the main proof ideas in the discussion section.
> >
> > We thank the reviewer again for the constructive feedback. We hope the updated version addresses the concerns.
> >
> > (End of Part 2/2)

---

> > > ### Comment · Reviewer_Ucng · 2026-05-03
> > > **Response to authors**
> > >
> > > Thank you for the detailed response. The revisions seem to address the main concerns in a substantive way. The changes to the theoretical assumptions, especially removing the no obtuse angle assumption and refining the assumptions behind Theorem 7, make the theory clearer and better calibrated.
> > >
> > > I like the new clarification of the baselines and the discussion of VGG versus ResNet gives me a better way to interpret the architectural dependence of the results. I still think this point, along with the ablation question, is only partially resolved, since the response mainly adds explanations rather than new empirical evidence. However, the added discussion is useful and makes the limitations clearer! Thanks and good work:)!

---

### Decision · Action_Editor_LG5W · 2026-05-19

**Recommendation:** Accept as is

**Audience:**

Yes

**Audience Explanation:**

The results are not currently state-of-the-art, but this is not a criterion for acceptance to TMLR, while I do believe that both the reviews and the response above cover why this is an interesting paper for the journal.

**Claims And Evidence:**

Yes

**Claims Explanation:**

The paper introduces TropNNC, a (training) data-free compression methodology that aims to compress networks with linear and convolutional layers by modeling neural network outputs as tropical rational functions. The work establishes a rigorous mathematical foundation for pruning that successfully extends to convolutional architectures. The transition from pure geometry to applied machine learning reveals several fundamental gaps in the current manuscript as the reviewers pointed out. However, after the rebuttal, the manuscript calibrated the claims and is now ready for acceptance.